# Temporally delayed linear modelling (TDLM) measures replay in both animals and humans

Yunzhe Liu[1,2,3]*, Raymond J Dolan[1,3,4], Cameron Higgins[5], Hector Penagos[6], Mark W Woolrich[5], H Freyja Ólafsdóttir[7], Caswell Barry[8], Zeb Kurth-Nelson[3,9], Timothy E Behrens[4,5]

[1]State Key Laboratory of Cognitive Neuroscience and Learning, IDG/McGovern Institute for Brain Research, Beijing Normal University, Beijing, China; [2]Chinese Institute for Brain Research, Beijing, China; [3]Max Planck University College London Centre for Computational Psychiatry and Ageing Research, London, United Kingdom; [4]Wellcome Centre for Human Neuroimaging, University College London, London, United Kingdom; [5]Wellcome Centre for Integrative Neuroimaging, University of Oxford, Oxford, United Kingdom; [6]Center for Brains, Minds and Machines, Picower Institute for Learning and Memory, Department of Brain and Cognitive Sciences, Massachusetts Institute of Technology, Cambridge, United States; [7]Donders Institute for Brain Cognition and Behaviour, Radboud University, Nijmegen, Netherlands; [8]Research Department of Cell and Developmental Biology, University College London, London, United Kingdom; [9]DeepMind, London, United Kingdom

**Abstract** There are rich structures in off-task neural activity which are hypothesized to reflect fundamental computations across a broad spectrum of cognitive functions. Here, we develop an analysis toolkit – temporal delayed linear modelling (TDLM) – for analysing such activity. TDLM is a domain-general method for finding neural sequences that respect a pre-specified transition graph. It combines nonlinear classification and linear temporal modelling to test for statistical regularities in sequences of task-related reactivations. TDLM is developed on the non-invasive neuroimaging data and is designed to take care of confounds and maximize sequence detection ability. Notably, as a linear framework, TDLM can be easily extended, without loss of generality, to capture rodent replay in electrophysiology, including in continuous spaces, as well as addressing second-order inference questions, for example, its temporal and spatial varying pattern. We hope TDLM will advance a deeper understanding of neural computation and promote a richer convergence between animal and human neuroscience.

*For correspondence:
yunzhe.liu@bnu.edu.cn

## Introduction

Human neuroscience has made remarkable progress in detailing the relationship between the representations of different stimuli during task performance (*Haxby et al., 2014*; *Kriegeskorte et al., 2008*; *Barron et al., 2016*). At the same time, it is increasingly clear that resting, off-task, brain activities are structurally rich (*Smith et al., 2009*; *Tavor et al., 2016*). An ability to study spontaneous activity with respect to task-related representation is important for understanding cognitive process beyond current sensation (*Higgins et al., 2021*). However, unlike the case for task-based activity, little attention has been given to techniques that can measure representational content of resting brain activity in humans.

Unlike human neuroscience, representational content of resting activity is studied extensively in animal neuroscience. One seminal example is 'hippocampal replay' (*Wilson and McNaughton, 1994*; *Skaggs and McNaughton, 1996*; *Louie and Wilson, 2001*; *Lee and Wilson, 2002*): during sleep, and quiet wakefulness, place cells in the hippocampus (that signal self-location during periods of activity) spontaneously recapitulate old, and explore new, trajectories through an environment. These internally generated sequences are hypothesized to reflect a fundamental feature of neural computation across tasks (*Foster, 2017*; *Ólafsdóttir et al., 2018*; *Pfeiffer, 2020*; *Carr et al., 2011*; *Lisman et al., 2017*). Numerous methods have been proposed to analyse hippocampal replay (*Davidson et al., 2009*; *Grosmark and Buzsáki, 2016*; *Maboudi et al., 2018*). However, they are not domain general in that they are designed to be most suited for specific needs, such as particular task design, data modality, or research question (*van der Meer et al., 2020*; *Tingley and Peyrache, 2020*). Most commonly, these methods apply to invasive electrophysiology signals, aiming to detect sequences in a linear track during spatial navigation task (*Tingley and Peyrache, 2020*). As a result, they cannot be directly adapted for analysing human resting activity collected using non-invasive neuroimaging techniques. Furthermore, in rodent neuroscience, it is non-trivial to adapt these algorithms to even small changes in tasks (such as 2D foraging). This may be a limiting factor in taking replay analyses to more interesting and complex tasks, such as complex mazes (*Rosenberg et al., 2021*).

Here, we introduce temporal delayed linear modelling (TDLM), a domain-general analysis toolkit, for characterizing temporal structure of internally generated neural representations in rodent electrophysiology as well as human neuroimaging data. TDLM is inspired by existing replay detection methods (*Skaggs and McNaughton, 1996*; *Davidson et al., 2009*; *Grosmark and Buzsáki, 2016*), especially those analysis of population of replay events (*Grosmark and Buzsáki, 2016*). It is developed based on the general linear modelling (GLM) framework and can therefore easily accommodate testing of 'second-order' statistical questions (*van der Meer et al., 2020*), such as whether there is more forward than reverse replay, or is replay strength changing over time, or differs between behavioural conditions. This type of question is ubiquitous in cognitive studies, but is typically addressed ad hoc in other replay detection methods (*van der Meer et al., 2020*). In TDLM, such questions are treated naturally as linear contrasts of effects in a GLM.

Here, we show TDLM is suited to measure the average amount of replay across many events (i.e. replay strength) in linear modelling. This makes it applicable to both rodent electrophysiology and human neuroimaging. Applying TDLM on non-invasive neuroimaging data in humans, we, and others, have shown it is possible to measure the average sequenceness (propensity for replay) in spontaneous neural representations (*Wimmer et al., 2020*; *Nour et al., 2021*; *Liu et al., 2019*; *Liu et al., 2021a*). The results resemble key characteristics found in rodent hippocampal replay and inform key computational principles of human cognition (*Liu et al., 2019*).

In the following sections, we first introduce the logic and mechanics of TDLM in detail, followed by a careful treatment of its statistical inference procedure. We test TDLM in both simulation (see section 'Simulating MEG data') and real human MEG/EEG data (see section 'Human replay dataset'). We then turn to rodent electrophysiology and compare TDLM to existing rodent replay methods, extending TDLM to work on a continuous state space. Lastly, using our approach we re-analyse rodent electrophysiology data from *Ólafsdóttir et al., 2016* (see section 'Rodent replay dataset') and show what TDLM can offer uniquely compared to existing methods in rodent replay analysis.

To summarize, TDLM is a general, and flexible, tool for measuring neural sequences. It facilitates cross-species investigations by linking large-scale measurements in humans to single-neuron measurements in non-human species. It provides a powerful tool for revealing abstract cognitive processes that extend beyond sensory representation, potentially opening doors for new avenues of research in cognitive science.

## Results

### Temporal delayed linear modelling

#### Overview of TDLM

Our primary goal is to test for temporal structure of neural representations in humans. However, to facilitate cross-species investigation (*Barron et al., 2021*), we also want to extend this method to

enable measurement of sequences in other species (e.g. rodents). Consequently, this sequence detection method has to be domain general. We chose to measure sequences in a decoded state space (e.g. posterior estimated locations in rodents [*Grosmark and Buzsáki, 2016*] or time course of task-related reactivations in humans [*Liu et al., 2019*]) as this makes results from different data types comparable.

Ideally, a general sequence detection method should (1) uncover structural regularities in the reactivation of neural activity, (2) control for confounds that are not of interest, and (3) test whether this regularity conforms to a hypothesized structure. To achieve these goals, we developed the method under a GLM framework, and henceforth refer to it as temporal delayed linear modelling, that is, TDLM. Although TDLM works on a decoded state space, it still needs to take account of confounds inherent in the data where the state space is decoded from. This is a main focus of TDLM.

The starting point of TDLM is a set of $n$ time series, each corresponding to a decoded neural representation of a task variable of interest. This is what we call the state space, $X$, with dimension of time by states. These time series could themselves be obtained in several ways, described in detail in a later section ('Getting the states'). The aim of TDLM is to identify task-related regularities in sequences of these representations.

Consider, for example, a task in which participants have been trained such that $n = 4$ distinct sensory objects (A, B, C, and D) appear in a consistent order : $A \rightarrow B \rightarrow C \rightarrow D$ (*Figure 1a, b*). If we are interested in replay of this sequence during subsequent resting periods (*Figure 1c, d*), we might want to ask statistical questions of the following form: 'Does the existence of a neural representation of A, at time T, predict the occurrence of a representation of B at time T+ $\Delta t$?' and similarly for $B \rightarrow C$ and $C \rightarrow D$.

In TDLM, we ask such questions using a two-step process. First, for each of the $n^2$ possible pairs of variables $X_i$ and $X_j$, we find the linear relation between the $X_i$ time series and the $\Delta t$-shifted $X_j$ time series. These $n^2$ relations comprise an empirical transition matrix, describing how likely each variable is to be succeeded at a lag of $\Delta t$ by each other variable (*Figure 1e*). Second, we linearly relate this empirical transition matrix with a task-related transition matrix of interest (*Figure 1f*). This produces a single number that characterizes the extent to which the neural data follow the transition matrix of interest, which we call 'sequenceness'. Finally, we repeat this entire process for all $\Delta t$ of interest, yielding a measure of sequenceness at each possible lag between variables and submit this for statistical inference (*Figure 1g*).

Note that, for now, this approach decomposes a sequence (such as $A \rightarrow B \rightarrow C \rightarrow D$) into its constituent transitions and sums the evidence for each transition. Therefore, it does not require that the transitions themselves are sequential: $A \rightarrow B$ and $B \rightarrow C$ could occur at unrelated times, so long as the within-pair time lag was the same. For interested readers, we address how to strengthen the inference by looking explicitly for longer sequences in Appendix 1: Multi-step sequences.

## Constructing the empirical transition matrix

In order to find evidence for state-to-state transitions at some time lag $\Delta t$, we could regress a time-lagged copy of one state, $X_j$, onto another, $X_i$ (omitting residual term $\varepsilon$ in all linear equations):

$$X_j(t + \Delta t) = X_i(t)\beta_{ij} \tag{1}$$

Instead, TDLM chooses to include all states in the same regression model for important reasons, detailed in section 'Moving to multiple linear regression':

$$X_j(t + \Delta t) = \sum_{k=1}^{n} X_k(t)\beta_{kj} \tag{2}$$

In this equation, the values of all states $X_k$ at time $t$ are used in a single multilinear model to predict the value of the single state $X_j$ at time $t + \Delta t$.

The regression described in *Equation 2* is performed once for each $X_j$, and these equations can be arranged in matrix form as follows:

$$X(\Delta t) = X\beta \tag{3}$$

Each row of $X$ is a time point, and each of the $n$ columns is a state. $X(\Delta t)$ is the same matrix as $X$,

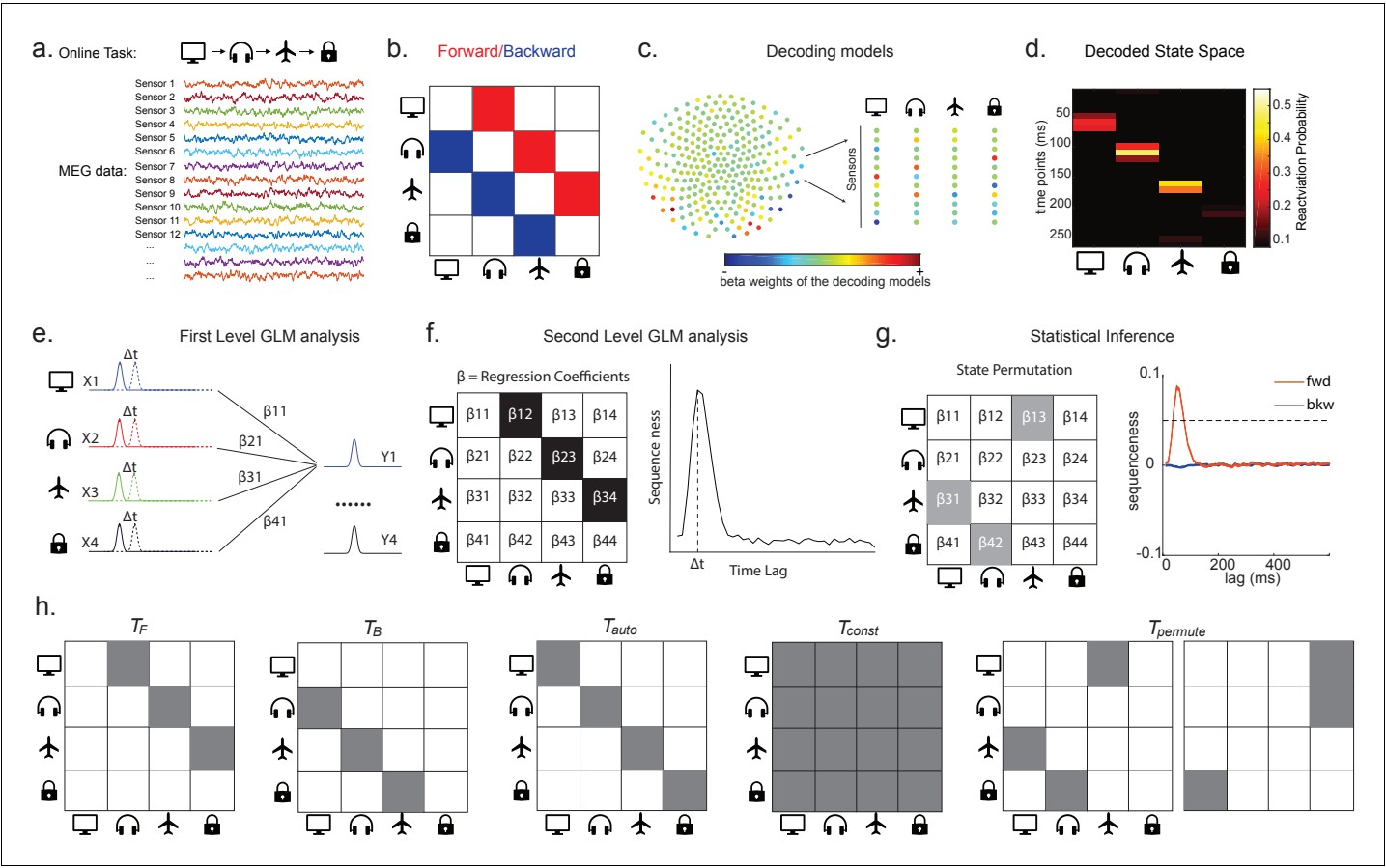

**Figure 1.** Task design and illustration of temporal delayed linear modelling (TDLM).  (a) Task design in both simulation and real MEG data. Assuming there is one sequence, A->B->C->D, indicated by the four objects at the top. During the task, participants are shown the objects and asked to figure out a correct sequence for these objects while undergoing MEG scanning. A snapshot of MEG data is shown below. It is a matrix with dimensions of sensors by time. (b) The transitions of interest are shown, with the red and blue entries indicating transitions in the forward and backward direction, respectively. (c) The first step of TDLM is to construct decoding models of states from task data, and (d) then transform the data (e.g. resting-state) from sensor space to the state space. TDLM works on the decoded state space throughout. (e) The second step of TDLM is to quantify the temporal structure of the decoded states using multiple linear regressions. The first-level general linear modelling (GLM) results in a state*state regression coefficient matrix (empirical transition matrix), β, at each time lag. (f) In the second-level GLM, this coefficient matrix is projected onto the hypothesized transition matrix (black entries) to give a single measure of sequenceness. Repeating this process for the number of time lags of interest generates sequenceness over time lags (right panel). (g) The statistical significance of sequenceness is tested using a non-parametric state permutation test by randomly shuffling the transition matrix of interest (in grey). To control for multiple comparisons, the permutation threshold is defined as the 95th percentile of all shuffles on the maximum value over all tested time lags. (h) The second-level regressors $T_{auto}$, $T_{const}$, $T_F$, and $T_B$, as well as two examples of the permuted transitions of interest, $T_{permute}$ (for constructing permutation test), are shown.

The online version of this article includes the following figure supplement(s) for figure 1:

**Figure supplement 1.** Source localization of stimuli-evoked neural activity in MEG.

but with the rows shifted forwards in time by $\Delta t$. $\beta_{ij}$ is an estimate of the influence of $X_i(t)$ on $X_j(t + \Delta t)$. $\beta$ is an $n \times n$ matrix of weights, which we call the *empirical transition matrix*.

To obtain $\beta$, we invert *Equation 3* by ordinary least squares regression:

$$\beta = \left(X^T X\right)^{-1} X^T X(\Delta t) \tag{4}$$

This inversion can be repeated for each possible time lag ($\Delta t = 1, 2, 3, \ldots$), resulting in a separate empirical transition matrix $\beta$ at every time lag. We call this step the first-level sequence analysis.

## Testing the hypothesized transitions

The first-level sequence analysis assesses evidence for all possible state-to-state transitions. The next step in TDLM is to test for the strength of a particular hypothesized sequence, specified as a transition matrix, $T$. Therefore, we construct another GLM which relates $T$ to the empirical transition matrix, $\beta$. We call this step the second-level sequence analysis:

$$\beta = \sum_{r=1}^{r} Z(r) * T_r \tag{5}$$

As noted above, $\beta$ is the empirical transition matrix obtained from first-stage GLM. It has dimension of $n$ by $n$, where $n$ is the number of states. Each entry in $\beta$ reflects the unique contribution of state $i$ to state $j$ at given time lag. Effectively, the above equation models this empirical transition matrix $\beta$ as a weighted sum of *prespecified template matrices*, $T_r$. Thus, $r$ is the number of regressors included in the second-stage GLM, and each scalar valued $Z(r)$ is the weight assigned to the $r$ th template matrix. Put in words, $T_r$ constitutes the regressors in the design matrix, each of which has a prespecified template structure, for example, $T_{auto}$, $T_{const}$, $T_F$, and $T_B$ (**Figure 1h**).

$T_F$ and $T_B$ are the transpose of each other (e.g. red and blue entries in **Figure 1b**), indicating transitions of interest in forward and backward direction, respectively. In 1D physical space, $T_F$ and $T_B$ would be the shifted diagonal matrices with ones on the first upper and lower off diagonals. $T_{const}$ is a constant matrix that models away the average of all transitions, ensuring that any weight on $T_F$ and $T_B$ reflects its unique contribution. $T_{auto}$ is the identity matrix. $T_{auto}$ models self-transitions to control for autocorrelation (equivalently, we could simply omit the diagonal elements from the regression).

$Z$ is the weights of the second-level regression, which is a vector with dimension of 1 by $r$. Each entry in $Z$ reflects the strength of the hypothesized transitions in the empirical ones, that is, sequenceness. Repeating the regression of **Equation 5** at each time lag ($\Delta t = 1, 2, 3, \ldots$) results in time courses of the sequenceness as a function of time lag (e.g. the solid black line in **Figure 1f**). $Z_F$, $Z_B$ are the forward and backward sequenceness, respectively (e.g. red and blue lines in **Figure 1g**).

In many cases, $Z_F$ and $Z_B$ will be the final outputs of a TDLM analysis. However, it may sometimes also be useful to consider the quantity:

$$D = Z_F - Z_B \tag{6}$$

$D$ contrasts forward and backward sequences to give a measure that is positive if sequences occur mainly in a forward direction and negative if sequences occur mainly in a backward direction. This may be advantageous if, for example, $Z_F$ and $Z_B$ are correlated across subjects (due to factors such as subject engagement and measurement sensitivity). In this case, $D$ may have lower cross-subject variance than either $Z_F$ or $Z_B$ as the subtraction removes common variance.

Finally, to test for statistical significance, TDLM relies on a non-parametric permutation-based method. The null distribution is constructed by randomly shuffling the identities of the $n$ states many times and re-calculating the second-level analysis for each shuffle (**Figure 1g**). This approach allows us to reject the null hypothesis that there is no relationship between the empirical transition matrix and the task-defined transition of interest. Note that there are many incorrect ways to perform permutations, which permute factors that are not exchangeable under the null hypothesis and therefore lead to false positives. We examine some of these later with simulations and real data. In some cases, it may be desirable to test slightly different hypotheses by using a different set of permutations; this is discussed later.

If the time lag $\Delta t$ at which neural sequences exist is not known a priori, then we must correct for multiple comparisons over all tested lags. This can be achieved by using the maximum $Z_F$ across all tested lags as the test statistic (see details in section 'Correcting for multiple comparisons'). If we choose this test statistic, then any values of $Z_F$ exceeding the 95th percentile of the null distribution can be treated as significant at $\alpha = 0.05$ (e.g. the grey dotted line in **Figure 1g**).

## TDLM steps in detail

### Getting the states

As described above, the input to TDLM is a set of time series of decoded neural representations or states. Here, we provide different examples of specific state spaces (*X*, with dimension of time by states) that we have worked with using TDLM.

## States as sensory stimuli

The simplest case, perhaps, is to define a state in terms of a neural representation of sensory stimuli, for example, face, house. To obtain their associated neural representation, we present these stimuli in a randomized order at the start of a task and record whole-brain neural activity using a non-invasive neuroimaging method, for example, Magnetoencephalography (MEG) or Electroencephalography (EEG). We then train a model to map the pattern of recorded neural activity to the presented image (*Figure 1—figure supplement 1*). This could be any of the multitude of available decoding models. For simplicity, we used a logistic regression model throughout.

The states here are defined in terms of stimuli-evoked neural activity. The classifiers are trained at 200 ms post-stimulus onset. For example, the stimuli are faces, buildings, body parts, and objects. Source localizing the evoked neural activity, we found that the activation patterns of stimuli in MEG signal are consistent with those reported in fMRI literature. For faces, activation peaked in a region roughly consistent with the fusiform face area (FFA) as well as the occipital face area (OFA). Activation for building stimuli was located between a parahippocampal place area (PPA) and retrosplenial cortex (RSC), a region also known to respond to scene and building stimuli. Activation for body part stimuli localized to a region consistent with the extrastriate body area (EBA). Activation for objects was in a region consistent with an object-associated lateral occipital cortex (LOC) as well as an anterior temporal lobe (ATL) cluster that may relate to conceptual processing of objects. Those maps are thresholded to display localized peaks. The full un-thresholded maps can be found at https://neurovault.org/collections/6088/. This is adapted from *Wimmer et al., 2020*.

In MEG/EEG, neural activity is recorded by multiple sensor arrays on the scalp. The sensor arrays record whole-brain neural activity at millisecond temporal resolution. To avoid a potential selection bias (given the sequence is expressed in time), we choose whole-brain sensor activity at a single time point (i.e. spatial feature) as the training data fed into classifier training.

Ideally, we would like to select a time point where the neural activity can be most truthfully read out. This can be indexed as the time point that gives the peak decoding accuracy. If the state is defined by the sensory features of stimuli, we can use a classical leave-one-out cross-validation scheme to determine the ability of classifiers to generalize to unseen data of the same stimulus type (decoding accuracy) at each time point (see Appendix 2 for its algorithm box). In essence, this cross-validation scheme is asking whether the classifier trained on this sensory feature can be used to classify the unseen data of the same stimuli (*Figure 2a, b*).

After we have identified the peak time point based on the cross-validation, we can train the decoding models based on the multivariate sensor data at this given time.

Specifically, let us denote the training data, *M*, with dimension of number of observations, *b*, by number of sensors, *s*. The labels, *Y*, have dimension of *b* by 1. The aim here is to obtain the classifier weights, W, so that $Y \approx \sigma(\mathrm{MW})$. $\sigma$ is the logistic sigmoid function.

Normally we apply L1 regularization on the inference of weights (we will detail the reasons in section 'Regularization'):

$$W = \underset{W}{\mathrm{argmax}}[\log(P(Y|M,W)) + b\lambda_{L1}||W||_1] \qquad (7)$$

Next, we translate the data at testing time (e.g. during rest), R, from sensor space to the decoded state space:

$$X = \sigma(\mathrm{RW}) \qquad (8)$$

where R is the testing data, with dimension of time by sensors, and X is the decoded state space, with dimension of time by states.

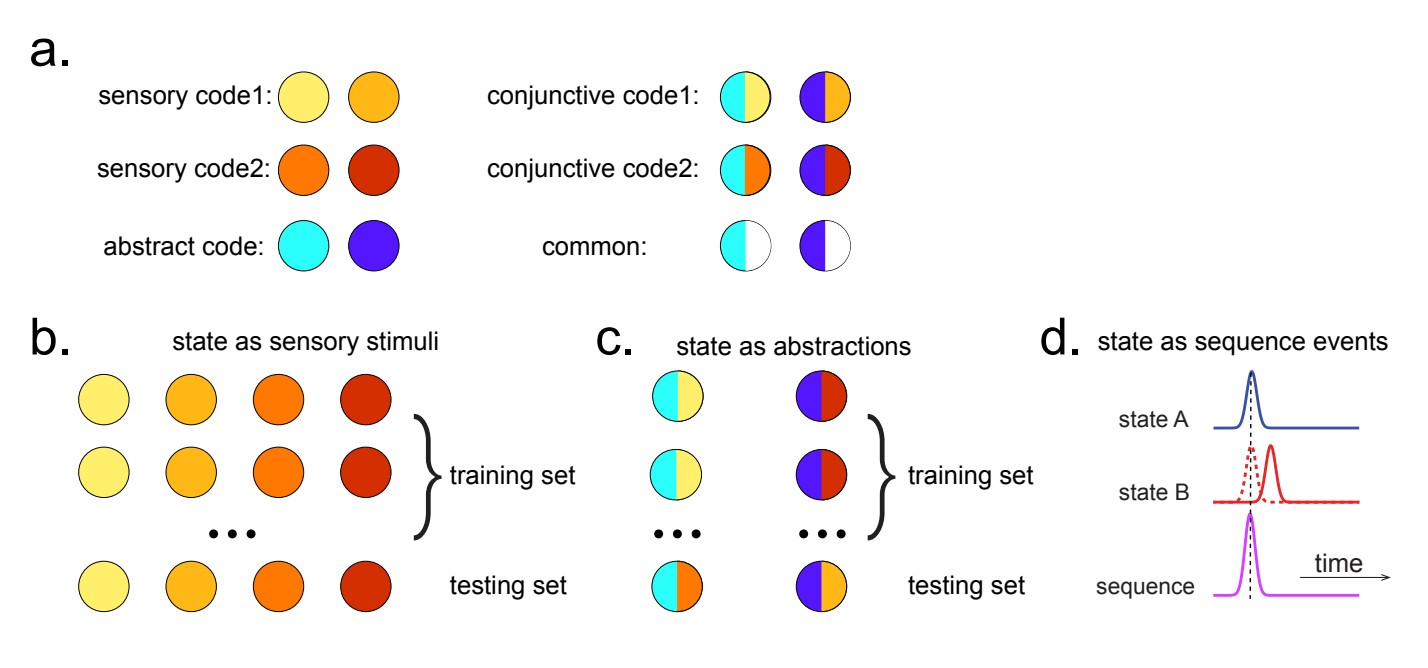

**Figure 2.** Obtaining different state spaces. (a) Assuming we have two abstract codes, each abstract code has two different sensory codes (left panel). The MEG/EEG data corresponding to each stimulus is a conjunctive representation of sensory and abstract codes (right panel). The abstract code can be operationalized as the common information in the conjunctive codes of two stimuli. (b) Training decoding models for stimulus information. The simplest state is defined by sensory stimuli. To determine the best time point for classifier training, we can use a classical leave-one-out cross-validation scheme on the stimuli-evoked neural activity. (c) Training decoding models for abstracted information. The state can also be defined as the abstractions. To extract this information, we need to avoid a confound of sensory information. We can train the classifier on the neural activity evoked by one stimulus and test it on the other sharing the same abstract representation. If neural activity contains both a sensory and abstract code, then the only information that can generalize is the common abstract code. (d) The state can also be defined as the sequence event itself.
The online version of this article includes the following figure supplement(s) for figure 2:

**Figure supplement 1.** Sequences of abstract code.

## States as abstractions

As well as sequences of sensory representations, it is possible to search for replay of more abstract neural representations. Such abstractions might be associated with the presented image (e.g. mammal vs. fish), in which case analysis can proceed as above by swapping categories for images (*Wimmer et al., 2020*). A more subtle example, however, is where the abstraction pertains to the sequence or graph itself. In space, for example, grid cells encode spatial coordinates in a fashion that abstracts over the sensory particularities of any one environment, and therefore can be reused across environments (*Fyhn et al., 2007*). In human studies, similar representations have been observed for the location in a sequence (*Liu et al., 2019*; *Dehaene et al., 2015*). For example, different sequences have shared representations for their second items (*Figure 2*). These representations also replay (*Liu et al., 2019*). However, to measure this replay we need to train decoders for these abstract representations. This poses a conundrum as it is not possible to elicit the abstract representations in the absence of the concrete examples (i.e., the sensory stimuli). Care is required to ensure that the decoders are sensitive to the abstract code rather than the sensory representations (see Appendix 2 for algorithm box of selecting time point for training abstract code). Useful strategies include training classifiers to generalize across stimulus sets and ensuring the classifiers are orthogonal to sensory representations (*Figure 2—figure supplement 1*; details in *Liu et al., 2019*). One way that excludes the possibility of sensory contamination is if the structural representations can be shown to sequence before the subjects have ever seen their sensory correlates (*Liu et al., 2019*).

TDLM can also be used iteratively to ask questions about the ordering of different types of replay events (*Figure 2d*). This can provide for powerful inferences about the temporal organization of

replay, such as the temporal structure between sequences, or the repeating pattern of the same sequence. This more sophisticated use of TDLM merits its own consideration and is discussed in Appendix 3: Sequences of sequences.

## Controlling confounds and maximizing sensitivity in sequence detection

Here, we motivate the key features of TDLM.

### Temporal correlations

In standard linear methods, unmodelled temporal autocorrelation can inflate statistical scores. Techniques such as autoregressive noise modelling are commonplace to mitigate these effects (*Colclough et al., 2015*; *Deodatis and Shinozuka, 1988*). However, autocorrelation is a particular burden for analysis of sequences, where it interacts with correlations between the decoded neural variables.

To see this, consider a situation where we are testing for the sequence $X_i \rightarrow X_j$. TDLM is interested in the correlation between $X_i$ and lagged $X_j$ (see *Equation 1*). But if the $X_i$ and $X_j$ time series contain autocorrelations and are also correlated with one another, then $X_i(t)$ will necessarily be correlated with $X_j(t + \Delta t)$. Hence, the analysis will spuriously report sequences.

Correlations between states are commonplace. Consider representations of visual stimuli decoded from neuroimaging data. If these states are decoded using an *n*-way classifier (forcing exactly one state to be decoded at each moment), then the *n* states will be anti-correlated by construction. On the other hand, if states are each classified against a null state corresponding to the absence of stimuli, then the *n* states will typically be positively correlated with one another.

Notably, in our case, because these autocorrelations are identical between forward and backward sequences, one approach for removing them is to compute the difference measure described above ($D = Z_F - Z_B$). This works well as shown in *Kurth-Nelson et al., 2016*. However, a downside is it prevents us from measuring forward and backward sequences independently. The remainder of this section considers alternative approaches that allow for independent measurement of forward and backward sequences.

### Moving to multiple linear regression

The spurious correlations above are induced because $X_j(t)$ mediates a linear relationship between $X_i(t)$ and $X_j(t + \Delta t)$. Hence, if we knew $X_j(t)$, we can solve the problem by simply controlling for it in a linear regression, as in Granger causality (*Eichler, 2007*):

$$X_j(t + \Delta t) = \beta_0 + X_i(t)\beta_{ij} + X_j(t)\beta_{jj} \tag{9}$$

Unfortunately, we do not have access to the ground truth of $X$ because these variables have been decoded noisily from brain activity. Any error in $X_j(t)$ but not $X_i(t)$ will mean that the control for autocorrelation is imperfect, leading to spurious weight on $\beta_{ij}$, and therefore spurious inference of sequences.

This problem cannot be solved without a perfect estimate of $X$, but it can be systematically reduced until negligible. It turns out that the necessary strategy is simple. We do not know ground truth $X_j(t)$, but what if we knew a subspace that included estimated $X_j(t)$? If we control for that whole subspace, we would be on safe ground. We can get closer and closer to this by including further coregressors that are themselves correlated with estimated $X_j(t)$ with different errors from ground truth $X_j(t)$. The most straightforward approach is to include the other states of $X(t)$, each of which has different errors, leading to the multiple linear regression of *Equation 2*.

*Figure 3a* shows this method applied to the same simulated data whose correlation structure induces false positives in the simple linear regression of *Equation 1*, and by the same logic, so too in cross-correlation. This is why previous studies based on a cross-correlation (*Eldar et al., 2018*; *Kurth-Nelson et al., 2016*) cannot look for sequenceness in forward and backward directions separately, but have to rely on their asymmetry. The multiple regression accounts for the correlation structure of the data and allows correct inference to be made. Unlike the simple subtraction method proposed above (*Figure 3a*, left panel), the multiple regression permits separate inference on forward and backward sequences.

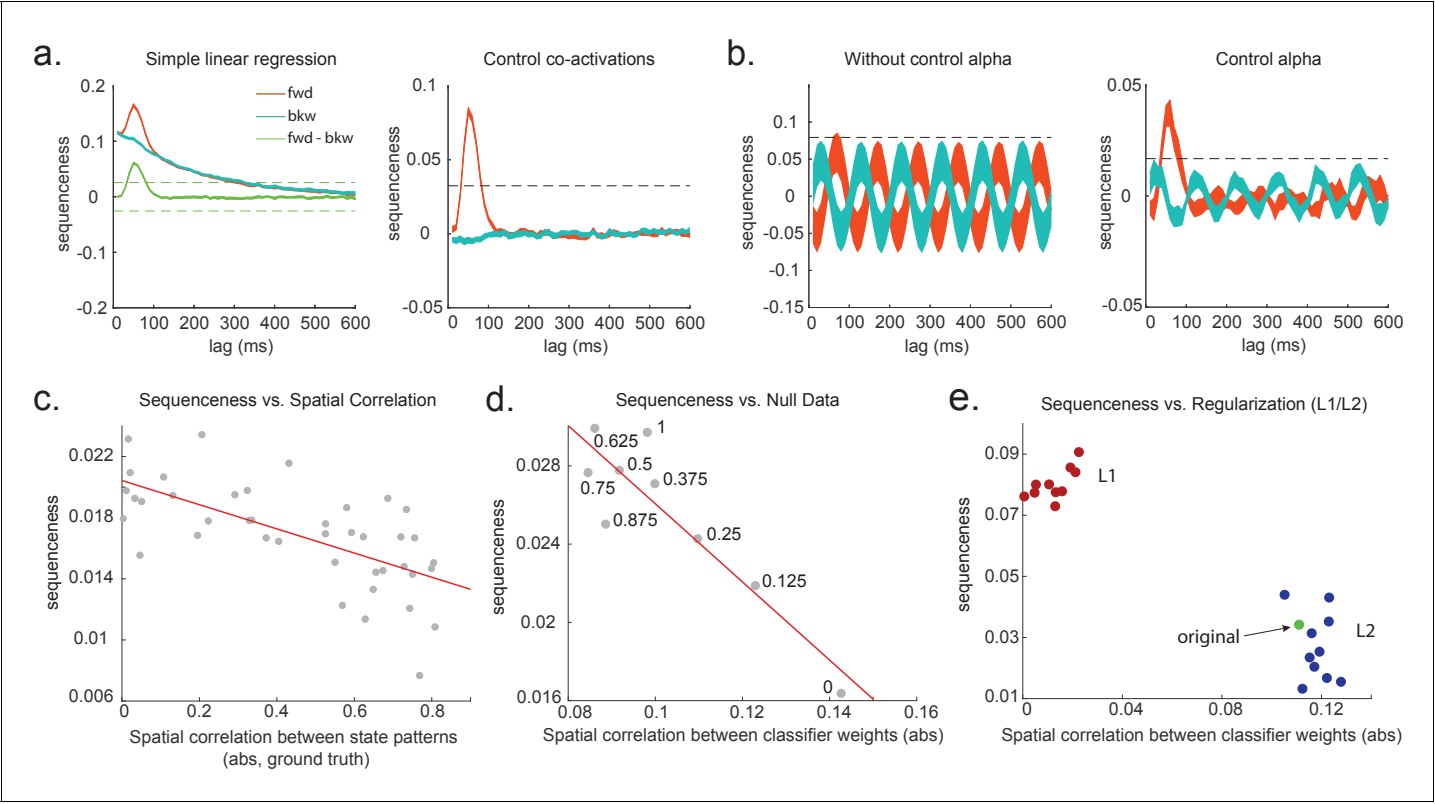

**Figure 3.** Effects of temporal, spatial correlations, and classifier regularization on temporal delayed linear modelling (TDLM). (**a**) Simple linear regression or cross-correlation approach relies on an asymmetry of forward and backward transitions; therefore, subtraction is necessary (left panel). TDLM instead relies on multiple linear regression. TDLM can assess forward and backward transitions separately (right panel). (**b**) Background alpha oscillations, as seen during rest periods, can reduce sensitivity of sequence detection (left panel), and controlling alpha in TDLM helps recover the true signal (right panel). (**c**) The spatial correlation between the sensor weights of decoders for each state reduces the sensitivity of sequence detection. This suggests that reducing overlapping patterns between states is important for sequence analysis. (**d**) Adding null data to the training set increases the sensitivity of sequence detection by reducing the spatial correlations of the trained classifier weights. Here, the number indicates the ratio between null data and task data. '1' means the same amount of null data and the task data. '0' means no null data is added for training. (**e**) L1 regularization helps sequence detection by reducing spatial correlations (all red dots are L1 regularization with a varying parameter value), while L2 regularization does not help sequenceness (all blue dots are L2 regularization with a varying parameter value) as it does not reduce spatial correlations of the trained classifiers compared to the classifier trained without any regularization (green point).

## Oscillations and long timescale autocorrelations

*Equation 2* performs multiple regression, regressing each $X_j(t + \Delta t)$ onto each $X_i(t)$ whilst controlling for all other state estimates at time *t*. This method works well when spurious relationships between $X_i(t)$ and $X_j(t + \Delta t)$ are mediated by the subspace spanned by the other estimated states at time *t* (in particular, $X_j(t)$). One situation in which this assumption might be challenged is when replay is superimposed on a large neural oscillation. For example, during rest (which is often the time of interest in replay analysis), MEG and EEG data often express a large alpha rhythm, at around 10 Hz.

If all states experience the same oscillation at the same phase, the approach correctly controls false positives. The oscillation induces a spurious correlation between $X_i(t)$ and $X_j(t + \Delta t)$, but, as before, this spurious correlation is mediated by $X_j(t)$.

However, this logic fails when states experience oscillations at different phases. This scenario may occur, for example, if we assume there are travelling waves in cortex (*Lubenov and Siapas, 2009*; *Wilson et al., 2001*) because different sensors will experience the wave at different times and different states have different contributions from each sensor. MEG sensors can be seen as measures of local field potential on the scalp, which contain background neural oscillations. In humans, this is dominantly alpha during rest.

In this case, $X_i(t)$ predicts $X_j(t + \Delta t)$ over and above $X_j(t)$. To see this, consider the situation where $\Delta t$ is $\frac{1}{4}\tau$ (where $\tau$ is the oscillatory period) and the phase shift between $X_i(t)$ and $X_j(t)$ is pi/2. Now every peak in $X_j(t + \Delta t)$ corresponds to a peak in $X_i(t)$ but a zero of $X_j(t)$.

To combat this, we can include phase-shifted versions/more time points of $X(t)$. If dominant background oscillation is at alpha frequency (e.g. 10 Hz), neural activity at time $T$ would be correlated with activity at time $T + \tau$. We can control for that by including $X(t + \tau)$, as well as $X(t)$, in the GLM (*Figure 3b*). Here, $\tau$ = 100 ms if assuming the frequency is 10 Hz. Applying this method to the real MEG data during rest, we see much diminished 10 Hz oscillation in sequence detection during rest (*Liu et al., 2019*).

## Spatial correlations

As mentioned above, correlations between decoded variables commonly occur. The simplest type of decoding model is a binary classifier that maps brain activity to one of two states. These states will, by definition, be perfectly anti-correlated. Conversely, if separate classifiers are trained to distinguish each state's representation from baseline ('null') brain data, then the states will often be positively correlated with each other.

Unfortunately, positive or negative correlations between states reduce the sensitivity of sequence detection because it is difficult to distinguish between states within the sequence: collinearity impairs estimation of $\beta$ in *Equation 2*. In *Figure 3c*, we show in simulation that the ability to detect real sequences goes down as the absolute value of a spatial correlation goes up. We took the absolute value here because the direction of correlation is not important, only the magnitude of the correlation matters.

Ideally, the state decoding models should be as independent as possible. We have suggested the approach of training models to discriminate one state against a mixture of other states and null data (*Liu et al., 2019*; *Kurth-Nelson et al., 2016*). This mixture ratio can be adjusted. Adding more null data causes the states to be positively correlated with each other, while less null data leads to negative correlation. We adjust the ratio to bring the correlation between states as close to zero as possible. In *Figure 3d*, we show in simulation the ensuing benefit for sequence detection. An alternative method is penalizing covariance between states in the classifier's cost function (*Weinberger et al., 1988*).

## Regularization

A key parameter in training high-dimensional decoding models is the degree of regularization. In sequence analysis, we are often interested in spontaneous reactivation of state representations, as in replay. However, our decoding models are typically trained on task-evoked data because this is the only time at which we know the ground truth of what is being represented. This poses a challenge insofar as the models best suited for decoding evoked activity at training may not be well suited for decoding spontaneous activity at subsequent tests. Regularizing the classifier (e.g. with an L1 norm) is a common technique for increasing out-of-sample generalization (to avoid overfitting). Here, it has the added potential benefit of reducing spatial correlation between classifier weights.

During classifier training, we can impose L1 or L2 constraints over the inference of classifier coefficients, $W$. This amounts to finding the coefficients, $W$, that maximize the likelihood of the data observations under the constraint imposed by the regularization term. L1 regularization can be phrased as maximizing the likelihood, subject to a regularization penalty on the L1 norm of the coefficient vector:

$$W = \underset{W}{\operatorname{argmax}}[\log(\mathrm{P}(\mathrm{Y}|\mathrm{M}, \mathrm{W})) + \mathrm{b}\lambda_{L1}||\mathrm{W}||_1] \tag{10}$$

L2 regression can be viewed as a problem of maximizing the likelihood, subject to a regularization penalty on the L2 norm of the coefficient vector:

$$W = \underset{W}{\operatorname{argmax}}[\log(\mathrm{P}(\mathrm{Y}|\mathrm{M}, \mathrm{W})) + \mathrm{b}\lambda_{L2}||\mathrm{W}||_2] \tag{11}$$

where $M$ is the task data, with dimension of number of observations, $b$, by number of sensors, $s$. $Y$ is the label of observations, a vector with dimension of $b$ by 1. $\mathrm{P}(\mathrm{Y}|\mathrm{M}, \mathrm{W}) = \sigma(\mathrm{MW})$, and $\sigma$ is the logistic sigmoid function.

We simulate data with varying numbers of true sequences at 40 ms lag and find that the beta estimate of sequence strength at 40 ms positively relates to the number of sequences. We also find that L1 weight regularization is able to detect sequences more robustly than L2 regularization, while L2 performs no better than an unregularized model (*Figure 3e*). The L1 models also have much lower spatial correlation, consistent with L1 achieving better sequence detection by reducing the covariances between classifiers.

In addition to minimizing spatial correlations, as discussed above, it can also be shown that L1-induced sparsity encodes weaker assumptions about background noise distributions into the classifiers as compared to L2 regularization (*Higgins, 2019*). This might be of special interest to researchers who want to measure replay during sleep. Here, the use of sparse classifiers is helpful as background noise distributions are likely to differ more substantially from the (awake state) training data.

## Statistical inference

So far, we have shown how to quantify sequences in representational dynamics. An essential final step is assessing the statistical reliability of these quantities.

All the tests described in this section evaluate the consistency of sequences *across subjects*. This is important because even in the absence of any real sequences of task-related representations spontaneous neural activity is not random but follows repeating dynamical motifs (*Vidaurre et al., 2017*). Solving this problem requires a randomized mapping between the assignment of physical stimuli to task states. This can be done across subjects, permitting valid inference at the group level.

At the group level, the statistical testing problem can be complicated by the fact that sequence measures do not in general follow a known distribution. Additionally, if a state-to-state lag of interest ($\Delta t$) is not known a priori, it is then necessary to perform tests at multiple lags, creating a multiple comparisons problem over a set of tests with complex interdependencies. In this section, we discuss inference with these issues in mind.

## Distribution of sequenceness at a single lag

If a state-to-state lag of interest ($\Delta t$) is known a priori, then the simplest approach is to compare the sequenceness against zero, for example, using either a signed-rank test or one-sample *t* test (assuming Gaussian distribution). Such testing assumes the data are centred on zero if there were no real sequences. We show this approach is safe in both simulation (assuming no real sequences) and real MEG data where we know there are no sequences.

In simulation, we assume no real sequences, but state time courses are autocorrelated. At this point, there is no systematic structure in the correlation between the neuronal representations of different states (see later for this consideration). We then simply select the 40 ms time lag and compare its sequenceness to zero using either a signed-rank test or one-sample *t* test. We compare false-positive rates predicted by the statistical tests with false-positive rates measured in simulation (*Figure 4a*). We see the empirical false positives are well predicted by theory.

We have tested this also on real MEG data. In *Liu et al., 2019*, we had one condition where we measured resting activity before the subjects saw any stimuli. Therefore, by definition these sensory stimuli could not be replayed, we can use classifiers from these stimuli (measured later) to test a false-positive performance of statistical tests on replay. Note, in our case, that each subject saw the same stimuli in a different order. They could not know the correct stimulus order when these resting data were acquired. These data provide a valid null for testing false positives.

To obtain many examples, we randomly permute the eight different stimuli 10,000 times and then compare sequenceness (at 40 ms time lag) to zero using either a signed-rank test or one-sample *t* test across subjects. Again, predicted and measured false-positive rates match well (*Figure 4b*, left panel). This holds true across all computed time lags (*Figure 4b*, right panel).

An alternative to making assumptions about the form of the null distribution is to compute an empirical null distribution by permutation. Given that we are interested in the sequence of states over time, one could imagine permuting either state identity or time. However, permuting time uniformly will typically lead to a very high incidence of false positives as time is not exchangeable under the null hypothesis (*Figure 4c*, blue colour). Permuting time destroys the temporal smoothness of neural data, creating an artificially narrow null distribution (*Liu et al., 2019*; *Kurth-Nelson et al.,*

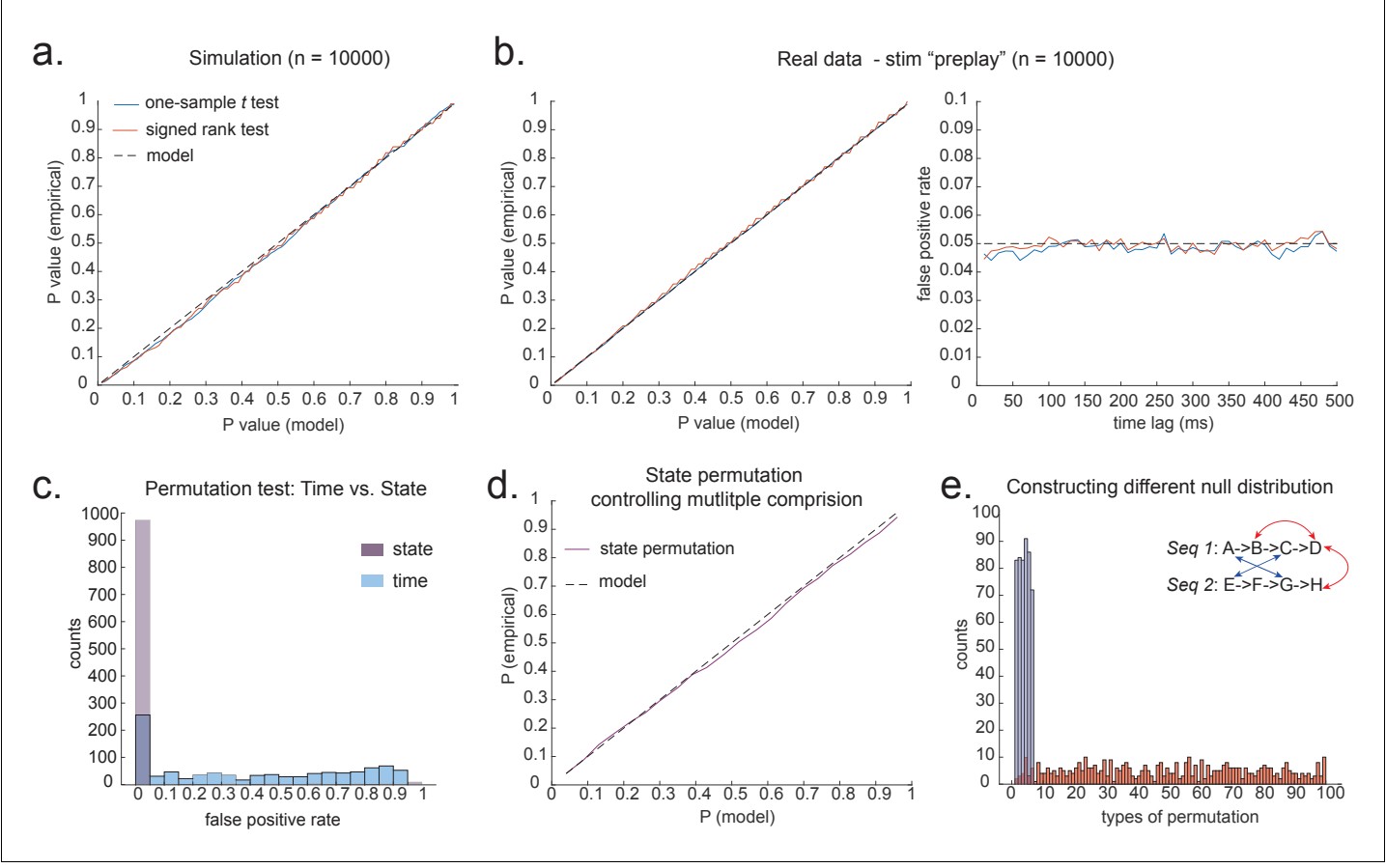

**Figure 4.** Statistical inference. (**a**) P-P plot of one-sample *t* test (blue) and Wilcoxon signed-rank test (red) against zero. This is performed in simulated MEG data, assuming autocorrelated state time courses, but no real sequences. In each simulation, the statistics are done only on sequenceness at 40 ms time lag, across 24 simulated subjects. There are 10,000 simulations. (**b**) We have also tested the sequenceness distribution on real MEG data. Illustrated is the pre-task resting state on 22 subjects from Liu et al., where the ground truth is the absence of sequences given the stimuli have not yet been shown. The statistics are done on sequenceness at 40 ms time lag, across the 22 subjects. There are eight states. The state identity is randomly shuffled 10,000 times to construct a null distribution. (**c**) Time-based permutation test tends to result in high false positive, while state identity-based permutation does not. This is done in simulation assuming no real sequences (n = 1000). (**d**) P-P plot of state identity-based permutation test over peak sequenceness. To control for multiple comparisons, the null distribution is formed by taking the maximal absolute value over all computed time lags within a permutation, and the permutation threshold is defined as the 95% percentile over permutations. In simulation, we only compared the max sequence strength in the data to this permutation threshold. There are 10,000 simulations. In each simulation, there are 24 simulated subjects, with no real sequence. (**e**) In state identity-based permutation, we can test more specific hypotheses by controlling the null distribution. Blue are the permutations that only exchange state identity across sequences. Red are the permutations that permit all possible state identity permutations. 500 random state permutations are chosen from all possible ones. The X axis is the different combinations of the state permutation. It is sorted so that the cross-sequence permutations are in the beginning.

*2016*). This false positive also exists if we circular shift the time dimension of each state. This is because the signal is highly non-stationary. Replays come in bursts, as recently analysed (*Higgins et al., 2021*), and this will break a circular shift (*Harris, 2020*). State permutation, on the other hand, only assumes that state identities are exchangeable under the null hypothesis, while preserving the temporal dynamics of the neural data represents a safer statistical test that is well within 5% false-positive rate (*Figure 4c*, purple colour).

## Correcting for multiple comparisons

If the state-to-state lag of interest is not known, we have to search over a range of time lags. As a result, we then have a multiple comparison problem. Unfortunately, we do not as yet have a good parametric method to control for multiple testing over a distribution. It is possible that one could use methods that exploit the properties of Gaussian random fields, as is common in fMRI

(*Worsley et al., 1996*), but we have not evaluated this approach. Alternatively, we could use Bonferroni correction, but the assumption that each computed time lag is independent is likely false and overly conservative.

We recommend relying on state identity-based permutation. To control for the family-wise error rate (assuming $\alpha = 0.05$), we want to ensure there is a 5% probability of getting the tested sequenceness strength ($S_{test}$) or bigger by chance in \*any\* of the multiple tests. We therefore need to know what fraction of the permutations gives $S_{test}$ or bigger in any of *their* multiple tests. If any of the sequenceness scores in each permutation exceed $S_{test}$, then the *maximum* sequenceness score in the permutation will exceed $S_{test}$, so it is sufficient to test against the maximum sequenceness score in the permutation. The null distribution is therefore formed by first taking the peak of sequenceness across all computed time lags of each permutation. This is the same approach as used for family-wise error correction for permutations tests in fMRI data (*Nichols, 2012*), and in our case it is shown to behave well statistically (*Figure 4d*).

## What to permute

We can choose which permutations to include in the null distribution. For example, consider a task with two sequences, $Seq1: A \rightarrow B \rightarrow C \rightarrow D$ and $Seq2: E \rightarrow F \rightarrow G \rightarrow H$. We can form the null distribution either by permuting all states (e.g. one permutation might be $E \rightarrow F \rightarrow A \rightarrow B$, $H \rightarrow C \rightarrow E \rightarrow D$), as implemented in *Kurth-Nelson et al., 2016*. Alternatively, we can form a null distribution which only includes transitions between states in different sequences (e.g. one permutation might be $D \rightarrow G \rightarrow A \rightarrow E$, $H \rightarrow C \rightarrow F \rightarrow B$), as implemented in *Liu et al., 2019*. In each case, permutations are equivalent to the test data under the assumption that states are exchangeable between positions and sequences. The first case has the advantage of many more possible permutations, and therefore may make more precise inferential statements in the tail. The second case may be more sensitive in the presence of a signal as the null distribution is guaranteed not to include permutations which share any transitions with the test data (*Figure 4e*). For example, in *Figure 4e*, the blue swaps are the permutations that only exchange state identity across sequences, as in *Liu et al., 2019*, while the red swaps are the permutations that permit all possible state identity permutations, as in *Kurth-Nelson et al., 2016*. Note that there are many more different state permutations in red swaps than in blue swaps. We can make different levels of inferences by controlling the range of the null distributions in the state permutation tests.

## Cautionary note on exchangeability of states after training

Until now, all non-parametric tests have assumed that state identity is exchangeable under the null hypothesis. Under this assumption, it is safe to perform state identity-based permutation tests on $Z_F$ and $Z_B$. In this section, we consider a situation where this assumption is broken.

More specifically, take a situation where the neural representation of states $A$ and $B$ is related in a systematic way or, in other words, the classifier on state $A$ is confused with state $B$, and we are testing sequenceness of $A \rightarrow B$. Crucially, to break the exchangeability assumption, representations of $A$ and $B$ have to be systematically more related than other states, for example, $A$ and $D$. This cannot be caused by low-level factors (e.g. visual similarity) because states are counterbalanced across subjects, so any such bias would cancel at the population level. However, such a bias might be *induced* by task training.

In this situation, it is, in principle, possible to detect sequenceness of $A \rightarrow B$ even in the absence of real sequences. In the autocorrelation section above, we introduced protections against the interaction of state correlation with autocorrelation. These protections may fail in the current case as we cannot use other states as controls (as we do in the multiple linear regression) because $A$ has systematic relationship with $B$, but not other states. State permutation will not protect us from this problem because state identity is no longer exchangeable.

Is this a substantive problem? After extensive training, behavioural pairing of stimuli can indeed result in increased neuronal similarity (*Messinger et al., 2001*; *Sakai and Miyashita, 1991*). These early papers involved long training in monkeys. More recent studies have shown induced representational overlap in human imaging within a single day (*Kurth-Nelson et al., 2015*; *Barron et al., 2013*; *Wimmer and Shohamy, 2012*). However, when analysed across the whole brain, such

representational changes tend to be localized to discrete brain regions (*Schapiro et al., 2013*; *Garvert et al., 2017*), and as a consequence may have limited impact on whole-brain decodeability.

Whilst we have not yet found a simulation regime in which false positives are found (as opposed to false negatives), there exists a danger in cases where, by experimental design, the states are not exchangeable.

## Source localization

Uncovering temporal structure of neural representation is important, but it is also of interest to ask where in the brain a sequence is generated. Rodent electrophysiology research focuses mainly on the hippocampus when searching for replay. One advantage of whole-brain non-invasive neuroimaging over electrophysiology (despite many known disadvantages, including poor anatomical precision, low signal-noise ratio) is in its ability to examine neural activity in multiple other brain regions. Ideally, we would like a method that is capable of localizing sequences of more abstract representation in brain regions beyond hippocampus (*Liu et al., 2019*).

We want to identify the *time* when a given sequence is very likely to unfold, so we can construct averages of independent data over these times. We achieve this by transforming from the space of original states, $X_{orig}$, to the space of sequence events, $X_{seq}$. First, based on the transition of interest, $T$, we can obtain the projection matrix, $X_{proj}$:

$$X_{proj} = X_{orig} \times T \tag{12}$$

If we know the state lag within sequence, $\Delta t$ (e.g. the time lag give rise to the strongest sequenceness), or have it a priori, we can obtain the time-lagged matrix, $X_{lag}$:

$$X_{lag} = X_{orig}(t - \Delta t) \tag{13}$$

Then, we obtain state space with sequence event as states by element-wise multiply $X_{proj}$ and $X_{lag}$:

$$X_{seq} = X_{lag} . * X_{proj} \tag{14}$$

Each element in $X_{seq}$ indicates the strength of a (pairwise) sequence at a given moment in time. At this stage, $X_{seq}$ is a matrix with number of time points as rows (same as $X_{orig}$), and with number of pairwise sequences (e.g. A->B; B->C; etc.) as columns. Now on this matrix, $X_{seq}$, we can either look for sequences of sequences (see Appendix 3), or sum over columns (i.e. average over pairwise sequence events), and obtain a score at each time point reflecting how likely it is to be a sequence member (*Figure 5a*).

We can use this score to construct averages of other variables that might co-vary with replay. For example, if we choose time points when this score is high (e.g. 95th percentile) after being low for the previous 100 ms and construct an average time-frequency plot of the raw MEG data aligned to these times, we can reconstruct a time-frequency plot that is, *on average,* associated with replay onset (*Figure 5b*). Note that although this method assigns a score for individual replay events as an intermediary variable, it results in an *average* measure across many events.

This approach is similar to spike-triggered averaging (*Sirota et al., 2008*; *Buzsáki et al., 1983*). Applying this to real MEG data during rest, we can detect increased hippocampal power at 120–150 Hz, at replay onset (*Figure 5b, c*). Source reconstruction in the current analysis was performed using linearly constrained minimum variance (LCMV) beamforming, a common method for MEG source localization. This is known to suffer from distal correlated sources (*Hincapié et al., 2017*). A better method may be Empirical Bayesian Beamfomer for accommodating correlated neural source as a priori (*O'Neill, 2021*).

## TDLM for rodent replay

So far, we have introduced TDLM in the context of analysing human MEG data. Relatedly, its application on human EEG data was also explored (Appendix 4: Apply TDLM to human whole-brain EEG data). Historically, replay-like phenomena have been predominantly studied in rodents with electrophysiology recordings in the hippocampal formation (*Davidson et al., 2009*; *Grosmark and Buzsáki, 2016*; *Tingley and Peyrache, 2020*). This raises interesting questions: how does TDLM

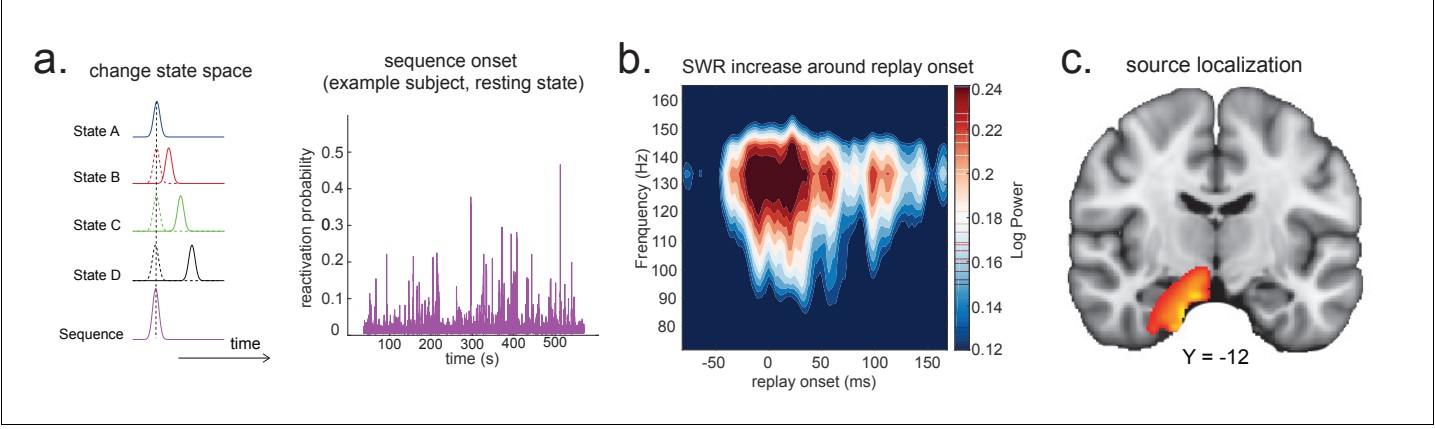

**Figure 5.** Source localization of replay onset. (**a**) Temporal delayed linear modelling indexes the onset of a sequence based on the identified optimal state-to-state time lag (left panel). Sequence onset during resting state from one example subject is shown (right panel). (**b**) There was a significant power increase (averaged across all sensors) in the ripple frequency band (120–150 Hz) at the onset of replay compared to the pre-replay baseline (100 to 50 ms before replay). (**c**) Source localization of ripple-band power at replay onset revealed significant hippocampal activation (peak Montreal Neurological Institute, i.e., MNI coordinate: X = 18, Y = −12, Z = −27). Panels (**b**) and (**c**) are reproduced from Figure 7A, C, *Liu et al., 2019*, *Cell*, published under the Creative Commons Attribution 4.0 International Public License (CC BY 4.0).

compare to the existing rodent replay methods, can TDLM be applied to spiking data for detecting rodent replays, and what are the pros and cons? In this section, we address these questions.

## Generality of graph- vs. line-based replay methods

Given that TDLM works on the decoded state space, rather than sensor (with analogy to cell) level, we compared TDLM to rodent methods that work on the posterior decoded position (i.e., state) space, normally referred to as Bayesian-based methods (*Tingley and Peyrache, 2020*). (Note that these methods are typically Bayesian in how position is decoded from spikes [*Zhang et al., 1998*] but not in how replay is measured from decoded position.) Two commonly used methods are Radon transform (*Davidson et al., 2009*) and linear weighted correlation (*Grosmark and Buzsáki, 2016*).

Both methods proceed by forming a 2D matrix, where one dimension is the decoded state (e.g. positions on a linear track), and the other dimension is time (note that the decoded state is embedded in 1D). The methods then try to discover if an ordered line is a good description of the relationship between state and (parametric) time. For this reason, we call this family of approaches 'line search'.

The radon method uses a discrete Radon transform to find the best line in the 2D matrix (*Toft, 1996*) and then evaluates the radon integral, which will be high if the data lie on a line (*Figure 6a*). It compares this to permutations of the same data where the states are reordered (*Tingley and Peyrache, 2020*). The linear weighted correlation method computes the average correlation between the time and estimated position in the 1D embedding (*Figure 6b*). The correlation is non-zero provided there is an orderly reactivation along the state dimension.

Both methods are applied to decoded positions, where they are sorted based on the order in a linearized state space. TDLM also works on the decoded position space, but instead of directly measuring the relationship between position and time, it measures the transition strength for each possible state to state transitions (*Figure 6c*).

This is a key difference between TDLM and these popular existing techniques. To reiterate, the latter rely on a continuous parametric embedding of behavioural states and time. TDLM is fundamentally different as it works on a graph and examines the statistical likelihood of some transitions happening more than others. This is therefore a more general approach that can be used for sequences drawn from any graph (e.g. 2D maze, *Figure 6d*), not just graphs with simple embeddings (like a linear track). For example, in a non-spatial decision-making task (*Kurth-Nelson et al., 2016*), all states lead to two different states and themselves can be arrived at from two other different states (*Figure 6e*). Existing 'line search' methods will not work because there is no linear relationship between time and states (*Figure 6f*).

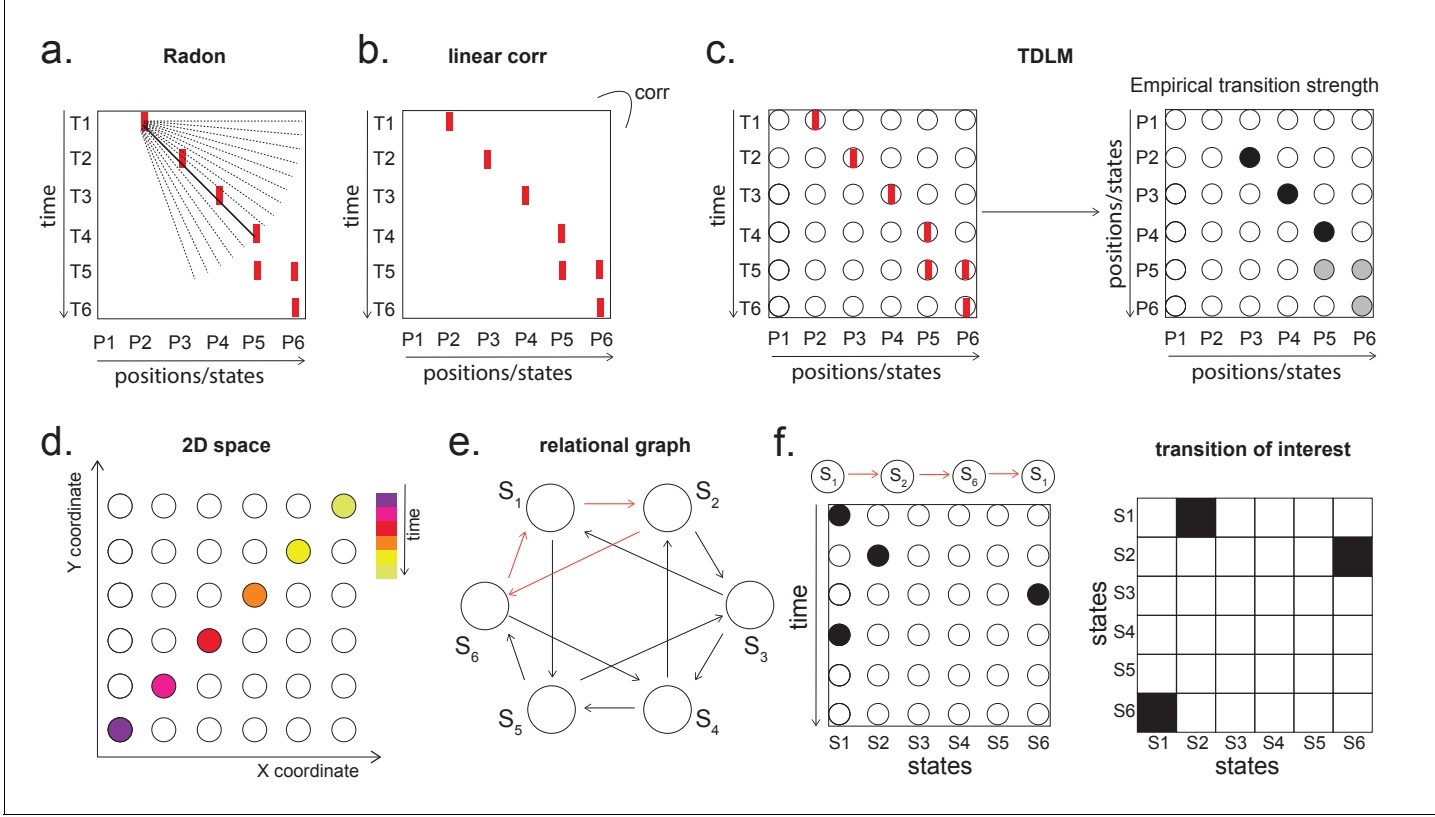

**Figure 6.** Temporal delayed linear modelling (TDLM) vs. existing rodent replay methods. (**a**) The Radon method tries to find the best fitting line (solid line) of the decoded positions as a function of time. The red bars indicate strong reactivation at a given location. (**b**) The linear correlation method looks for correlations between time and decoded position. (**c**) The TDLM method, on the other hand, does not directly measure the relationship between state and time, but quantifies the likelihood of each transition. In the right panel, likelihood is indicated by darkness of shading. For example, P5 can be followed by either P5 or P6, making each transition half as likely as the deterministic P4->P5 transition. Later this empirical transition matrix is compared to a theoretical one, quantifying the extent to which the empirical transitions fit with a hypothesis. (**d**) Sequences in 2D space are in three dimensions, which is hard to translate into a line search problem, i.e., time*position spaces. (**e**) This is the transition matrix used in *Kurth-Nelson et al., 2016*, which cannot be translated into a linear state space. The transitions in red are an example of a trajectory. (**f**) Putting the example trajectory into the time by state matrix, we can see that there is no linear relationship between them (left panel). In TDLM, this is tested by forming a hypothesis regressor in the state-to-state transition matrix (right panel).

## Multi-scale TDLM

While continuous spaces can be analysed in TDLM by simply chunking the space into discrete states, TDLM in its original form may potentially be less sensitive for such analyses than techniques with built-in assumptions about the spatial layout of the state space, such as the linear relationship between time and reactivated states (Appendix 5 'Less sensitivity of TDLM to skipping sequences'). In essence, because TDLM works on a graph, it has no information about the Euclidean nature of the state space, while techniques that make assumptions about the linear relationship between space and time benefit from these assumptions. For example, detecting state 1 then state 5 then state 10 counts as replay in these techniques, but not in TDLM.

However, TDLM can be extended to address this problem. For continuous state spaces, we first need to decide how to best discretize the space. If we choose a large scale, we will miss replays that occur predominantly within a spatial bin. If we choose a small scale, we will miss transitions that jump spatial bins. A simple solution is to apply TDLM at multiple different scales and take an (variance-weighted) average of the sequenceness measures across different scales. For example, when measuring replay at the same speed, we can average events that travel 5 cm in 10 ms together with events that travel 10 cm in 20 ms.

Specifically, to perform multi-scale TDLM, we discretize position bins at multiple widths. This generates rate maps at multiple scales (e.g. 5 cm, 10 cm, 20 cm, 40 cm), and hence a multi-scale state

space. For each replay speed of interest, we apply TDLM separately at each scale, and then take a variance-weighted average of replay estimates over all scales.

$$\beta_M = \frac{\sum\limits_{i=1}^{n} \beta_i / V_i}{\sum\limits_{i=1}^{n} 1 / V_i} \tag{15}$$

where $\beta_i$ is the sequence strength of given speed (i.e. state-to-state lag) measured at scale $i$, $V_i$ is the variance of its $\beta_i$ estimator, and $n$ is the number of scales. In the end, statistical testing is performed on the precision weighted averaged sequence strength, $\beta_M$, in the same way as we do in the original TDLM.

It is easy to see why this addresses the potential concerns raised above as some scales will capture the 1 -> 2 -> 3 transitions, whilst others will capture the 1 -> 10 -> 20 transitions: because the underlying space is continuous, we can average results of the same replay speed together, and this will reinstate the Euclidean assumptions.

## Applying multi-scale TDLM to real rodent data (place cells in CA1)

We demonstrate the applicability of multi-scale TDLM by analysing CA1 place cell spiking data from *Ólafsdóttir et al., 2016*. In *Ólafsdóttir et al., 2016*, rats ran multiple laps on a 600 cm Z maze and were then placed in a rest enclosure for 1.5 hr (*Figure 7a*). The Z maze consists of three tracks, with its ends and corners baited with sweetened rice to encourage running from one end to the other. The animal's running trajectory was linearized, dwell time and spikes were binned into 2 cm bins and smoothed with a Gaussian kernel (σ = 5 bins). We generated rate maps separately for inbound (track 1 -> track 2 -> track 3) and outbound (track 3 -> track 2 -> track 1) running (see details in section 'Rodent replay dataset').

As in *Ólafsdóttir et al., 2016*, cells recorded in CA1 were classified as place cells if their peak firing field during track running was above 1 Hz with a width of at least 20 cm (see an example in *Figure 7b*). The candidate replay events were identified based on multi-unit (MU) activity from place cells during rest time. Periods exceeding the mean rate by three standard deviations of MU activity were identified as possible replay events. Events less than 40 ms long, or which included activity from less than 15% of the recorded place cell ensemble, were rejected (see an example of putative replay event in *Figure 7c*), and the remaining events were labelled putative replay events.

We analysed data from one full recording session (track running for generating rate map, post-running resting for replay detection) from Rat 2192 reported in *Ólafsdóttir et al., 2016*. Following the procedure described above, we identified 58 place cells and 1183 putative replay events. Replay analysis was then performed on the putative replay events, separately for inbound and outbound rate maps given the same position has a different decoded state depending on whether it was during an outbound or inbound run.

A forward sequence is characterized by states from the outbound map occurring in the outbound order or states from the inbound map occurring in the inbound order. Conversely, a backward sequence is when states from the inbound map occur in the outbound order or states from the outbound map occur in the inbound order. Candidate events were decoded based on a rate map, transforming the ncells * ntime to nstates * ntime. Each entry in this state space represents the posterior probability of being in this position at a given time. Replay analysis was performed solely on this decoded state space.

Note that TDLM is applied directly to the concatenated rather than individual replay events. This is because TDLM is a linear modelling framework. Applying TDLM on each single replay event and then averaging the beta estimates (appropriately weighted by the variances) is equivalent to running TDLM once on the concatenated replay events. It quantifies the average amount of replay across many events, which is different compared to existing replay methods that focus on single replay events. Because TDLM addresses statistical questions in linear modelling, it does not require secondary statistics to ask whether the 'counts' of individual events are more likely than chance or more likely in one situation than another.

During the whole sleep period, TDLM identified a significant forward sequence for the outbound map with a wide speed range around from 1 to 10 m/s (*Figure 7d*, left panel), consistent with recent

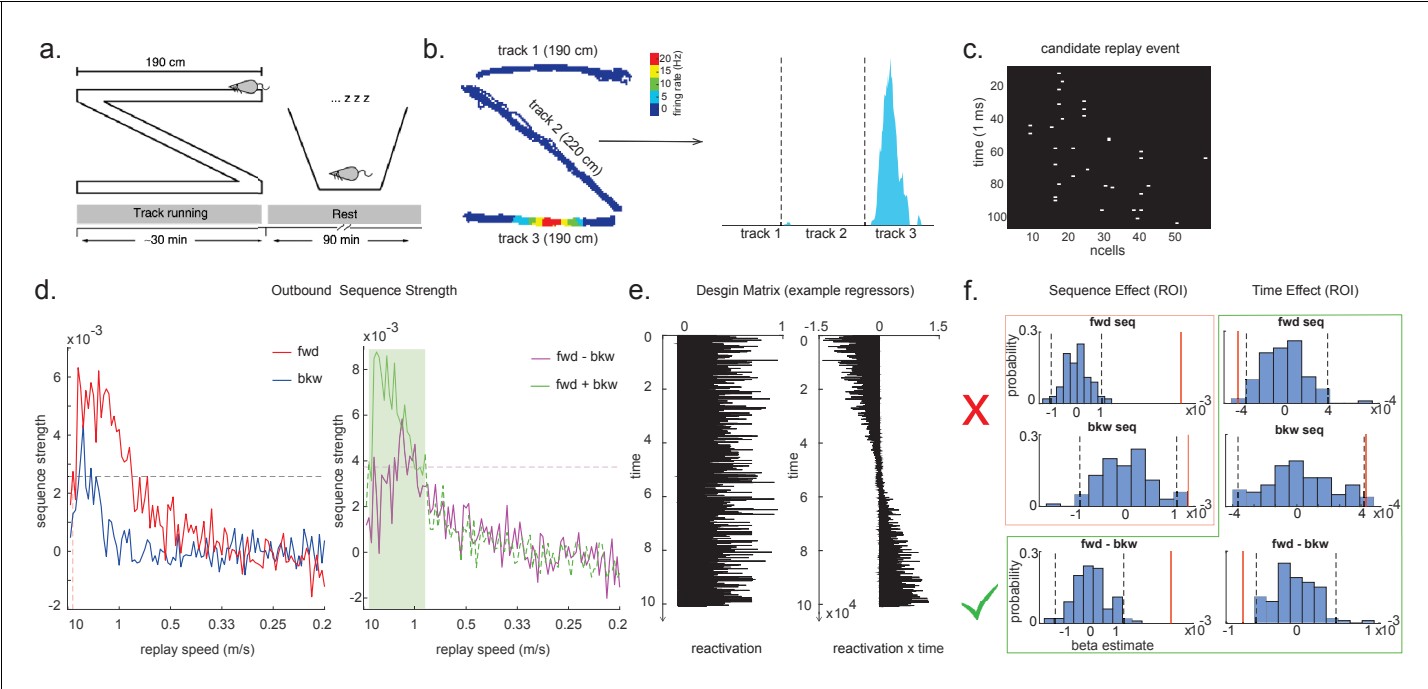

**Figure 7.** Temporal delayed linear modelling (TDLM) applied to real rodent data. (**a**) The experimental design of *Ólafsdóttir et al., 2016*. Rats ran on Z maze for 30 min, followed by 90 min rest. (**b**) An example rate map for a place cell. The left panel shows its spatial distribution on the Z maze, and the right panel is its linearized distribution. (**c**) An example of a candidate replay event (spiking data). (**d**) Sequence strength as a function of replay speed is shown for the outbound rate map. Black dotted line is the permutation threshold after controlling for multiple comparisons. Left panel: forward sequence (red) and backward sequence (blue). The red dotted line indicates the fastest replay speed that is significant – 10 m/s. Right panel: forward–backward sequence. The pink dotted line indicates the multiple comparison-corrected permutation threshold for the replay difference. The green line is the sum of sequence strength between forward and backward direction. The solid line (with green shading) indicates the significant replay speeds (0.88–10 m/s) after controlling for multiple comparisons. We use this as a region of interest (ROI) to test for time-varying effect on replay in (**f**). (**e**) Illustration of two exemplar regressors in the design matrix for assessing time effect on replay strength. The 'reactivation' regressor is a lagged copy of reactivation strength of given position and is used to obtain sequence effect. The 'reactivation × time' regressor is the element-wise multiplication between this position reactivation and time (z-scored); it explicitly models the effect of time on sequence strength. Both regressors are demeaned. (**f**) Beta estimate of the sequence effect (left panel), as well as time modulation effect on sequence (right panel) in the ROI, is shown. Negative value indicates replay strength decreases over time, while positive value means replay increases as a function of sleep time. The statistical inference is done based on a permutation test. The two black dotted lines in each panel indicate the 2.5th and 97.5th percentile of the permutation samples, respectively. The red solid line indicates the true beta estimate of the effect. Note that there is a selection bias in performing statistical inference on forward and backward sequence strength (red rectangle) within this ROI, given the sum of forward and backward sequence is correlated with either forward or backward sequence alone. There is no selection bias in performing statistics on the difference of sequence effects or effects relating to time (green rectangle).

findings from *Denovellis, 2020* on varying replay speed (similar results were obtained for inbound map, not shown here for simplicity). In our analysis, the fastest speed is up to 10 m/s, which is around 20× faster than its free running speed, representing approximately half a track-arm in a typical replay event, consistent with previous work (*Lee and Wilson, 2002*; *Davidson et al., 2009*; *Karlsson and Frank, 2009*; *Nádasdy et al., 1999*).

## Second-order inferences

As pointed out by *van der Meer et al., 2020*, there are two types of statistical questions: a 'first-order' sequence question, which concerns whether an observed sequenceness is different from random (i.e. do replays exist?); and a 'second-order' question, which requires a comparison of sequenceness across conditions (i.e. do replays differ?). Because it is embedded in a linear regression framework, TDLM is ideally placed to address such questions. There are two ways of asking such questions in linear modelling: contrasts and interactions. We explain them with examples here.

## Linear contrasts

After fitting a regression model, resulting in coefficients for different regressors, we can test hypotheses about these coefficients by constructing linear combinations of the coefficients that would be zero under the null hypothesis. For example, if we want to test whether effect A is greater than effect B, then we can compute the linear contrast A – B (which would be zero under the null hypothesis) and perform statistics on this new measure. If we want to test whether replay increases linearly over five conditions [A, B, C, D, E], we can compute the linear contrast −2*A − B + 0*C + D + 2*E (which would be zero under the null hypothesis) and perform statistics on this new measure. Statistics (within or across animals) can operate with these contrasts in exactly the same way as with the original coefficients from the linear model. Here, we demonstrate this by showing in our example dataset that there was a greater preponderance for forward than backward replay. We construct the contrast (forwards – backwards) and test it against zero using a multiple-comparison-controlled permutation test (*Figure 7d*, right panel, pink line). By constructing a different contrast (forwards + backwards), we can also show that the total replay strength across both types of replays was significant (*Figure 7d*, right panel, green line).

## Interactions

A second method for performing second-order tests is to introduce them into the linear regression as interaction terms, and then perform inference on the regression weights for these interactions. This means changing Equation 2 to include new regressors. For example, if interested in how reactivations change over time, one could build new regressors ($Xtime_k(t)$), obtained by element-wise multiplying the state regressor, e.g. $X_k(t)$ with time indices ($Xtime_k(t) = X_k(t).*\text{time}$). Now the first-level GLM is constructed as (omitting residual term ε, same as *Equation 2*):

$$X_j(t+\Delta t) = \sum_{k=1}^{n} X_k(t)\beta_{kj} + Xtime_k(t)\beta t_{kj} \tag{16}$$

Example regressors in the design matrix can be seen in *Figure 7e*. The first regressor, $X_k(t)$, is one of the state reactivation regressors used in standard TDLM. The second regressor, $Xtime_k(t)$, is the same as $X_k(t)$ multiplied by time. (There are k regressors of each form in regressor matrix.) Here, we chose to demean the time regressor before the interaction, so the early half of the regressor is negative and the late half is positive. This has no effect on the regression coefficients of the interaction term, but, by rendering the interaction approximately orthogonal to $X_k(t)$, it makes it possible to estimate the main effect and the interaction in the same regression.

Note that the interaction regressor is orthogonal to the state reactivation regressor, so it will have no effect on the first-order regression terms. If we include such regressors for all states, then we can get two measures for each replay direction (sequence effect and time effect). The first tells us the average amount of replay throughout the sleep period (first order). The second tells us whether replay increases or decreases as time progresses through the sleep period (second order).

## Orthogonal tests in regions of interest

When examining forward–backward replay above, we did separate inference for each replay speed, and then performed multiple comparison testing using the max-permutation method (see section 'Statistical inference'). We now take the opportunity to introduce another method common in human literature.

To avoid such multiple comparison correction, it is possible to select a 'region of interest' (ROI), average the measure in question over that ROI, and perform inference on this average measure. Because we are now only testing one measure, there is no multiple comparison problem. Critical in this endeavour, however, is that we do not use the measure under test or anything that correlates with that measure as a means to define the ROI. This will induce a selection bias (*Kriegeskorte et al., 2009*). In the example in *Figure 7f*, we have used the average replay (forwards + backwards) to select the ROI. We are interested in speeds in which there is detectable replay on average across both directions and the whole sleep period (*Figure 7d*, right panel, green shaded area). If we select our ROI in this way, we cannot perform unbiased inference on first-order forward or backward replay because forward and backward regressors correlate with their sum (*Figure 7f*, statistical inference in the red rectangle is biased). However, we can perform unbiased

inference on several second-order effects (*Figure 7f*, statistical inference in the green rectangle). We can test (forwards – backwards) assuming the difference of terms is orthogonal to their sum (as it is in this case). Further, we can test any interaction with time because the ROI is defined on the average over time and the interaction looks for *differences* as a function of time. When we perform these tests in our example dataset (*Figure 7f*, green rectangle), we confirm that there are more forward than backward replay on average. We further show that forward replay is decreasing with time during sleep, and that backward replay is increasing with time. Their difference (forwards – backwards) is also significant.

In addition to the time-varying effect, we can also test the spatial modulation effect, that is, how replay strength (at the same replay speed) changes as a function of its spatial content. For example, is replay stronger for transitions in the start of track compared to the end of the track? As an illustrative example, we have used the same ROI defined above and test the spatial modulation effect on forward replay. Note that this test of spatial modulation effect is also unbiased from the overall strength of forward replay, and thereby no selection bias in this ROI, as well.

For visualization purposes, we have first plotted the estimated strength for each pairwise forward sequence (*Figure 8a*), separately within each scale (from 1 to 4, with increasing spatial scales). The pairwise sequences are ordered from the start of the track to the end of the track. Alongside the pairwise sequence plot, we have plotted the mean replay strength over all possible pairwise transitions (in red) in comparison to the mean of all control transitions (in grey; as expected, they are all around 0). Note that we cannot perform inference on the difference between the red and grey bars here because they have been selected from a biased ROI. It is simply for illustration purposes. We have therefore put them in red squares to match *Figure 7f*.

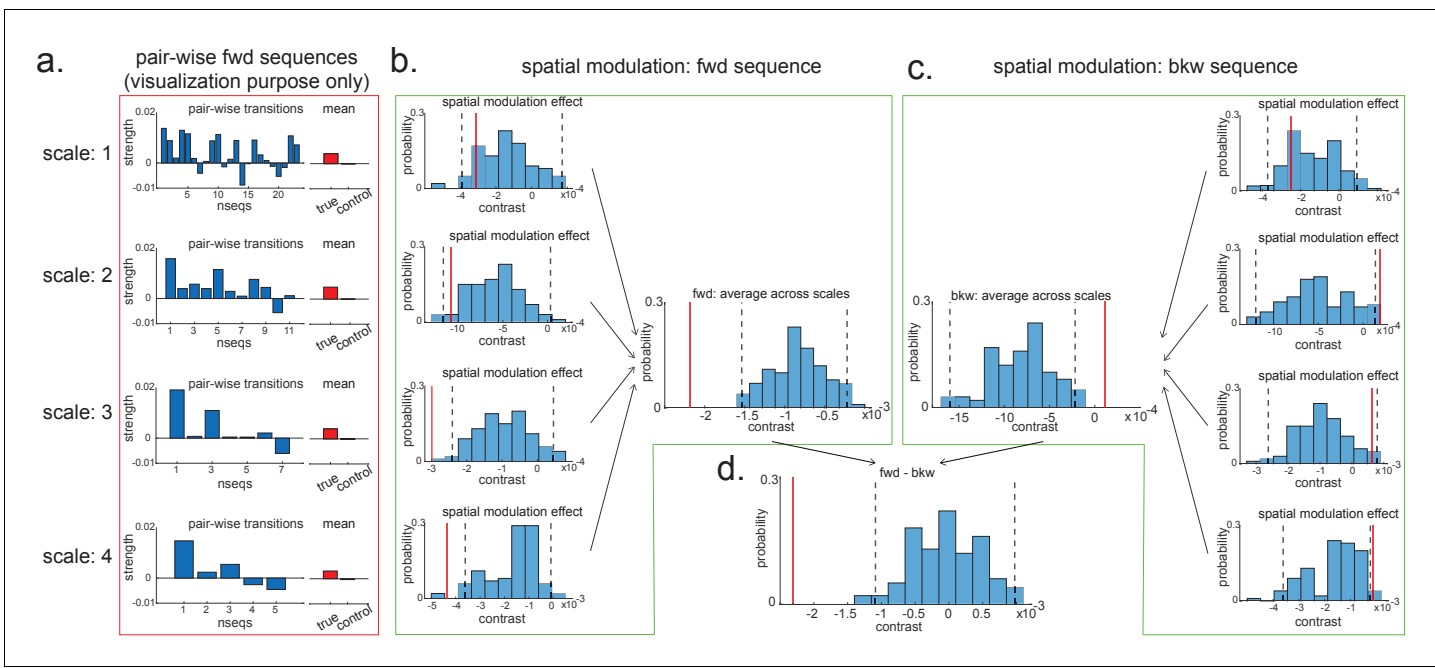

**Figure 8.** Pairwise sequence and spatial modulation effect. (**a**) Within each scale, strengths of each pairwise forward sequences in the region of interest (ROI) (significant replay speeds, compare with *Figure 7d*, green shading) are ordered from the start of maze to the end of the maze; alongside that, the mean sequence strength across all of these valid pairwise transitions is plotted (red) in comparison to the mean of all control transitions (grey). This is for visualization purpose only and is included in the red rectangle. (**b**) The contrast defining a linear change in forward sequenceness across the track (spatial modulation) is shown (red line), both separately for each scale, and average across scales, and compared to permutations. On average, forward replay is stronger at the beginning of the track. (**c**) Same as panel (**b**), but this is for the backward sequences. Unlike forward replay, backward replay is stronger at the end of the track. Note that both panels (**b**) and (**c**) are about spatial modulation effect, which is orthogonal to overall sequence strength, allowing valid inference. They are therefore included in green boxes. (**d**) The difference of this spatial modulation effect between forward and backward sequence is also significant. The black dotted lines indicate the 2.5th and 97.5th percentile of the permutation samples. The red solid line indicates the estimate of the true contrast effect.

To formally test the spatial modulation effect, we can use the exact same approach as outlined above in section 'Linear contrasts'. Here, we test a linear increase or decrease across different *transitions*. We take the linear contrast weight vector, *c* ([-2,-1,0,1,2] for the largest scale, [-3:3] for the next scale, [-5:5] for the next scale, and [-12:12] for the smallest scale), and multiply these by the beta estimates of the transitions:

$$contrast = c^T \beta \tag{17}$$

If this new measure, *contrast*, is different from zero, then there is a linear increase/decrease from one end of the track to the other. Note that this new contrast is no longer biased by the ROI selection as each transition contributed equally to the ROI selection, but we are now comparing between transitions. Inference on this contrast is therefore valid. We have therefore put them in green boxes to match *Figure 7f* (*Figure 8b, c*).

Within the larger two scales, these contrasts are significantly negative (tested against permutations in exactly the same way as the 'mean' contrasts). Since we are still in the linear domain, we can now just average these contrasts across the four scales and get a single measure for spatial modulation of replay. This average measure is significantly negative (*Figure 8b*). Hence, on average, forward replay is stronger at the beginning of the track.

We can do the same thing for backward replay. We found an opposite pattern, that is, strength of backward replay is stronger at the end of the track, and similarly, it is not significant in the smallest scale and becomes significant in the largest scale, and also significant on average across all scales (*Figure 8c*). Again, since we are in the linear domain, we can further contrast these contrasts, asking if this effect is different for forward and backward replay. We found that the difference is indeed significant (*Figure 8d*). This set of results is consistent with previous rodent literature (*Diba and Buzsáki, 2007*). Note that we would like to stress again that this analysis is not about a single replay event but is testing for average differences across all replay events.

Notably, extra care needs to be exercised for second-order questions (compared to first-order ones). Problems can emerge due to biases in second-order inference, such as in behavioural sampling (e.g. track 1 may be experienced more than track 2 during navigation; this creates a bias when evaluating replay in tack 1 vs. track 2 during rest). Such issues are real but can be finessed by experimental design considerations of a sort commonly applied in the human literature. For example:

1. Ensure that biases that might occur within subjects will not occur consistently in the same direction *across subjects* (e.g. by randomizing stimuli across participants).
2. Compare across conditions in each subject.
3. Perform a random effects inference across the population by comparing against the between-subject variance.

Such approaches are not yet common in rodent electrophysiology and may not be practical in some instances. In such cases, it remains important to be vigilant to guard against these biases with TDLM as with other techniques. If these approaches are feasible, the machinery for computing second-order inferences is straightforward in a linear framework like TDLM.

## Generality of TDLM

We have now discussed the applicability of TDLM in relation to human MEG, as well as in rodent electrophysiology (with comparisons to standard replay detection methods). A preliminary attempt at detecting replay in human EEG is also shown in Appendix 4. We believe that this establishes TDLM as a domain-general sequence analysis method: TDLM works at the level of decoded state space, rather than the sensor/cell level of the data. It can be applied to a wide range of data types and settings in both humans and rodents, stimulating cross-fertilization across disciplines. It is based on the GLM framework, and this lends it flexibility for regressing out potential confounds while offering an intuitive understanding of the overall approach.

In this section, we discuss the generality of TDLM.

### States

TDLM assesses the statistical likelihood of certain transitions on a graph. In its original form, TDLM works on discrete states (i.e. nodes in the graph). Continuous spaces can be incorporated by

chunking them into discrete spaces. Furthermore, by averaging the same replay speeds measured at multiple scales of discretization (see section 'TDLM for rodent replay'), the statistical benefits of an assumption of a Euclidean geometry can be recovered.

## Time length

The longer the time length, the more accurate the estimates in TDLM. This is because TDLM assesses sequence evidence based on a GLM framework, where time length is the sample size. Higher sample size will lead to more accurate estimates. In the case of rodent analysis, we recommend applying TDLM to aggregated replay events rather than to a single event because this results in (1) more time samples for estimation and (2) more activated states in the analysis time framework. Unlike other techniques which search for a single replay in a single event, this aggregation can be implemented without losing generality as TDLM is able to handle multiple sequences in the same data with respect to different directions, contents, or speeds. Furthermore, by aggregating linearly across all replay events of the same condition, it provides a natural measure for comparing replay strength, speed, and direction across different experimental conditions.

TDLM has already proved important in human experiments where complex state spaces have been used (*Wimmer et al., 2020*; *Liu et al., 2019*; *Liu et al., 2021a*; *Kurth-Nelson et al., 2016*). We expect this generality will also be important as rodent replay experiments move beyond 1D tracks, for example, to foraging in 2D, or in complex mazes.

## Discussion

TDLM is a domain-general analysis framework for capturing sequence regularity of neural representations. It is developed on human neuroimaging data and can be extended to other data sources, including rodent electrophysiology recordings. It offers hope for cross-species investigations on replay (or neural sequences in general) and potentially enable studies of complex tasks in both humans and animals.

TDLM adds a new analysis toolkit to the replay field. It is especially suited for summarizing replay strength across many events, comparing replay strength between conditions, and analysing replay strength in complex behavioural paradigms. Its linear modelling nature makes it amenable to standard statistical tests and thereby allows wide use across tasks, modalities, and species. Unlike alternative tools, we have not shown TDLM applied to individual replay events.

The temporal dynamics of neural states have been studied previously with MEG (*Vidaurre et al., 2017*; *Baker et al., 2014*). Normally such states are defined by common physiological features (e.g. frequency, functional connectivity) during rest and termed resting state networks (e.g. default mode network [*Raichle et al., 2001*]). However, these approaches remain agnostic about the *content* of neural activity. The ability to study the temporal dynamics of representational content permits richer investigations into cognitive processes (*Higgins et al., 2021*) as neural states can be analysed in the context of their roles with respect to a range of cognitive tasks.

Reactivation of neural representations has also been studied previously (*Tambini and Davachi, 2019*) using approaches similar to the decoding step of TDLM or multivariate pattern analysis (MVPA) (*Norman et al., 2006*). This has proven fruitful in revealing mnemonic functions (*Wimmer and Shohamy, 2012*), understanding sleep (*Lewis and Durrant, 2011*), and decision-making (*Schuck et al., 2016*). However, classification alone does not reveal the rich *temporal structures* of reactivation dynamics. We have described the application of TDLM mostly during off-task state in this paper. The very same analysis can be applied to on-task data to test for cued sequential reactivation (*Wimmer et al., 2020*) or sequential decision-making (*Eldar et al., 2020*). For example, the ability to detect sequences on-task allows us to tease apart clustered from sequential reactivation, where this may be important for dissociating decision strategies (*Eldar et al., 2018*) and their individual differences (*Wimmer et al., 2020*; *Eldar et al., 2020*). TDLM, therefore, may allow testing of neural predictions from process models such as reinforcement learning during task performance (*Dayan and Daw, 2008*), which have proved hard to probe previously (*Wimmer et al., 2020*; *Nour et al., 2021*; *Liu et al., 2019*; *Liu et al., 2021a*).

In the human neuroimaging domain, we have mainly discussed the application of TDLM with respect to MEG data. In Appendix 4, we show that TDLM also works well with EEG data. This is not surprising given EEG and MEG are effectively measuring the same neural signature, namely local

field potential (or associated magnetic field) on the scalp. We do not have suitable fMRI data to test TDLM. However, related work has suggested that it might be possible to measure sequential reactivation using fMRI (*Schuck and Niv, 2019*), but particular methodological caveats need to be considered (e.g. a bias from last events due to slow haemodynamic response) (*Wittkuhn and Schuck, 2021*). We believe that TDLM can deal with this, given it models out non-specific transitions, although further work is needed. In future, we consider it will be useful to combine the high temporal resolution available in MEG/EEG and the spatial precision available in fMRI to probe region-specific sequential computations.

In the rodent electrophysiology domain, we show what TDLM (its multi-scale version) has to offer uniquely compared to existing rodent replay methods. Most importantly, TDLM works on an arbitrary graph and its generality makes replay studies in complex mazes possible. Its linear framework makes the assessment of time-varying effect on replay (*Figure 7*) or other second-order sequence questions straightforward. In future work, a promising direction will be to further separate process noise (e.g. intrinsic variability within sequences) and measurement noise (e.g. noise in MEG recording). This might be achieved by building latent state-space models as have been explored recently in rodent community (*Maboudi et al., 2018*; *Denovellis, 2020*).

Together, we believe that TDLM opens doors for novel investigations of human cognition, including language, sequential planning, and inference in non-spatial cognitive tasks (*Eldar et al., 2018*; *Kurth-Nelson et al., 2016*), as well as complicated tasks in rodents, for example, forging in 2D mazes. TDLM is particularly suited to test specific neural predictions from process models, such as reinforcement learning. We hope that TDLM can promote an across-species synthesis between experimental and theoretical neuroscience and, in so doing, shed novel light on neural computation.

## Materials and methods

### Simulating MEG data
We simulate the data so as to be akin to human MEG.

#### Task data for obtaining state patterns
We generate ground truth multivariate patterns (over sensors) of states. We then add random Gaussian noise on the ground truth state patterns to form the task data. We train a logistic regression classifier on the task data so as to obtain a decoding model for each of the state patterns. Later we use this decoding model to transform the resting-state data from sensor space (with dimension of time by sensors) to the state space (with dimension of time by states).

#### Rest data for detecting sequences
First, to imitate temporal autocorrelations and spatial correlations commonly seen in human neuroimaging data, we generate the rest data using an autoaggressive model with multivariate (over sensors) Gaussian noise and add a dependence among sensors. In some simulations, we also add a rhythmic oscillation (e.g. 10 Hz).

Second, we inject a sequence of state patterns in the rest data. The sequences follow the ground truth of state transitions of interest. The state-to-state time lag is assumed to follow a gamma distribution. We vary the number of sequences to be injected in the rest data to control the strength of sequences.

Lastly, we project the rest data to the decoding model of states obtained from the task data. TDLM will then work on the decoded state space.

An example of the MATLAB implementation is called 'Simulate_Replay' from the Github link: https://github.com/yunzheliu/TDLM (copy archived at swh:1:rev: 015c0e90a14d3786e071345760b97141700d6c85), *Liu, 2021b*.

### Human replay dataset
#### Task design
Participants were required to perform a series of tasks with concurrent MEG scanning (see details in *Liu et al., 2019*). The functional localizer task was performed before the main task and was used to train a sensory code for eight distinct objects. Note that the participants were provided with no

structural information at the time of the localizer. These decoding models, trained on the functional localizer task, capture a sensory-level neural representation of stimuli (i.e. stimulus code). Following that, participants were presented with the stimuli and were required to unscramble the 'visual sequence' into a correct order, that is, the 'unscrambled sequence' based on a structural template they had learned the day before. After that, participants were given a rest for 5 min. At the end, stimuli were presented again in random order, and participants were asked to identify the true sequence identity and structural position of the stimuli. Data in this session are used to train a structural code (position and sequence) for the objects.

## MEG data acquisition, preprocessing, and source reconstruction

We follow the same procedure that has been reported in *Liu et al., 2019*. We have copied it here for references.

'MEG was recorded continuously at 600 samples/s using a whole-head 275-channel axial gradiometer system (CTF Omega, VSM MedTech), while participants sat upright inside the scanner. Participants made responses on a button box using four fingers as they found most comfortable. The data were resampled from 600 to 100 Hz to conserve processing time and improve signal-to-noise ratio. All data were then high-pass-filtered at 0.5 Hz using a first-order IIR filter to remove slow drift. After that, the raw MEG data were visually inspected, and excessively noisy segments and sensors were removed before independent component analysis (ICA). An ICA (FastICA, http://research.ics.aalto.fi/ica/fastica) was used to decompose the sensor data for each session into 150 temporally independent components and associated sensor topographies. Artefact components were classified by combined inspection of the spatial topography, time course, kurtosis of the time course, and frequency spectrum for all components. Eye-blink artefacts exhibited high kurtosis (>20), a repeated pattern in the time course and consistent spatial topographies. Mains interference had extremely low kurtosis and a frequency spectrum dominated by 50 Hz line noise. Artefacts were then rejected by subtracting them out of the data. All subsequent analyses were performed directly on the filtered, cleaned MEG signal, in units of femtotesla.

All source reconstruction was performed in SPM12 and FieldTrip. Forward models were generated on the basis of a single shell using superposition of basis functions that approximately corresponded to the plane tangential to the MEG sensor array. LCMV beamforming (*Van Veen et al., 1997*) was used to reconstruct the epoched MEG data to a grid in MNI space, sampled with a grid step of 5 mm. The sensor covariance matrix for beamforming was estimated using data in either broadband power across all frequencies or restricted to ripple frequency (120–150 Hz). The baseline activity was the mean neural activity averaged over $-100$ ms to $-50$ ms relative to sequence onset. All non-artefactual trials were baseline corrected at source level. We looked at the main effect of the initialization of sequence. Non-parametric permutation tests were performed on the volume of interest to compute the multiple comparison (whole-brain corrected) p-values of clusters above 10 voxels, with the null distribution for this cluster size being computed using permutations (n = 5000 permutations)'.

## Rodent replay dataset

### Data description

This data is from *Ólafsdóttir et al., 2016*. We analysed one full recording session (track running for generating rate map, post-running resting for replay detection) from Rat 2192.

### Task description

In *Ólafsdóttir et al., 2016*, rats ran multiple laps on a Z maze and were then placed in a rest enclosure. The two parallel sections of the Z (190 cm each) were connected by a diagonal section (220 cm). Animals were pretrained to run on the track. At the recording session, rats were placed at one end of the Z-track. The ends and corners of the track were baited with sweetened rice to encourage running from one end to the other. In each session, rats completed 20 full laps (30–45 min). Following the track session, rats were placed in the rest enclosure for 1.5 hr.

## Preprocessing

Following *Ólafsdóttir et al., 2016*, when generating rate maps we excluded data from both the ends and corners because the animals regularly performed non-perambulatory behaviours there. Periods when running speed was less than 3 cm/s were also excluded. Running trajectories were then linearized, dwell time and spikes were binned into 2 cm bins and smoothed with a Gaussian kernel ($\sigma$ = 5 bins). We generated rate maps separately for inbound (track 1 -> track 2 -> track 3) and outbound (track 3 -> track 2 -> track 1) running.

As in *Ólafsdóttir et al., 2016*, cells recorded in CA1 were classified as place cells if their peak firing field during track running was above 1 Hz and at least 20 cm wide. The candidate replay events were identified based on MU activity from place cells during rest time. Only periods exceeding the mean rate by three standard deviations of MU activity were identified as putative replay events. Events less than 40 ms long or which included activity from less than 15% of the recorded place cell ensemble were rejected.

We analysed data from one full recording session (track running for generating rate map, post-running resting for replay detection) of Rat 2192 reported in *Ólafsdóttir et al., 2016*. Following the procedure described above, we have identified 58 place cells and 1183 putative replay events. Replay analysis was then performed on the putative replay events, separately for inbound and out-bound rate maps.

## Code availability

Source code of TDLM can be found at https://github.com/yunzheliu/TDLM.

## Data availability

Data are also available at https://github.com/yunzheliu/TDLM.

## Acknowledgements

We thank Matthew A Wilson for help with rodent theta sequence analysis. We thank Elliott Wimmer and Toby Wise for helpful discussion and generous sharing of their data. We thank Matt Nour for helpful comments on a previous version of the manuscript. YL is also grateful for the unique opportunity provided by the Brains, Minds and Machines Summer Course. We acknowledge fundings from the Open Research Fund of the State Key Laboratory of Cognitive Neuroscience and Learning to YL, Wellcome Trust Investigator Award (098362/Z/12/Z) to RJD, Wellcome Trust Senior Research Fellowship (104765/Z/14/Z), and Principal Research Fellowship (219525/Z/19/Z), together with a James S McDonnell Foundation Award (JSMF220020372), to TEJB; and Wellcome Trust Senior Research Fellowship (212281/Z/18/Z) to CB. Both Wellcome Centres are supported by core funding from the Wellcome Trust: Wellcome Centre for Integrative Neuroimaging (203139/Z/16/Z), Wellcome Centre for Human Neuroimaging (091593/Z/10/Z). The Max Planck UCL Centre is a joint initiative supported by UCL and the Max Planck Society.

## Additional information

### Competing interests

Timothy E Behrens: Senior editor, *eLife*. Zeb Kurth-Nelson: Zeb Kurth-Nelson is affiliated with Deep-Mind. The author has no other competing interests to declare. The other authors declare that no competing interests exist.

### Funding

| Funder | Grant reference number | Author |
| --- | --- | --- |
| Wellcome | 098362/Z/12/Z | Raymond J Dolan |
| Wellcome | 104765/Z/14/Z | Timothy E Behrens |
| Wellcome | 219525/Z/19/Z | Timothy E Behrens |
| James S. McDonnell Founda- | JSMF220020372 | Timothy E Behrens |

tion

| Wellcome | 212281/Z/18/Z | Caswell Barry |
|---|---|---|
| Max Planck Society | | Yunzhe Liu |
| Wellcome | 203139/Z/16/Z | Timothy E Behrens |
| Wellcome | 091593/Z/10/Z | Raymond J Dolan |

The funders had no role in study design, data collection and interpretation, or the decision to submit the work for publication.

### Author contributions

Yunzhe Liu, Conceptualization, Data curation, Software, Formal analysis, Validation, Investigation, Visualization, Methodology, Writing - original draft, Project administration, Writing - review and editing; Raymond J Dolan, Zeb Kurth-Nelson, Supervision, Writing - original draft, Writing - review and editing; Cameron Higgins, Writing - review and editing; Hector Penagos, Mark W Woolrich, Writing - original draft; H Freyja Ólafsdóttir, Resources; Caswell Barry, Resources, Writing - review and editing; Timothy E Behrens, Supervision, Investigation, Methodology, Writing - original draft, Project administration, Writing - review and editing

### Author ORCIDs

Yunzhe Liu https://orcid.org/0000-0003-0836-9403
Raymond J Dolan http://orcid.org/0000-0001-9356-761X
Timothy E Behrens http://orcid.org/0000-0003-0048-1177

### Ethics

Human subjects: The human dataset used in this study was reported in Liu et al 2019. All participants were recruited from the UCL Institute of Cognitive Neuroscience subject pool, had a normal or corrected-to-normal vision, no history of psychiatric or neurological disorders, and had provided written informed consent prior to the start of the experiment, which was approved by the Research Ethics Committee at University College London (UK), under ethics number 9929/002.

### Decision letter and Author response

Decision letter https://doi.org/10.7554/eLife.66917.sa1
Author response https://doi.org/10.7554/eLife.66917.sa2

## Additional files

### Supplementary files

• Transparent reporting form

### Data availability

No new data is used or generated in the current paper. Data relevant for current paper is available at https://github.com/YunzheLiu/TDLM (copy archived at https://archive.softwareheritage.org/swh:1:rev:015c0e90a14d3786e071345760b97141700d6c85). This dataset is from Ólafsdóttir et al., 2016.

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

# Appendix 1

## Multi-step sequences

TDLM can be used iteratively. One extension of TDLM of particular interest is: multi-step sequences. It asks about a consistent regularity among multiple states.

So far, we introduced methods for quantifying the extent to which the state-to-state transition structure in neural data matches a hypothesized task-related transition matrix. An important limitation of these methods is that they are blind to hysteresis in transitions. In other words, they cannot tell us about multi-step sequences. In this section, we describe a methodological extension to measure evidence for sequences comprising more than one transition: for example, $A \rightarrow B \rightarrow C$.

The key ingredient is controlling for shorter sub-sequences (e.g. $A \rightarrow B$ and $B \rightarrow C$) in order to find evidence unique to a multi-step sequence of interest.

Assuming constant state-to-state time lag, $\Delta t$, between A and B, and between B and C. We can create a new state space AB by shifting B up $A \rightarrow B$, and element-wise multiply it with state A. This new state AB measures the reactivation strength of $A \rightarrow B$, with time lag $\Delta t$. In the same way, we can create a new state space, BC, AC, etc. Then we can construct the same first-level GLM on the new state space. For example, if we want to determine the evidence of $A \rightarrow B \rightarrow C$ at time lag $\Delta t$, we can regress AB onto state time course C at each $\Delta t$ (**Equation 1**). But we want to know the unique contribution of AB to C. More specifically, we want to test if the evidence of $A \rightarrow B \rightarrow C$ is stronger than $X \rightarrow B \rightarrow C$, where X is any other state but not A. Therefore, similar to **Equation 2**, we need to control CB, DB, when looking for evidence of AB of C. Applying this method, we show that TDLM successfully avoids false positives arising out of strong evidence for shorter length (see simulation results in **Appendix 1—figure 1a**, and see results obtained on human neuroimaging data in **Appendix 1—figure 1b**). This process can be generalized to any number of steps.

TDLM, in its current form, assumes a constant intra-sequence state-to-state time lag. If there is a variability between state transitions TDLM can still cope, but not very elegantly. Assume there is a three-state sequence, $A \rightarrow B \rightarrow C$, with intra-sequence variance. TDLM will need to test all possible combinations of state-to-state time lags in $A \rightarrow B$ and $B \rightarrow C$. If there are $n$ number of time lag of interest in either of the two transitions, TDLM will then have to test $n^2$ possible time lag combinations. This is a large search space and one that increases exponentially as a function of the length of a sequence.

We note that this analysis is different from a typical rodent replay analysis which assesses the overall evidence for a sequence length (**Davidson et al., 2009**; **Grosmark and Buzsáki, 2016**). TDLM asks if there is more evidence for A -> B -> C, above and beyond evidence for B -> C, for example. If the main question of interest is 'do we have evidence of A -> B -> C in general', as normally is the case in the rodent replay analysis (**Davidson et al., 2009**; **Grosmark and Buzsáki, 2016**), we should not control for shorter lengths. Instead, we can simply average the evidence together, as implemented in **Kurth-Nelson et al., 2016**.

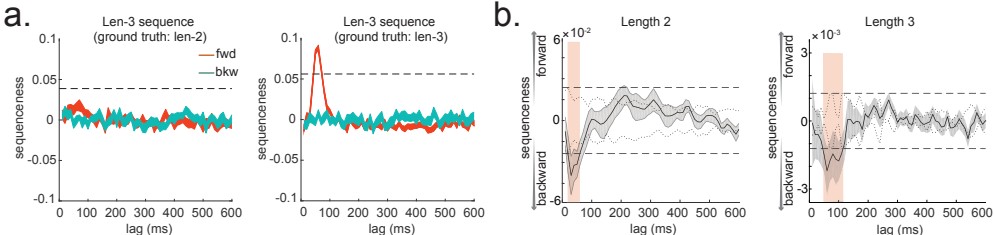

**Appendix 1—figure 1.** Extension to temporal delayed linear modelling (TDLM): multi-step sequences. (**a**) TDLM can quantify not only pairwise transition, but also longer length sequences. It does so by controlling for evidence of shorter length to avoid false positives. (**b**) Method applied to human MEG data, incorporating control of both alpha oscillation and co-activation for length-2 and length-3 sequence length. Dashed line indicates the permutation threshold. This is reproduced from Figure 3A, C, **Liu et al., 2019**, *Cell*, published under the Creative Commons Attribution 4.0 International Public License (CC BY 4.0).

# Appendix 2

## Pseudocode of sensory code and abstract code cross-validations

Algorithm 1. Hold-one-out cross-validation to compute classification accuracy.
Here $N$ is the number of trials, $D$ is the number of data dimensions, and $P$ is the number of classes.

---

**Algorithm 1: Hold-one-out cross-validation**

---

**Input:** Dataset $\mathfrak{D} = \{X_i, Y_i\}_{i=1}^N (X_i \in R^D; Y_i \in Z_2^P)$
**Output:** Cross validated classification accuracy $\{a \in R: 0 \leq a \leq 1\}$
Randomly partition $\mathfrak{D}$ into $K = \frac{N}{P}$ equally sized subsets, $\mathfrak{D} = \{\mathfrak{D}_1, \mathfrak{D}_2, \ldots, \mathfrak{D}_K\}$ such that each $\mathfrak{D}_i$ contains a single random sample from each class $\mathfrak{y}$;
**for** $k$ in $K$ **do**
 Create a training dataset $\mathfrak{T}_k = \{\mathfrak{D}_i : i \neq k\}$;
 Train a logistic regression classifier $\beta_k$ on $\mathfrak{T}_k$;
 Compute classification accuracy $a_k$ of $\beta_k$ on $\mathfrak{D}_k$;
**end**
Compute mean accuracy $a = \frac{1}{K} \sum\limits_{k=1}^{K} a_k$

---

Algorithm 2. Test a classifier's abstraction ability across different datasets with a common structure.

---

**Algorithm 2: Classifier abstraction**

---

Testing abstraction ability of classifiers over different datasets with a common structure.
**Input:** Dataset $\mathfrak{D} = \{X_i, Y_i\}_{i=1}^N (X_i \in R^D; Y_i \in \{A, B, C, D, A', B', C', D'\})$
**Output:** Abstraction accuracy $\{a \in R: 0 \leq a \leq 1\}$
Randomly partition $\mathfrak{D}$ into two subsets each of which exclusively contain trials from one or other structure sequence:
$\mathfrak{D}_1 = \{X_i, Y_i\}_{i=1}^N : X_i \in R^D; Y_i \in \{A, B, C, D\}$ and $\mathfrak{D}_2 = \{X_i, Y_i\}_{i=1}^N : X_i \in R^D; Y_i \in \{A', B', C', D'\}$
**for** $k$ in $\{1, 2\}$ **do**
 Train a logistic regression classifier $\beta_k$ on $\mathfrak{D}_k$;
 Compute classifier predictions $p_k$ of $\beta_k$ on $\mathfrak{D}_{3-k}$;
 Compute abstraction accuracy $a_k$ as proportion of samples for which the prediction $p_k$ correctly identifies the sequence location (e.g., $A$ predicted for $A'$);
**end**
Compute mean abstraction accuracy $a = \frac{1}{2} \sum\limits_{k=1}^{2} a_k$

---

## Appendix 3

### Sequences of sequences

We have detailed the use of either sensory or abstract representations as the states in TDLM. We now take a step further and use sequences themselves as states. Using this kind of hierarchical analysis, we can search for sequences of sequences. This is useful because it can reveal temporal structure not only within sequence, but also between sequences. The organization between sequences is of particular interest for revealing neural computations. For example, the forward and backward search algorithms hypothesized in planning and inference (*Penny et al., 2013*) can be cast as sequences of sequences problem: the temporal structure of forward and backward sequence. This can be tested by using TDLM iteratively.

To look for sequences between sequences, we need first to define sequences as new states. To do so, the raw state course, for example, state B, needs to be shifted up by the empirical within-sequence time lag $\Delta t$ (determined by the two-level GLM), to align with the onset of state A, if assuming sequence $A \to B$ exist (at time lag $\Delta t$). Then, we can element-wise multiply the raw state time course A with the shifted time course B, resulting in a new state AB. Each entry in this new state time course indicates the reactivation strength of sequence AB at a given time.

The general two-level GLMs framework still applies, but now with one important caveat. The new sequence state (e.g. AB) is defined based on the original states (A and B), and where we are now interested in a reactivation regularity, that is, sequence, between sequences, rather than the original states. We need therefore to control for the effects of the original states. Effectively, this is like controlling for main effects (e.g. state A and shifted state B) when looking for their interaction (sequence AB). TDLM achieves this by including time-lagged original state regressors A, B, in addition to AB, in the first-level GLM sequence analysis.

Specifically, let us assume that the sequence state matrix is $X_{seq}$, after transforming the original state space to sequence space based on the empirical within-sequence time lag $\Delta t_w$. Each column at $X_{seq}$ is sequence state, denoted by $S_{ij}$, which indicates the strength of sequence $i \to j$ reactivation. The raw state $i$ is $X_i$, and the shifted raw state $j$ is $X_{jw}$ (by time lag $\Delta t_w$).

In the first level GLM, TDLM ask for the strength of a unique contribution of sequence state $S_{ij}$ to $S_{mn}$ while controlling for original states ($X_i$ and $X_{jw}$). For each sequence state $ij$, at each possible time lag $\Delta t$, TDLM estimated a separate linear model:

$$S_{mn} = X_i(\Delta t)\beta_i + X_{jw}(\Delta t)\beta_j + S_{ij}(\Delta t)\beta_{ij}(\Delta t) \tag{18}$$

Repeat this process for each sequence state separately at each time lag, resulting in a sequence matrix $\beta_{seq}$.

At the second-level GLM, TDLM asks how strong the evidence for a sequence of interest is compared to sequences that have the same starting state or end state at each time lag. This second-level GLM will be the same as *Equation 5*, but with additional regressors to control for sequences that share the same start or end state. In simulation, we demonstrate, applying this method, that TDLM can uncover hierarchical temporal structure: state A is temporally leading state B with 40 ms lag, and the sequence A -> B tends to repeat itself with a 140 ms gap (*Appendix 3—figure 1a*). One interesting application of this is to look for theta sequence (*Mehta et al., 2002*; *McNaughton et al., 2006*; *Buzsáki and Moser, 2013*). One can think of theta sequence, a well-documented phenomenon during rodent spatial navigation, as a neural sequence repeating itself in theta frequency (6–12 Hz).

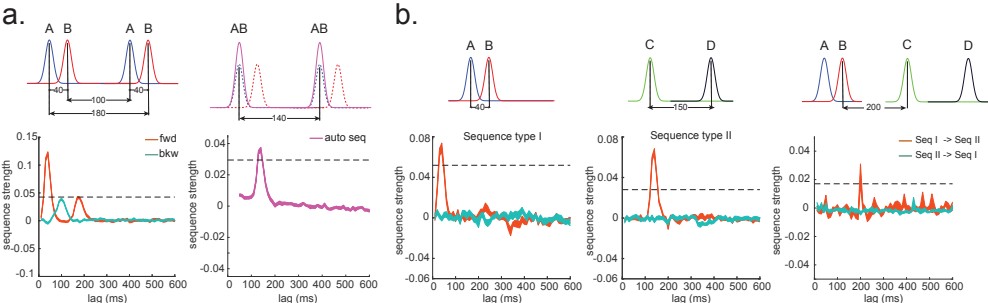

**Appendix 3—figure 1.** Sequences of sequences. (**a**) Temporal delayed linear modelling (TDLM) can also be used iteratively to capture the repeating pattern of a sequence event itself. Illustration in the top panel describes the ground truth in the simulation. Intra-sequence temporal structure (left) and inter-sequence temporal structure (right) can be extracted simultaneously. (**b**) Temporal structure between and within different sequences. Illustration of two sequence types with different state-to-state time lag within sequence, and a systematic gap between the two types of sequences on top. TDLM can capture the temporal structures both within (left and middle panel) and between (right panel) the two sequence types.

In addition to looking for temporal structure of the same sequence, the method is equally suitable when searching for temporal relationships between different sequences. For example, assuming two different types of sequences, one sequence type has a within-sequence time lag at 40 ms; while the other has a within-sequence time lag at 150 ms (*Appendix 3—figure 1b*, left and middle panel); and there is a gap of 200 ms between the two types of sequences (*Appendix 3—figure 1b*, right panel). These time lags are set arbitrarily for illustration purposes. TDLM can accurately capture the dynamics both within and between the sequences, supporting a potential for uncovering temporal relationships between sequences under the same framework.

## Appendix 4

### Apply TDLM to human whole-brain EEG data

An autocorrelation is commonplace in neuroimaging data, including EEG and fMRI. TDLM is designed to specifically take care of this confound, and, on this basis, we should be able to work with EEG and fMRI data. We do not have suitable fMRI data available to test TDLM but are interested to investigate this in more depth in our future work. We had collected EEG data from one participant to test whether TDLM would *just* work.

The task was designed to examine on-task sequential replay in decision-making by Dr. Toby Wise. This is a 'T-maze' like task, where a participant needs to choose a left or right path based on the value received at the end of the path. We could decode seven objects well on the whole-brain EEG data using just raw amplitude feature (same with our MEG-based analysis) and could detect fast backward sequenceness (peaked at 30 ms time lag) during choice/planning time (*Appendix 4—figure 1*), similar to our previous MEG findings (*Kurth-Nelson et al., 2016*). As this result is from one subject, we are cautious about making an excessive claim, but nevertheless we believe that the data show the TDLM approach is highly promising for EEG data.

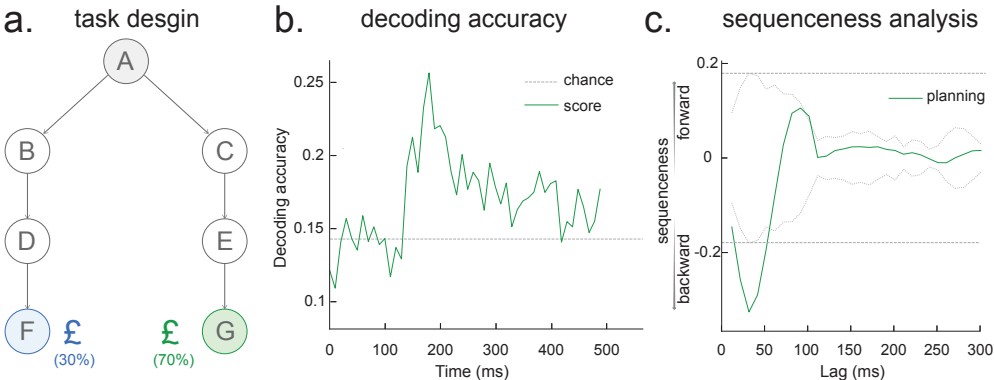

**Appendix 4—figure 1.** Sequence detection in EEG data (from one participant). (**a**) Task design. At each trial, the participant starts at state A, and he/she needs to select either 'BDF' or 'CEG' path, based on the final reward receipt at terminal states F or G. All seven states are indexed by pictures. (**b**) The leave-one-out cross-validated decoding accuracy is shown, with a peak at around 200 ms after stimulus onset, similar to our previous MEG findings. (**c**) Temporal delayed linear modelling method is then applied on the decoded state time course where we find a fast backward sequenceness that conforms to task structure. Shown here is a subtraction between forward and backward sequenceness, where a negative sequenceness indicates stronger backward sequence replay. The dotted line is the peak of the absolute state permutation at each time lag, the dashed line the max over all computed state time lags, thereby controlling for multiple comparisons. This is the same statistical method used in our previous work and in the current paper. These EEG sequence results replicate our previous MEG-based findings at planning/decision time (see Figure 3 in *Kurth-Nelson et al., 2016* and also see Figure 3f in *Liu et al., 2019*).

## Appendix 5

### Less sensitivity of TDLM to skipping sequences

In a linear track where replays only go in a single direction, it is possible that TDLM is less sensitive compared to the linear correlation or the Radon method, given the latter assumes a parametric relationship between space and time. For example, if only the first and last states are activated, but not the intermediate states, the existing methods *will* report replay, but TDLM will not, because in existing methods space and time are parametric quantities (*Appendix 5—figure 1*). In contrast, TDLM only knows about transitions on a graph.

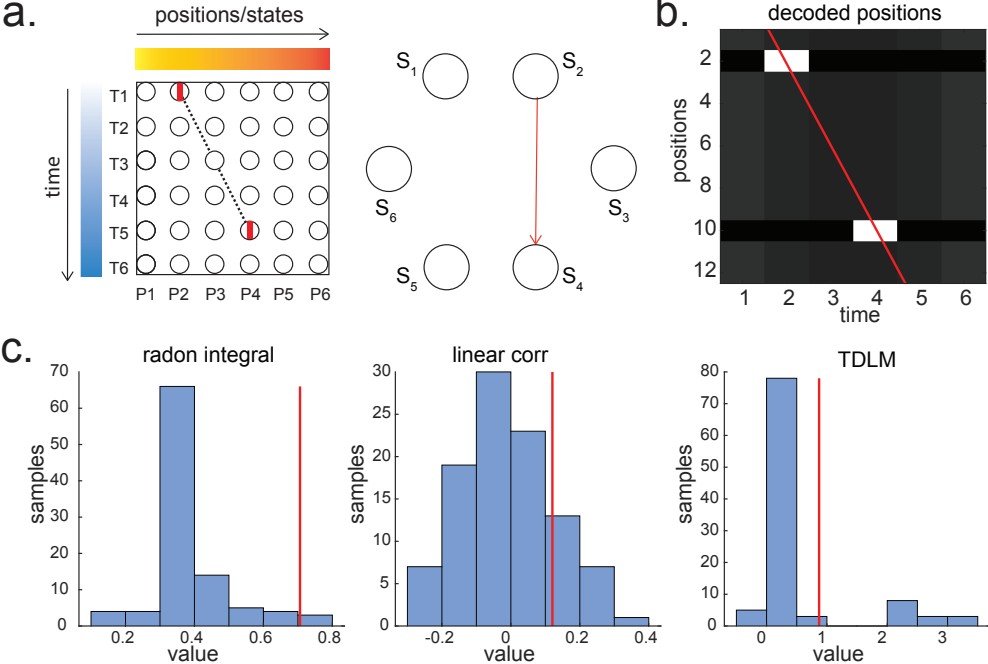

**Appendix 5—figure 1.** Parametric relationship between space and time vs. graph transitions. (a) A scheme for skipping sequence (left). Both Radon and linear weighted correlation methods aim to capture a parametric relationship between space and time. Temporal delayed linear modelling (TDLM), on the other hand, tries to capture transitions in a graph (shown in right, with the red indicating the transition of interest). (b) A decoded time by position matrix from simulated spiking data. (c) Replay analysis using all three methods. TDLM is less sensitive compared to existing 'line search' methods, like radon or linear correlation. The red line indicates the true sequence measure from each of these methods. The bar plots are permutation samples by randomly shuffling the rate maps.

