## [Decision Letter]

**Acceptance summary:**

This paper represents a valuable new tool for detecting and quantifying sequential structure in neural activity. Spontaneously generated sequences (a.k.a. "replay") is thought to be an important way for the brain to make efficient use of experience, facilitating learning and consolidation of information. Replay has been studied in the rodent hippocampus for decades, but it has recently become possible to detect such activity in human MEG (and perhaps even fMRI) data, generating much current excitement and promise in bringing together these fields. The approach of this work enables investigators to assess the overall level of replay present in a dataset (rather than locating individual events). In particular, a strength of the general linear modeling framework is that both the primary hypothesis "is replay prevalent in this data set?" and secondary hypotheses (e.g., "is there more of one type of replay than another?", "does replay strength change over time?") can be assessed.

**Decision letter after peer review:**

[Editors’ note: the authors submitted for reconsideration following the decision after peer review. What follows is the decision letter after the first round of review.]

Thank you for submitting your work entitled "Measuring Sequences of Representations with Temporally Delayed Linear Modelling" for consideration by *eLife*. Your article has been reviewed by 3 peer reviewers, one of whom is a member of our Board of Reviewing Editors, and the evaluation has been overseen by a Senior Editor. The following individuals involved in review of your submission have agreed to reveal their identity: Matthijs van der Meer (Reviewer #2), and Caleb Kemere (Reviewer #3).

Our decision has been reached after consultation between the reviewers. Based on these discussions and the individual reviews below, we regret to inform you that we have decided to reject the paper in its current form.

The reviewers all felt that the work is extremely valuable; a domain-general replay detection method would be of wide interest and utility. However, we felt that the paper was lacking context and comparisons to existing methods. Most critically, the paper would be more impactful if comparisons with standard replay methods were included, and the reviewers felt that would be too substantial a change to ask for as a revision. There were also concerns about lack of detail in the description of the methods and data that diminished enthusiasm. The authors would be welcome to make changes along these lines and submit the paper again as a new submission.

*Reviewer #1:*

This paper describes temporal delayed linear modelling (TDLM), a method for detecting sequential replay during awake rest periods in human neuroimaging data. The method involves first training a classifier to decode states from labeled data, then building linear models that quantify the extent to which one state predicts the next expected state at particular lags, and finally assessing reliability by running the analysis with permuted labels.

This method has already been fruitfully used in prior empirical papers by the authors, and this paper serves to present the details of the method and code such that others may make use of it. Based on existing findings, the method seems extremely promising, with potential for widespread interest and adoption in the human neuroimaging community. The paper would benefit, however, from more discussion of the scope of the applicability of the method and its relationship to methods already available in the rodent and (to a lesser extent) human literature.

1. TDLM is presented as a general tool for detecting replay, with special utility for non-invasive human neuroimaging modalities. The method is tested mainly on MEG data, with one additional demonstration in rodent electrophysiology. Should researchers expect to be able to apply the method directly to EEG or fMRI data? If not, what considerations or modifications would be involved?

2. How does the method relate to the state of the art methods for detecting replay in electrophysiology data? What precludes using those methods in MEG data or other noninvasive modalities? And conversely, do the authors believe animal replay researchers would benefit from adopting the proposed method?

3. It would be useful for the authors to comment on the applicability of the method to sleep data, especially as rodent replay decoding methods are routinely used during both awake rest and sleep.

4. How does the method relate to the Wittkuhn and Schuck fMRI replay detection method? What might be the advantages and disadvantages of each?

5. The authors make the point that spatial correlation as well as anti-correlation between state patterns reduces the ability to detect sequences. The x axis for Figure 3c begins at zero, demonstrating that lower positive correlation is better than higher positive correlation. Given the common practice of building one classifier to decode multiple states (as opposed to a separate classifier for each state), it would be very useful to provide a demonstration that the relationship in Figure 3c flips (more correlation is better for sequenceness) when spatial correlations are in the negative range.

6. In the Results, the authors specify using a single time point for spatial patterns, which would seem to be a potentially very noisy estimate. In the Methods, they explain that the data were downsampled from 600 to 100 Hz to improve SNR. It seems likely that downsampling or some other method of increasing SNR will be important for the use of single time point estimates. It would be useful for the authors to comment on this and provide recommendations in the Results section.

7. While the demonstration that the method works for detecting theta sequences in navigating rodents is very useful, the paper is missing the more basic demonstration that it works for simple replay during awake rest in rodents. This would be important to include to the extent that the authors believe the method will be of use in comparing replay between species.

8. The authors explain that they "had one condition where we measured resting activity before the subjects saw any stimuli. Therefore, by definition these stimuli could not replay, but we can use the classifiers from these stimuli (measured later) to test the false positive performance of statistical tests on replay." My understanding of the rodent preplay literature is that you might indeed expect meaningful "replay" prior to stimulus exposure, as existing sequential dynamics may be co-opted to represent subsequent stimulus sequences. It may therefore be tricky to assume no sequenceness prior to stimulus exposure.

*Reviewer #2:*

This paper addresses the important overall issue of how to detect and quantify sequential structure in neural activity. Such sequences have been studied in the rodent hippocampus for decades, but it has recently become possible to detect them in human MEG (and perhaps even fMRI) data, generating much current excitement and promise in bringing together these fields.

In this paper, the authors examine and develop in more detail the method previously published in their ground-breaking MEG paper (Liu et al. 2019). The authors demonstrate that by aiming their method at the level of decoded neural data (rather than the sensor-level data) it can be applied to a wide range of data types and settings, such as rodent ephys data, stimulating cross-fertilization. This generality is a strength and distinguishes this work from the typically ad hoc (study-specific) methods that are the norm; this paper could be a first step towards a more domain-general sequence detection method. A further strength is that the general linear modeling framework lends itself well to regressing out potential confounds such as autocorrelations, as the authors show.

However, enthusiasm for the paper is limited by several overall issues:

1. It seems a major claim is that the current method is somehow superior to other methods (e.g. from the abstract: "designed to take care of confounds" implying that other methods do not do this, and "maximize sequence detection ability" implying that other methods are less effective at detection). But there is very little actual comparison with other methods made to substantiate this claim, particularly for sequences of more than two states which have been extensively used in the rodent replay literature (see Tingley and Peyrache, Proc Royal Soc B 2020 for a recent review of the rodent methods; different shuffling procedures are applied to identify sequenceness, see e.g. Farooq et al. Neuron 2019 and Foster, Ann Rev Neurosci 2017). The authors should compare their method to some others in order to support these claims, or at a minimum discuss how their method relates to/improves upon the state of the art.

2. The scope or generality of the proposed method should be made more explicit in a number of ways. First, it seems the major example is from MEG data with a small number of discrete states; how does the method handle continuous variables and larger state spaces? (The rodent ephys example could potentially address this but not enough detail was provided to understand what was done; see specific comments below.) Second, it appears this method describes sequenceness for a large chunk of data, but cannot tell whether an individual event (such as a hippocampal sharp wave-ripple and associated spiking) forms a sequence not. Third, there is some inconsistency in the terminology regarding scope: are the authors aiming to detect *any* kind of temporal structure in neural activity (first sentence of "Overview of TDLM" section) which would include oscillations, or only sequences? These are not fatal issues but should be clearly delineated.

3. The inference part of the work is potentially very valuable because this is an area that has been well studied in GLM/multiple regression type problems. However, the authors limit themselves to asking "first-order" sequence questions (i.e. whether observed sequenceness is different from random) when key questions – including whether or not there is evidence of replay – are actually "second-order" questions because they require a comparison of sequenceness across two conditions (e.g. pre-task and post-task; I'm borrowing this terminology from van der Meer et al. Proc Royal Soc B 2020). The authors should address how to make this kind of comparison using their method.

*Reviewer #3:*

The methods used by the authors seem like potentially really useful tools for research on neural activity related to sequences of stimuli. We were excited to see that a new toolbox might be available for these sorts of problems, which are widespread. The authors touch on a number of interesting scenarios and raise relevant issues related to cross-validation and inference of statistical significance. However, given (1) the paucity of code that they've posted, and its specificity to specific exact data and (2) the large literature on latent variable models combined with surrogate data for significance testing, I would hesitate to call TDLM a "framework". Moreover, in trying to present it in this generic way, the authors have muddled the paper, making it difficult to understand exactly what they are doing.

Overall: This paper presents a novel approach for detecting sequential patterns in neural data however it needs more context. What's the contribution overall? How and why is this analysis technique better than say Bayesian template matching? Why is it so difficult to understand the details of the method?

The first and most important problem with this paper is that it is intended (it appears) to be a more detailed and enhanced retelling of the author's 2019 Cell paper. If this is the case, then it's important that it also be clearer and easier to read and understand than that one was. The authors should follow the normal tradition in computational papers:

1. Present a clear and thorough explanation of one use of the method (i.e., MEG observations with discrete stimuli), then present the next approach (i.e., sequences?) with all the details necessary to understand it.

2. The authors should start each section with a mathematical explanation of the X's – the equation(s) that describes how they are derived from specific data. Much of the discussion of cross validation actually refers to this mapping.

3. Equation 5 also needs a clearer explanation – it would be better to write it as a sum of matrices (because that is clearer) than with the strange "vec" notation. And TAUTO, TF and TR should be described properly – TAUTO is "the identity matrix", TF and TR are "shift matrices, with ones on the first upper and lower off diagonals".

4. The cross validation schemes need a clear description. Preferably using something like a LaTeX "algorithm" box so that they are precisely explained.

Recognizing the need to balance readability for a general reader and interest, perhaps the details could be given for the first few problems, and then for subsequent results, the detail could go into a Methods section. Alternatively, the methods section could be done away with (though some things, such as the MEG data acquisition methods are reasonably in the methods).

Usually, we think about latent variable model problems from a generative perspective. The approach taken in this paper seems to be similar to a Kalman filter with a multinomial observation (which would be equivalent to the logistic regression?), but it's unclear. Making the connection to the extensive literature on dynamical latent variable models would be helpful.

[Editors’ note: further revisions were suggested prior to acceptance, as described below.]

Thank you for resubmitting your work entitled "Temporally delayed linear modelling (TDLM) measures replay in both animals and humans" for further consideration by *eLife*. Your revised article has been reviewed by 3 reviewers and the evaluation has been overseen by Laura Colgin as the Senior Editor, and a Reviewing Editor.

While impressed by the novelty and potential utility of the method, the reviewers had a specific critical concern. Namely, do high sequenceness scores truly capture activation of patterns that widely span the specified sequence space (i.e., many complete "ABCDE" sequences) or only a collection of pairwise correlations (i.e., "AB", "DE", "BC")? Presenting more examples from experimental data that demonstrate the former was perceived as critical to demonstrate the utility of TDLM as an approach for "replay detection". Ultimately, the reviewers reached a consensus that in the current presentation, what TLDM actually detects remains opaque, and the impact of the work is diminished. We considered requesting an action plan, but upon reflection, I think that the main issue is one of semantics. However, if you wish to describe TDLM as a method for detecting "replay", it needs to be critically addressed.

In addition to the reviews below, in our discussion, one reviewer noted: "even though they now explain in more detail what they did to analyze theta sequences, the result (~12 Hz) is still seemingly at odds with the ~8-9 Hz repetition one would expect from the literature. I'm actually not sure this adds a whole lot to the paper so I think it would be better to just take this part out."

*Reviewer #1:*

Overall, I find great value in the effort to provide researchers working with very different animal models and datasets a similar toolkit to apply and analyze reactivation and replay. But I also have significant concerns about the potential for these methods, if poorly understood and applied, to further confound the field. Fully understanding this paper and the described methods and its caveats is not easy for most experimentalists, yours truly included. I am concerned that investigators will misapply these tools and claim significant replay in instances where there is none. These concerns may be addressable by better diagnostics and related guidance for interpretation.

Nevertheless, an important caveat in the work is that it does not detect "replay" per se, but rather temporal biases in decoded activity. Thus I think the title should be amended. In some places, the authors describe this as "sequenceness", or "temporal delays" which are both preferable to "replay". Prior work (e.g. Skaggs and McNaughton) used a similar measure, but referred to it as "temporal bias". While this temporal bias is certainly related to replay and sequences, it is only an indirect measure of these, and more akin to "reactivation", as it's largely pairwise analyses. Clarity on such issues is particularly important in this area given the excessive ambiguity in terminology for the replay and reactivation phenomena.

My other major concern is that the analysis is rather opaque in an area where there is much need for transparency, especially considering the existing debates and controversy surrounding appropriate methodology and conclusions that can be drawn from it. For example, in most of the figures that the authors provide, it's unclear whether the sequenceness score is driven by one particular pair of stimuli, or equally so among most possible pairs. Perhaps a transition graph could be composed from the data, but I did not find this except in cartoon diagrams. I think it would be important for the authors to provide guidance regarding these details. A related question is whether these biased pairs typically appear in isolation, or as part of larger multi-item events? It's not clear if there is a straightforward way to extract this important information. Some sample events could be shown, but examples can also be cherry-picked and non-representative. Probably a histogram of sequence bout lengths would be valuable.

Part of the claimed value in these methods is their possible application to spike trains, e.g. from rodents, particularly for detecting and describing individual events. The authors claim that this is possible. However, while they analyze two rodent datasets in different parts, they do not apply it to any real data, but only on rather contrived (low noise) simulated data. This raises the concern that TDLM is not sufficiently sensitive for detecting individual events. The theta sequence analysis results shown in Supplementary Figure 3d are also concerning to me. They show a very noisy signal that passes threshold in one particular bin. If such a figure were a major result in a new manuscript, and the current *eLife* manuscript was cited as validation for the methods, would reviewers be obliged to accept it for publication? If not, what additional criterion or diagnostic steps would be recommended?

Comments for the authors:

P2, Line 32: "and" seems misplaced.

P2, Line 34: "periods of activity".

P3, Line 4: perhaps "single neuron" rather than "cellular".

P4, Lines 34-35: it is not clear here what the authors mean by "over and above variance that can be explained by other states at the same time." It gets more clear later in page 11, section "Moving to multiple linear regression". The authors might consider either referring to that section at the end of the sentence or removing the unnecessary details that might confuse the reader at this juncture.

P5, Line 31: This sentence is a bit confusing. It is not clear what "in which" refers to. It might be better to start a new sentence for describing Zf and Zb.

P6, Lines 7-8: The authors might refer to the section "Correcting for multiple comparisons" on page 16, where more details are provided.

P8, Lines 42-46: the description of abstraction may benefit from additional background and exposition.

P9, Line 18-24, I found this entire paragraph very confusing.

P10, Line 7: "that share the same abstract representation" is redundant.

P10, Line 12: "tested" should be corrected to test it.

P10, Lines 23-24: Confusing sentence with multiple "whiches".

P12, Line 24: Is it possible for the auto-correlation structure of Xi(t) to generate false positives for Xj(t+dt), or would this necessarily be entirely accounted for by Xj(t)?

P13, Lines 23-24: How the regularization (penalizing for the norm of W) is supposed to make the model "trained on stimulus-evoked data" generalize better to off-task/rest data? The authors might add a sentence or two at the end of the paragraph to make the aim more clear and motivate the next paragraphs in the section. Based on the descriptions in the first paragraph of Page 14, the regularization seems to add more sparsity to the estimated W that minimizes spatial correlations between the states, etc. Something similar to these descriptions could be used here.

P13, Line 28: It is not exactly clear what authors mean by "the prior term"? Does it refer to the equation before adding any regularization or to some prior probability distribution over W? I think in this context, we should be cautious with using the words like prior, posterior, etc.

P16, Line 30, extra "as".

P17, section "Considerations on second-order inferences". It seems that this section should be placed later in the manuscript, in the section "TDLM FOR RODENT REPLAY".

P23 Line 39, missing "that".

Supplementary Note 5: what shuffle was used?

*Reviewer #2:*

This paper addresses the important overall issue of how to detect and quantify sequential structure in neural activity. Spontaneously generated sequences (a.k.a. "replay") is thought to be an important way for the brain to make efficient use of experience, facilitating learning and consolidation of information. Replay has been studied in the rodent hippocampus for decades, but it has recently become possible to detect such activity in human MEG (and perhaps even fMRI) data, generating much current excitement and promise in bringing together these fields.

However, comparison and cross-fertilization between different replay studies – even within the same species, let alone across species – has been hampered by a fragmented landscape of different analysis methods, which are often ad-hoc and task-specific. In this study, the authors develop and extend the method previously published in their groundbreaking MEG paper (Liu et al. 2019), notably demonstrating that by aiming their method at the level of decoded neural data (rather than the sensor-level data) it can be applied to a wide range of data types and settings, including human MEG, EEG, and rodent ephys data. A further strength is that the general linear modeling framework lends itself well to regressing out potential confounds and formally testing second-order questions such as whether tthere is more forward vs. reverse replay, or if replay strength changes with time.

In this revised submission, the authors have made several major additions to the manuscript, including a substantive analysis of rodent replay data, and associated multi-scale extension of the method to make it suitable for continuous state spaces. This addition is an important component of the paper, in that it demonstrates the generality of a framework that can be applied to different kinds of data and task state spaces across species. Another much appreciated change is the expanded and much more clear explanations throughout, such as those of the TDLM method itself and of the data analysis pipelines for the various kinds of data. With these additions, I think the paper really makes good on the promise of a domain-general method for the detection of sequences in spontaneous neural activity, and provides a timely impetus to the study of replay across tasks and species.

There is one important area that the paper still falls short on, but that should be straightforward to address with some simple additional analysis and discussion. A distinctive strength of the GLM framework is that it easily accommodates testing important and ubiquitous "second-order" questions, such as whether there is more forward than reverse replay, is replay strength changing over time, and so on. However, the message in the current version that one could simply subtract sequenceness scores doesn't address how one would formally test for a difference, or test for some factor like time being important. For forward vs. reverse, because this is fit to the same data, this is a comparison between different betas in the second-level GLM (Figure 1f). I am not a statistician, but my textbooks say there are a few ways of doing this, for instance z = (β_fwd_ – β_rev_) / σ_(β_fwd_ – β_rev_) where the crucial variance term can be estimated as the sqrt(σ(β_fwd_)^2^ + σ(β_rev_)^2^), matching the intuition that a formal test requires an estimate of variance of the difference between the betas, not just the means.

For early vs. late sleep, things are more complicated because you are not comparing models/betas fit to the same data. I suppose the above approach could also work as long as all the betas are standardized, but wouldn't a better test be to include time as a regressor in the second-level GLM and formally test for an interaction between time and T_fwd_ (and T_bwd_)?

In either case, there are two important points to make to demonstrate the utility of the GLM framework for sequence analysis: (1) these ubiquitous second-order inference questions require a bit more consideration than just saying you can subtract the sequenceness scores; there is another level of statistical inference here, and (2) the TDLM framework, unlike many other approaches, is in fact very well suited to doing just that – it benefits from the existing machinery of linear statistical modeling and associated model comparison tools to make such comparisons.

These points are not currently clear from the "Considerations on second-order inferences" section. If in addition the GLM framework allows for the regressing out of some potential confounds that can come up in rodent data that the authors now reference that's an additional benefit but not the main point I am asking the authors to speak to.

*Reviewer #3:*

This revised manuscript describes a method for detecting "offline replay" events in human and animal electrophysiology. I have found the TDLM method described in the manuscript to be very valuable for the field. Its detailed description in the present manuscript will be very useful, first because it presents its different components very clearly and in detail – something which is not possible in a manuscript focused on a particular finding obtained using this method. Second, the authors show that this method can be applied not only to human MEG data, but also to rodent electrophysiology data.

I found the manuscript to be well written: it describes clearly and adequately the different analytic steps that have to be applied to avoid confounds arising from known features of electrophysiological signals (autocorrelations, oscillations), and from task features themselves. The fact that they show results obtained using their method from simulated and real data is also a strength of the manuscript. The level of computational detail regarding the equations appears adequate and well balanced for a broad readership, and example code using the method has been posted online.

I have not reviewed the original version of the manuscript, but have first read the revised manuscript before consulting the initial reviews and the responses provided by the authors.

The authors' responses to the initial reviews are extensive and insightful. I wonder how much the presentation of replay detection in EEG data from a *single* subject is particularly convincing, but nevertheless I agree with the authors that it shows that their method indeed *can* work using EEG data instead of MEG data.

The authors' extensive response regarding the comparison between their method and other state-of-the-art methods for detecting replay events in electrophysiology data was very clear and useful. The revised manuscript adequately includes a full section on the use of their method for rodent electrophysiology data. The authors also discuss their method in light of other efforts, from Wittkuhn and Schuck using fMRI data for example.

Regarding the more important reservations of Reviewer #2, I found that the authors provided adequate and convincing responses to each of them. Regarding the concerns of Reviewer #3, I found that the revised manuscript allowed understanding the method quite clearly relative to my limited understanding of the details of the method after reading recent empirical papers from the authors (e.g., Liu et al., 2019, Cell). The fact that the authors have posted code online on GitHub is also very useful, and I find that the level of detail regarding the equations is sufficient (and well balanced) for the broad readership of a journal like *eLife*.

[Editors' note: further revisions were suggested prior to acceptance, as described below.]

Thank you for resubmitting your work entitled "Temporally delayed linear modelling (TDLM) measures replay in both animals and humans" for further consideration by *eLife*. Your revised article has been evaluated by Laura Colgin as the Senior Editor, and a Reviewing Editor.

We shared the revised manuscript with the reviewers. After some discussion, it was concluded that the manuscript has been improved but there are some remaining issues that need to be addressed, as outlined below:

1. The reviewers were confused by the data in Figure 7e. We finally concluded that it was an attempt to explain how the regression was formed, but it took lots of back and forth. Given that this is a tools paper, there seems to be no reason why each analysis figure can't be backed with equations that identify the regressions being done, and the variables being regressed.

2. Figure 5 appears to be about analyzing the MEG data when events are detected. (Isn't TDLM an approach for measuring sequenceness over a population of events rather than finding single ones?) Even though this is previously published work, the methods need significant expansion (see Point 1). The text refers to a section that appears to be missing? Here's the text: "We want to identify the time when a given sequence is very likely to unfold. We achieve this, by transforming from the space of states to the space of sequence events. This is the same computation as in the section "States as sequence events". " (Search for "sequence events" yields no results.) Perhaps this refers to Appendix 3, but the text there doesn't really help much.

3. One reviewer had previously asked "in most of the figures that the authors provide, it's unclear whether the sequenceness score is driven by one particular pair of stimuli, or equally so among most possible pairs". To clarify, it seems that the question is: if the model proposed is A→B→C→D, and the data are largely A→B, can that be detected? Or alternatively, can you give a proper way of comparing two models (in this case, A→B→C→D vs A→B)?

---

## [Author Response]

[Editors’ note: the authors resubmitted a revised version of the paper for consideration. What follows is the authors’ response to the first round of review.]

The reviewers all felt that the work is extremely valuable; a domain-general replay detection method would be of wide interest and utility. However, we felt that the paper was lacking context and comparisons to existing methods. Most critically, the paper would be more impactful if comparisons with standard replay methods were included, and the reviewers felt that would be too substantial a change to ask for as a revision. There were also concerns about lack of detail in the description of the methods and data that diminished enthusiasm. The authors would be welcome to make changes along these lines and submit the paper again as a new submission.

We thank all reviewers for their positive evaluation of our work, and more so, for pointing to areas for improvement, including comparisons to existing rodent methods and absence of relevant details in the original submission. In this revised manuscript, we have described our approach – temporal delayed linear modelling (TDLM) in more detail, and with more focus on a comparison to existing methods.

Importantly, we have extended TDLM to better cope with continuous state spaces (as is normally the case in rodent experiments where physical positions are states to be decoded, while in human experiments, the states are often discrete, as indexed by visual stimuli). We call this method multi-scale TDLM.

We have also re-analysed rodent electrophysiology data from Ólafsdóttir, Carpenter, and Barry (2016) using our approach, and show TDLM can offer unique perspective compared to existing “line searching” methods in rodent replay analysis.

As a result, the paper is substantially re-written to highlight its generality as a sequence detection method with applicability to both rodents and humans. It is general because it can test any transition of interest on an arbitrary graph, and this goes beyond “line searching” on a linearized state space. This has already proved important in human experiments where complex state-spaces have been used. We expect this generality (by this method or others) will also be important as rodent replay experiments move beyond 1D tracks, for example to foraging in 2D, or in complex mazes.

Below we address all the reviewers’ comments in a point-by-point manner.

Reviewer #1:This paper describes temporal delayed linear modelling (TDLM), a method for detecting sequential replay during awake rest periods in human neuroimaging data. The method involves first training a classifier to decode states from labeled data, then building linear models that quantify the extent to which one state predicts the next expected state at particular lags, and finally assessing reliability by running the analysis with permuted labels.This method has already been fruitfully used in prior empirical papers by the authors, and this paper serves to present the details of the method and code such that others may make use of it. Based on existing findings, the method seems extremely promising, with potential for widespread interest and adoption in the human neuroimaging community. The paper would benefit, however, from more discussion of the scope of the applicability of the method and its relationship to methods already available in the rodent and (to a lesser extent) human literature.

We thank the reviewer for this positive feedback and agree it will be useful to discuss TDLM method in the context of existing ones. We have done so in the revised paper. Below we address each of the reviewer’s specific concerns and questions.

1. TDLM is presented as a general tool for detecting replay, with special utility for noninvasive human neuroimaging modalities. The method is tested mainly on MEG data, with one additional demonstration in rodent electrophysiology. Should researchers expect to be able to apply the method directly to EEG or fMRI data? If not, what considerations or modifications would be involved?

Yes, we expect this same method can be applied to human EEG, fMRI, as well as rodent electrophysiology data. In the revision, we show how TDLM can be extended to work on rodent replay during sleep (as detailed later below), We also applied TDLM to real human EEG data to demonstrate its applicability. We did not have suitable fMRI data at hand to test TDLM, but we believe the same procedure should work on fMRI as well. In the revision, we include this point in the Discussion section, along with analysis results on real rodent data in the main text (detailed later when responding to Q2), and analysis on human EEG in the supplemental material under the section “Apply TDLM to human whole-brain EEG data”. Below we have copied the relevant text in discussion, and supplemental material for reference:

Main text – Discussion, page 25-26, line 45-5

“In the human neuroimaging domain, we have mainly discussed the application of TDLM with respect to MEG data. […] In future, we consider it will be useful to combine the high temporal resolution available in M/EEG and the spatial precision available in fMRI to probe region specific sequential computations.”

Supplemental material – Supplementary Note 4: Apply TDLM to human whole-brain EEG data

**“**Supplementary Note 4: Apply TDLM to human whole-brain EEG data

An autocorrelation is commonplace in neuroimaging data, including EEG and fMRI. and[…] These EEG sequence results replicate our previous MEG-based findings based on analyses at planning/decision time (see Figure 3 in Kurth-Nelson et al., 2016, and also see Figure 3f in Liu et al., 2019).”

2. How does the method relate to the state of the art methods for detecting replay in electrophysiology data? What precludes using those methods in MEG data or other noninvasive modalities? And conversely, do the authors believe animal replay researchers would benefit from adopting the proposed method?

This is a great question. We thank all three reviewers for bringing this to our attention. To answer this, we discuss TDLM in comparison to existing rodent methods, in both simulation and real data, including rodent hippocampal electrophysiology (place cells in CA1) and human whole-brain MEG. To show the benefit of detecting replay using TDLM in rodents, we have now re-analyzed the data from Ólafsdóttir et al. (2016).

Firstly, we would like to highlight a key difference between TDLM and popular existing techniques (both methods we compare against below, and all methods we know about). These existing techniques rely on a continuous parametric embedding of behavioural states and the relationship between this embedding time (parametrically encoded). As far as we know existing techniques only use 1D embeddings, but this could likely be generalised. Essentially, they are looking for a line/linear relationship between time and decoded positions. We will henceforth refer them as “line search” approaches.

TDLM is fundamentally different to this as it operates on a graph and tests the statistical likelihood of some transitions happening more than others. This is therefore a more general approach that can be used for sequences drawn from any graph, not just graphs with simple embeddings (like a linear track).

For example, in a non-spatial decision-making task (Kurth-Nelson et al., 2016), all states lead to two different states and themselves can be arrived at from two other different states (Figure 6). Existing methods will not work here because there is no linear relationship between time and states.

TDLM vs. existing rodent methods

TDLM works on a decoded state space, rather than sensor (with analogy to cell) level. We compared TDLM to rodent methods that work on the posterior decoded position (i.e., state) space, normally referred to as Bayesian-based methods (Tingley and Peyrache, 2020). Note, the “Bayesian” part is the decoding of an animals’ ‘location’ during rest/sleep based on spike counts and mean firing rate (Zhang, Ginzburg, McNaughton, and Sejnowski, 1998), not replay detection. Two commonly used methods are Radon transform (Davidson, Kloosterman, and Wilson, 2009) and linear weighted correlation (Grosmark and Buzsáki, 2016).

Both methods proceed by forming a 2D matrix, where one dimension is the decoded state (e.g., positions on a linear track), and the other dimension is time (note that, as stated above, a decoded state is embedded in 1D). These approaches then endeavour to discover if an ordered line is a good description of the relationship between state and (parametric) time. For this reason, we call this family of approaches “line search”.

The radon method uses a discrete Radon transform to find the best line in the 2D matrix (Toft, 1996) and then evaluates the radon integral, which will be high if the data lie on a line (Author response image 1). It compares this to permutations of the same data where the states are reordered (Tingley and Peyrache, 2020). The linear weighted correlation method computes the average correlation between the time and estimated position in the 1D embedding (Author response image 1). The correlation is non-zero provided there is an orderly reactivation along the state dimension.

Both these methods are performed on decoded positions, where these are sorted based on their order in the linearized state space. TDLM also works on the decoded position space, but instead of directly measuring the relationship between position and time, it measures the transition strength for each possible state to state transitions (Author response image 1).

**Author response image 1. sa2fig1:** Illustration of three replay detection methods on the decoded time by position/state spaces. a. The Radon method tries to find the best fitting line (solid line) of the decoded positions as a function of time. The red bars indicate strong reactivation at given locations. b. The linear correlation method tests for a correlation between time and decoded position. c. The TDLM method on the other hand, does not directly measure relationship between state and time, but quantifies the strength of evidence for each possible transition, indicated by the solid black/grey dots, where the colour gradient indicates transition strength. For example, P5→P6 is lighter than P4→P5, this is because following reactivation of P5 in time T4, both P5 and P6 are reactivated at the same time – T5. Later this empirical transition matrix is compared to a theorical/ hypothesised one, to quantify the extent to which the empirical transitions fit with an experimental hypothesis.

Applying TDLM to candidate replay events

Single sequence in a candidate replay event

In a simple simulation of spiking data, all methods work equally well (Author response image 2). All replay analysis is performed on a decoded posterior position space (time* decoded positions). The permutation is implemented by shifting the rate map of each neuron. This is similar to our state identity-based permutation in TDLM. Effectively, they both invoke a null hypothesis that state identities (i.e., positions) are exchangeable. The results shown in TDLM is the sequenceness (red line) at a time lag of 2, which is the ground truth of the simulated sequence. The shuffled samples (blue, Author response image 2B) shown in TDLM is the sequenceness estimates on the time lag that gives rise to the strongest evidence over all computed time lags in the shuffled data (to control for multiple comparison). This is the same statistical approach performed in the current paper.

To be more specific on how TDLM is applied.

1. We follow the same procedure to obtain a rate map and position decoding as other methods. This decoded position matrix is a state space with dimension of number of time bins by number of position bins.

2. To use TDLM, we need to specify the transition matrix of interest. We generally use the pairwise transition matrix in the space, i.e., postion1→ position2, postion2→ position3, postion4→ position5, etc. We run TDLM at each time lag, normally from time bin 1 to time 10, which corresponding to 10 ms to 100 ms, where each time bin is 10 ms in this simulation.

3. We search for the time lag (over all computed time lags) that give rise to the highest sequenceness; we consider that as the sequenceness score for this ripple event.

4. To assess statistical significance, we take the peak time lag over all computed time lags in the simulation, and then take the 95% percentile on that peak (over permutation samples) as the significance threshold. This is the same procedure we outline in the current paper.

**Author response image 2. sa2fig2:** "line search” approach vs. TDLM on the simulated spiking data (assuming single ground-truth sequence). (**a**) The rate map of the simulated place cells (n=40) over a linearized space with 80 positions. It is smoothed with 2 sample gaussian kernel, to mimic overlapping place fields. (**b**) We simulated a ground truth sequence with time lag of 2 time samples between successive firings in the ripple event. The histogram is the sequence distribution of the shuffled data (in blue). The red line is the sequence results for the true data. The permutation is done by shuffling the rate map of each cell, so that the place fields are randomized. (**c**) We randomly shuffled the firing order in the ripple event, so that there is no structured sequence in the simulation. We show that all methods report non-significant evidence of sequenceness.

More than one sequences in a candidate replay event.

As we have indicated above, TDLM is a more general approach (allowing a broader range of experiments). Even in a 1D state-space, TDLM can be sensitive to replays that do not reflect a line in State-Time space.

To see this more clearly, we have simulated a regime where there are two sequences, but in perfect opposite directions, within one ripple event (Author response image 3). We use this simulation to demonstrate a situation where TDLM might be more suitable.

It is not surprising that the linear weighted correlation method would fail (Author response image 3), given it is looking for a general time and decoded space correlation (which would be near zero). The Radon method is still fine, if we ignore fitting lines exceed certain range (e.g., vertical lines here), but it will not be able to capture both sequences, as compared to TDLM. In situations with many candidate sequences, the Radon method will not be able to assess the evidence for each of them but will only focus on the best fitting line.

TDLM can capture both sequences because it looks for evidence of any transition of interest in an arbitrary graph, and it characterizes sequence strength as a function of direction and speed (not shown here, see more details in real rodent replay analysis below).

**Author response image 3. sa2fig3:** “line search” approach vs. TDLM on the simulated spiking data (assuming two ground-truth sequences). (**a**) The rate map of the simulated place cells (n=40) over a linearized space with 80 positions, smoothed with 2 sample gaussian kernel, to mimic overlapping place fields. (**b**) We simulated two ground truth sequences with time lag of 2 time samples between successive firings in the ripple event. We also show the decoded position space right next to the ripple event. All replay analyses are performed on this decoded space. The histogram is the sequence distribution of the shuffled data (in blue). The red line is the sequence results in the true data. (**c**) We randomly shuffled the firing order in the ripple event, so that there are no structured sequences in the simulation. We show there is no temporal structure in the decoded position space. As a result, TDLM reports no significant sequenceness value on the decoded position space.

Extending TDLM

In linear track where replays only go in a single direction, it is possible that TDLM is less sensitive compared to the linear correlation or the Radon method, given the latter assumes a parametric relationship between space and time. For example, if the first and last state alone are activated, but not intermediate states, the existing methods will report replay but TDLM will not, because in existing methods space and time are parametric quantities (Appendix 5—figure 1). In contrast, TDLM only knows about transitions on a graph.

Linear embedding (multiscale) TDLM in physical space

To solve this problem, we propose to perform TDLM in a linear embedding manner. The idea is to measure the same replay speed multiple times at different scales. For example, the speed of replay of 5cm per 10 ms, is the same as 10 cm per 20 ms, and 20 cm per 40 ms. Therefore, we can measure replay in multi-scale state spaces separately, and average the replay strength of the same speed across scales later. To take into account potential differences in signal to noise ratio between state spaces, we estimate not only transition strength but also the uncertainty in its estimate within each state space, so that at the end we can do precision weighted averaging across scales. This multi-scale approach has the benefit of not missing out on meaningful transitions e.g., state 1 → state 5 in original state space, and therefore could capture the parametric relationship between reactivated positions and time.

Specifically, to perform multi-scale TDLM, we discretise position bins as a function of width, for example, from 5 cm to 40 cm. This generates rate maps in different scales (e.g., 5 cm per bin, 10 cm per bin, 20 cm per bin, 40 cm per bin), and hence multi-scale state space. We then apply TDLM separately in each state space. We estimate both the replay strength and its uncertainty within each state space. This uncertainty estimate becomes important later when averaging the strength of the same replay speed across scales. Essentially, we are measuring the same thing multiple times, and average the measurements together while minimizing the variance.

In equations, the transition strength – 𝛽 is estimated by regressing the decoded state space – X to its lag copy – 𝑋(∆𝑡). In ordinary least squares (OLS) solution, 𝛽 is given by Equation 4 in the paper.

𝛽 = (𝑋^T^𝑋)^-1^𝑋^T^𝑋(∆𝑡)

The covariance matrix of 𝛽 can be estimated by

V = MSEC _*_(*X^T^X*)^-1^

Where, MSEC is the calibrated mean squared error, given by:

MSEC=∑i=1T(Xi(Δt)−Xiβ)2T−df 𝑇 is the number of time samples, and 𝑑𝑓 denotes the degrees of freedom consumed by model parameters. For OLS, each parameter takes away one degree of freedom.

For each replay speed of interest, we implement this separately in each state space and end up with [𝛽_1_ 𝑉_1_], [𝛽_2_, 𝑉_2_], [𝛽_2_, 𝑉_2_]… [𝛽_n_, 𝑉_n_], with 𝑛 being the number of scales. Precision weighted averaging of sequence strength for this replay speed can be performed as:

βM=∑i=1nβiVi∑i=1n1/Vi We do this for all replay speeds of interest, with statistical testing then performed on the precision weighted averaged sequence strength in a similar manner to what we do in original TDLM.

To render this concrete, we simulate a scenario where multi-scale TDLM is more sensitive, e.g., when there are gaps on a trajectory of interest. The multi-scale TDLM is more sensitive because it encompasses this path more often than the original TDLM (Author response image 4).

**Author response image 4. sa2fig4:** Multi-scale TDLM. (**a**) Illustration of a change in state space for the same replay speed. (**b**) A possible scenario for the application of multi-scale TDLM, where only subsets of state on a path were reactivated. The red line is the best linear fit based on Radon method. (**c**) TDLM method. TDLM with multi-scaling is more sensitive than TDLM in the original state space and reports significant sequenceness.

Apply multi-scale TDLM to real rodent data (place cells in CA1)

Next, we apply multi-scale TDLM to real rodent data to demonstrate what it can reveal. The unique benefit of TDLM is that it does not need to assume a parametric relationship between state and time, which make it ideal for detecting sequences in an open maze. But even in a linearized maze, it can provide unique perspectives. For example, it measures sequence strength as a function of speed, which is not the case in either Radon or linear correlation approach. More so, because TDLM is built on linear modelling, it has good statistical properties if one wants to ask a second order statistical question: e.g., Is the strength of forward sequence stronger than backward sequence? or is replay stronger in early vs. middle vs. late sleep stage. We illustrate this by applying multi-scale TDLM to CA1 place cells spiking data from Ólafsdóttir et al. (2016).

In Ólafsdóttir et al. (2016), rats ran multiple laps on a Z maze, and were then placed in a rest enclosure for 1.5 hours (Figure 7A). The Z maze consisted of 3 tracks, with its ends and corners baited with sweetened rice to encourage running from one end to the other. Following Ólafsdóttir et al. (2016), we excluded both the ends and corners when generating the rate map, given an animal regularly performs non-perambulatory behaviors at these locations. Periods when running speed were less than 3 cm/s were also excluded. The running paths were then linearized, and dwell time and spikes were binned into 2 cm bins, smoothed with a Gaussian kernel (σ = 5 bins). We generated rate maps separately for inbound (track1→track2→track3) and outbound (track3→track2→track1) running.

As in Ólafsdóttir et al. (2016), cells recorded in CA1 were classified as place cell if their peak firing field during track running was above 1 Hz with at least 20 cm width (see an example in Figure 7B). Candidate replay events were identified based on multi-unit (MU) activity from place cells during rest time. Only periods exceeding the mean rate by 3 stand deviation of MU activity were determined as putative replay events. Events less than 40 ms long, or which included activity from less than 15% of the recorded place cell ensemble, were rejected (see an example of putative replay event in Figure 7C).

We analyzed data from one full recording session (track running for generating rate map, post-running resting for replay detection) from Rat 2192, as reported in Ólafsdóttir et al. (2016). Following the procedure described above, we identified 58 place cells, and 1183 putative replay events. Replay analysis was then performed on these putative replay events, separately for inbound and outbound rate maps. Critically, because there are separate rate maps for inbound and outbound runs, we can separate forward and backward replay (as the same position has a different decoded state depending on whether it was obtained during an outbound or inbound run).

A forward sequence is defined as when states from the outbound map occur in the outbound order, or states from the inbound map occur in the inbound order. A backward sequence is when states from the inbound map occur in the outbound order or states from the outbound map occur in the inbound order. Candidate events were decoded based on a rate map, transforming the ncells * ntime to nstates * ntime. Each entry in this state space represents the posterior probability of being in this position at any given time. Replay analysis was performed solely on this decoded state space.

During the rest/sleep period, TDLM identified significant forward and backward sequences for both outbound and inbound maps with a wide speed range of 100 – 1000 cm/s, consistent with recent findings from Denovellis et al. (2020) on replay speed. In our analysis, the fastest speed (while still have a stronger evidence than the random sequences), is up to 1000 cm/s, which is around 20X faster than its corresponding free running speed, representing approximately half a track-arm in a typical replay event (e.g., 100 cm in 100 ms), consistent with previous work (Davidson et al., 2009; Karlsson and Frank, 2009; Lee and Wilson, 2002; Nádasdy, Hirase, Czurkó, Csicsvari, and Buzsáki, 1999).

Furthermore, because TDLM is a linear method, it can straightforwardly assess differences between replay in different conditions. For example, we can ask questions of the following form, is there more forward replay in early vs late sleep time? To answer this, we simply divide sleep into early (first 1/3 sleep time), middle (2/3), and late (the last 1/3 sleep), and average sequence evidence separately for different sleep times. We can see that sequence strength is stronger in early compared to middle and late sleep time, especially so for an outbound rate map (Author response image 5). Being able to perform such comparisons in a linear framework should be useful to the rodent research community.

**Author response image 5. sa2fig5:** TDLM applied to real rodent data. (**A**) Sequence strength as a function of speed and direction is shown for outbound rate map. Dotted line is the permutation threshold. We have both significant forward (blue) and backward (red) sequence with peak speed around 167 – 333 cm/s. (**B**) Sequence strength for an inbound rate map is shown. (**C**) Sequence strength estimated separately for early, middle and late sleep time, also for inbound, outbound rate map. There are more replays in the early sleep time than middle or late, especially in the speed range of 100-500 cm/s. Dotted line is the permutation threshold.

The utility of existing rodent methods for human neuroimaging data?

As discussed above, both Radon transform, and the linear correlation methods are essentially “line search” methods. These work well on identifying best trajectory (or linear time-position relationship) in a candidate replay event, but are less suitable at detecting multiple sequence, e.g., in varying speed, directions or contents. This makes it hard to apply these methods to human neuroimaging data, given for example it is unlikely that only one sequence will exist in a 5min resting period.

In addition, existing rodent methods treat the position estimates at each time bin separately and do not endeavour to control for co-activation and auto-correlation. But correlations in both time and space (e.g., voxels in fMRI, sensors in EEG/MEG) are common in neuroimaging data, and, if not controlled, they will lead to false positives (e.g., when compared to zero).

In sum, TDLM can be applied to rodent spiking data to detect replay and has flexibility to control for other confounding variables. These confound-controlling concerns, which may not necessarily be important for spiking data, are crucial when it comes to human neuroimaging data analysis. TDLM is also a more general sequence detection method given it does not require a linearized transition. Finally, by placing replay analyses in the GLM framework, it allows us to specify and test hypotheses (such as differential replay across conditions) with established powerful statistical inference procedures.

In the revision, we have devoted a whole new section titled “TDLM FOR RODENT REPLAY” in the main text to address this question. Under this section, there are four topics. They are “Generality of graph- vs line-based replay methods”; “Comparisons in simulated data”; “Multiscale TDLM to deal with continuous state space”; and “Applying multi-scale TDLM to real rodent data (place cells in CA1)”.

We have also included a supplemental note “Supplementary Note 5: Applying TDLM to candidate replay events” for detailed simulation results. There are three topics, namely “Single sequence in a candidate replay event”; “More than one sequence in a candidate replay event” and “Lesser sensitivity of TDLM to skipping sequences”.

3. It would be useful for the authors to comment on the applicability of the method to sleep data, especially as rodent replay decoding methods are routinely used during both awake rest and sleep.

This is a great suggestion. We have not worked with human sleep data before.

We think it should be possible, though there will be technical problems to solve along the way.

For example, multivariate classifiers trained on whole-brain data pattern are sensitive to both stimulus-specific patterns of activity and the specific background noise distribution. We would anticipate that this background noise distribution in particular would differ substantially between the awake and sleep state, potentially resulting in degraded classifier performance that may pose a challenge to replay detection.

In our own studies we have shown that classifiers with sparse L1 regularisation are more sensitive to replay (Liu et al., 2019; Wimmer, Liu, Vehar, Behrens, and Dolan, 2020). It is also worth noting that the L1-induced sparsity encodes weaker assumptions about background noise distributions into the classifiers as compared to L2 (Higgins, 2019). We expect that the use of sparse classifiers will be of greater importance when applied to sleep data, where background noise distributions differ more substantially from (awake state) training data.

We have added this point in the revised manuscript:

Main text – Regularization, page 14, line 10-15

“In addition to minimizing spatial correlations, as discussed above, it can also be shown that L1-induced sparsity encodes weaker assumptions about background noise distributions into the classifiers, as compared to L2 regularization (Higgins, 2019). […] Here, the use of sparse classifiers is helpful as background noise distributions are likely to differ more substantially from the (awake state) training data.”

4. How does the method relate to the Wittkuhn and Schuck fMRI replay detection method? What might be the advantages and disadvantages of each?

Wittkuhn and Schuck applied a simple linear regression between the serial position of the stimuli and their classification probabilities at every TR. This is similar to the linear weighted correlation method applied in rodent studies. There is no issue with respect to statistical inference given they are implicitly comparing to the strengths of other sequences. The problem is with generality. As indeed noted by these authors, their approach does not permit an examination of forward and backward sequence separately (Wittkuhn and Schuck, 2020). We make this clear in the discussion.

5. The authors make the point that spatial correlation as well as anti-correlation between state patterns reduces the ability to detect sequences. The x axis for Figure 3c begins at zero, demonstrating that lower positive correlation is better than higher positive correlation. Given the common practice of building one classifier to decode multiple states (as opposed to a separate classifier for each state), it would be very useful to provide a demonstration that the relationship in Figure 3c flips (more correlation is better for sequenceness) when spatial correlations are in the negative range.

We believe there is a misunderstanding here, and we apologise for lack of clarity. The X axis in Figure 3c is the absolute value of the correlation, and indeed when there is a higher negative correlation this leads to lower sequence detection. We took the absolute value here because the direction of correlation is not important, only the degree of the shared covariance matters. We add this point in the revised manuscript.

Main text – Spatial correlations, page 13, line 5-10

“Unfortunately, positive or negative correlations between states reduces the sensitivity of sequence detection, because it is difficult to distinguish between states within the sequence: collinearity impairs estimation of *β* in Equation 2[…] We took the absolute value here because the direction of correlation is not important, only the magnitude of the correlation matters.”

6. In the Results, the authors specify using a single time point for spatial patterns, which would seem to be a potentially very noisy estimate. In the Methods, they explain that the data were downsampled from 600 to 100 Hz to improve SNR. It seems likely that downsampling or some other method of increasing SNR will be important for the use of single time point estimates. It would be useful for the authors to comment on this and provide recommendations in the Results section.

In this paper, we try to focus on the sequence detection for the reactivation of representations, the principal aim of TDLM. As such, we have not explored different combinations of pre-processing pipelines (e.g., down sampling, averaged or single time bin) or selection of the raining time point. It is possible some other ways of preprocessing and feature selection is better than just training classifiers on a single time point, but we have not explored this. For simplicity, and for the purposes of this paper, which is already long and detailed, we hope the reviewer will allow us to leave this issue for future work.

7. While the demonstration that the method works for detecting theta sequences in navigating rodents is very useful, the paper is missing the more basic demonstration that it works for simple replay during awake rest in rodents. This would be important to include to the extent that the authors believe the method will be of use in comparing replay between species.

This is a great suggestion. In answering Q2, we have demonstrated both in simulation and real data, that TDLM is able to detect replay in hippocampal ripple data.

8. The authors explain that they "had one condition where we measured resting activity before the subjects saw any stimuli. Therefore, by definition these stimuli could not replay, but we can use the classifiers from these stimuli (measured later) to test the false positive performance of statistical tests on replay." My understanding of the rodent preplay literature is that you might indeed expect meaningful "replay" prior to stimulus exposure, as existing sequential dynamics may be co-opted to represent subsequent stimulus sequences. It may therefore be tricky to assume no sequenceness prior to stimulus exposure.

This is a good point. We would add that this is different to the preplay phenomena observed in rodent literature. In rodents, preplay happens before the rodent enters the novel maze, the authors of the relevant paper suggest this is due to a pre-defined canonical dynamic in the hippocampus (Dragoi and Tonegawa, 2011, 2014). Crucially, this is possible in physical space because the transitions and representations can contain information about these fixed relationships. One cell can always fire before another, and cellular relationship can *determine* which cells fire later at which location, and therefore what the decoder looks like.

The analogy in our experiment is that we might build a decoder at the time the sequence is experienced (as is done in rodent studies). Then the decoder might potentially rely on 2 pieces of information – the stimulus and the position in the sequence. These representations might indeed preplay, because pre-existing position representations might be built into the representation of the stimulus in the sequence. We have shown this is true, and indeed that the position portion of this representation does replay (Liu et al., 2019).

However, the replay we are looking at here cannot preplay, as not only is the resting data acquired before seeing the sequence, but crucially also the stimulus classifiers are built before the subject sees the sequence. Thus, they do not contain any representation of sequence position. Furthermore, the order of images in the sequence is randomised across subjects. On this basis there is no means for a subject to know which sequence to preplay.

We have added this point to the section “Distribution of sequenceness at a single lag” in the revised manuscript.

“We have tested this also on real MEG data. […] Indeed, in our case, each subject saw the same stimuli in a different order. They could not know the correct stimulus order when these resting data were acquired.”

Reviewer #2:[…] Enthusiasm for the paper is limited by several overall issues:1. It seems a major claim is that the current method is somehow superior to other methods (e.g. from the abstract: "designed to take care of confounds" implying that other methods do not do this, and "maximize sequence detection ability" implying that other methods are less effective at detection). But there is very little actual comparison with other methods made to substantiate this claim, particularly for sequences of more than two states which have been extensively used in the rodent replay literature (see Tingley and Peyrache, Proc Royal Soc B 2020 for a recent review of the rodent methods; different shuffling procedures are applied to identify sequenceness, see e.g. Farooq et al. Neuron 2019 and Foster, Ann Rev Neurosci 2017). The authors should compare their method to some others in order to support these claims, or at a minimum discuss how their method relates to/improves upon the state of the art.

We thank the reviewer for the helpful suggestion. In the original manuscript we wrote “designed to take care of confounds”, “maximize sequence detection ability” with the cross-correlation method (Eldar, Bae, Kurth-Nelson, Dayan, and Dolan, 2018; Kurth-Nelson et al., 2016) in mind, and showed comparison results with TDLM in Figure 3. But we agree we should also compare TDLM with other replay detection methods, especially from rodent literature. This is also requested by Reviewer 1 (Question 2). I

In response, we have performed detailed comparisons in simulation and real data with other techniques. Since the response covered many pages, we have not reproduced these here. We very much hope the reviewer will understand, and we refer the response to Reviewer 1 (Question 2).

In the revised manuscript. we provide a whole new section titled “TDLM for rodent replay” in the main text that addresses this question. We also include a supplemental note “Supplementary Note 5: Applying TDLM to candidate replay events” for detailed simulations.

In brief, TDLM is a more general method that assesses statistical likelihood of certain transitions on an arbitrary graph, rather than testing for a parametric relationship between time and decoded positions in a linearized space (Author response images 1 and 2). While TDLM, in its original form, is sensitive to skipping states (Author response image 4), its hierarchal version – multi-scale TDLM can deal with continuous state space. To show this, we have applied multi-scale TDLM to rodent replay data described in Ólafsdóttir et al. (2016) (Figure 7A-C, Author response image 5). Moreover, its GLM framework makes it straightforward to answer second-order statistical questions, e.g., is replay strength stronger in early vs. late sleep stage (Author response image 5).

2. The scope or generality of the proposed method should be made more explicit in a number of ways. First, it seems the major example is from MEG data with a small number of discrete states; how does the method handle continuous variables and larger state spaces? (The rodent ephys example could potentially address this but not enough detail was provided to understand what was done; see specific comments below.) Second, it appears this method describes sequenceness for a large chunk of data, but cannot tell whether an individual event (such as a hippocampal sharp wave-ripple and associated spiking) forms a sequence not. Third, there is some inconsistency in the terminology regarding scope: are the authors aiming to detect any kind of temporal structure in neural activity (first sentence of "Overview of TDLM" section) which would include oscillations, or only sequences? These are not fatal issues but should be clearly delineated.

We thank the reviewer for helpful suggestions.

1. States:

In this paper, we used 4 or 8 discrete states that formed one or two sequences. The number of states is not a constraint of the method, although decoding accuracy might drop gradually as a function of the number of states to be decoded, which could make the sequence analysis nosier. But this is a signal to noise ratio problem, not a problem of the sequence detection method itself.

Continuous spaces are amenable to TDLM by simply chunking the space into discrete states. It is the case that TDLM in its original form may potentially be less sensitive for such analyses than techniques that build-in assumptions about the spatial lay-out of the state space. This is because TDLM works on a graph, and has no information about the Euclidean nature of the state space. Techniques that make assumptions about linear structure benefit from these prior assumptions. For example, detecting state 1 then state 5 then state 10 counts as replay in these techniques, but this is not so in TDLM.

This divergence can be alleviated by a simple adjustment to TDLM to render it multi-scale – i.e., by chunking the continuous Euclidean space into a discretisation of different scales, computing TDLM sequenceness separately at each scale, and then taking the precision weighted average across scales to recover a single sequenceness measure. It is easy to see why this addresses the concerns hinted at above, as some scales will capture the 1→2→3 transitions, whilst others will capture the 1→10→20 transitions. Because the underlying space is continuous, we can average results of the same *speed* together, and this will reinstate Euclidean assumptions. Applying multi-scale TDLM, we show it can deal with skipping states as well as a continuous state space in both simulation (Figure 7, Author response image 5) and in rodent data analysis (Author response image 6). Indeed, this enables us to measure the speed of replay in cm/s.

**Author response image 6. sa2fig6:** Circular time shift vs. state identity-based permutation on real human whole brain MEG data. (**a**) The permutation (blue) is done by circularly shifting the time dimension of each state on the decoded state space of the MEG data during pre-stimuli resting time, where the ground truth is no sequence. There is a false positive. (**b**) The permutation (blue) is done by randomly shuffling the state identity. There is no false positive.

2. Time length: The TDLM method can be applied to a single ripple event, though it is true that the sequence estimate is more accurate if applied to aggregated ripple events. This is because we have more samples and consequently more states are likely be activated in the analysis time frame. We argue this is a strength rather than weakness compared to traditional methods. Most of the time, we care about whether there are significant sequences in general rather than within a specific ripple event. Most existing methods assess sequence strength within ripple events, because they either search for a best fitting line or for correlations between time and state. Those methods cannot deal with multiple sequences, while TDLM can, because TDLM is instead looking for the averaged evidence for certain transitions (Appendix 5—figure 1). Furthermore, many interesting questions rely on comparing replay across different situations (i.e., “second-order questions”). We argue this is more naturally done in a linear framework, such as TDLM, which can compute an aggregate measure across all instances of each situation and simply compare these aggregates than the alternative, which requires counting events that cross threshold criteria and comparing the counts across conditions.

3. Application scope: TDLM is designed to detect sequences alone, not oscillations, and in some cases, we deliberately control for neural oscillations to enable sequence detection. We now make this clearer in the revised manuscript.

We have added this new section “generality of TDLM” in the revised manuscript.

**“**Generality of TDLM

We have so far discussed the applicability of TDLM in relation to human MEG, as well as in rodent electrophysiology (with comparisons to standard replay detection methods). and[…] We expect this generality will also be important as rodent replay experiments move beyond 1D tracks, for example to foraging in 2D, or in complex mazes.”

3. The inference part of the work is potentially very valuable because this is an area that has been well studied in GLM/multiple regression type problems. However, the authors limit themselves to asking "first-order" sequence questions (i.e. whether observed sequenceness is different from random) when key questions – including whether or not there is evidence of replay – are actually "second-order" questions because they require a comparison of sequenceness across two conditions (e.g. pre-task and post-task; I'm borrowing this terminology from van der Meer et al. Proc Royal Soc B 2020). The authors should address how to make this kind of comparison using their method.

We thank the reviewer for appreciating our use of GLM framework for statistical inference. This first-order and second-order distinction is helpful. We are also grateful for pointing us towards the relevant literature. Based on van der Meer, Kemere, and Diba (2020), the second-order questions concerns (1) the comparison between forward and backward replay of the same sequence, (2) sequence events in different time, and (3) sequence strengths between different representational contents. The potential problems in addressing the second-order questions are: (a) Biases in cell sampling; (b) Biases in behavioural sampling; (c) Non-stationary tuning curves.

In the linear framework, the machinery for computing these second-order inferences is simple. We can just subtract the sequenceness measure between conditions. Indeed, in the original version of this manuscript there were various examples of this procedure for quantifying (Forwards – Backwards) replay. In the new version, we show a similar thing in rodent data, comparing between early-, mid- and late sleep (Figure 7, Author response image 5).

Whilst the machinery is simple, biases may still exist. We would, however, like to take the liberty of bringing some insights from the human neuroscience world. In human experiments, this type of problem is addressed as follows:

1. Ensure that biases that might occur within subjects will not occur consistently in the same direction across subjects (e.g., by randomising stimuli across participants)

2. Compare across conditions in each subject, to give a summary measure of the comparison in each subject.

3. Perform random effects inference across the population. That is, infer against the between-subject variance.

If this procedure is followed, then it can ensure that any biases that might be present at an individual experimental subject level will not cause false positives for a population inference.

We realise that such a solution might not always be possible in rodent studies which may not have the sample size necessary for random effects inference, or where a large proportion of the recorded cells may come from a small portion of animals. However, we think that as largescale recordings become cheaper, such practices will become more common.

In the meantime, it will be necessary to be vigilant to these biases in rodent experiments that use TDLM, as is the case with other techniques.

In the human data we present here, we believe that the analogous problems are all controlled for by the approach described above. We randomised stimuli and included strict controls (for example comparing rewarded and neutral sequences in Liu et al. 2019).

We have now added the following point in the revised manuscript:

“Considerations on second-order inferences

We can consider making a distinction between levels of inference, following van der Meer et al. (2020).[…] We can subtract the sequenceness measure between conditions.”

Reviewer #3:The methods used by the authors seem like potentially really useful tools for research on neural activity related to sequences of stimuli. We were excited to see that a new toolbox might be available for these sorts of problems, which are widespread. The authors touch on a number of interesting scenarios and raise relevant issues related to cross-validation and inference of statistical significance. However, given (1) the paucity of code that they've posted, and its specificity to specific exact data and (2) the large literature on latent variable models combined with surrogate data for significance testing, I would hesitate to call TDLM a "framework". Moreover, in trying to present it in this generic way, the authors have muddled the paper, making it difficult to understand exactly what they are doing.Overall: This paper presents a novel approach for detecting sequential patterns in neural data however it needs more context. What's the contribution overall? How and why is this analysis technique better than say Bayesian template matching? Why is it so difficult to understand the details of the method?

We thank the reviewer for the positive feedback. We are sorry for lack of context and code. We now provided greater context and addressed specific concerns in detail in the following section.

We would like to keep calling TDLM a “framework” given it is based on the general linear model that can allow flexible and iterative use to meet different aims. Many different questions about sequences can be embedded in the same simple set-up. The set-up accounts for common problems and gives general solutions to questions such as inference. As noted by both reviewer 1 and reviewer 2, this paper aims to develop a domain-general sequence detection method that is not tied to any specific data modality (in this revision, we will show its applicability to detect replay within ripple events in rodents, and human EEG data as well as the MEG and theta sequences that were present in the original manuscript).

We agree with the reviewer that the paper needs more context. We provide this in the revised manuscript (also detailed in the response to specific concerns).

We believe the overall contribution of this paper is our introduction of a domain general sequence detection method under a GLM framework.

We have now compared TDLM with the existing methods in the rodent research, and shown the applicability of TDLM in both simulation, and real data (both rodent ephys data and human MEG and EEG, as a response to Reviewer 1 Question 1).

To make our points concrete, we re-analysed rodent electrophysiology data from Ólafsdóttir et al. (2016) using our approach, and shown what TDLM can offer uniquely compared to existing “line searching” methods in rodent replay analysis. Please see details in section “TDLM for rodent replay” in the revised manuscript, also our response to reviewer 1 question 2.

We apologise for not being clear in the original submission. We have revised the paper accordingly based on reviewer’s suggestions.

The first and most important problem with this paper is that it is intended (it appears) to be a more detailed and enhanced retelling of the author's 2019 Cell paper. If this is the case, then it's important that it also be clearer and easier to read and understand than that one was.

We apologies for not being sufficiently clear. It is true we have dominantly used data from our previous paper as examples, but we emphasise that this current paper is not intended as an extension of our previous paper. Rather, we want to expand on the method, and develop a domain general sequence detection method for wider use. In the revision, we have now shown its applicability across multiple datasets and have extended TDLM to work on a continuous state space, e.g., physical space, as mostly used in rodent literature.

However, we understand where the reviewer is coming from and we hope we make our aim clearer in the revised manuscript. In the following section, we address specific concerns and follow the reviewer’s suggestion wherever we can in order to make the paper more accessible.

The authors should follow the normal tradition in computational papers:1. Present a clear and thorough explanation of one use of the method (i.e., MEG observations with discrete stimuli), then present the next approach (i.e., sequences?) with all the details necessary to understand it.

Thanks for the suggestion. We have largely rewritten the paper, with the same order as per the description, but also are more precise and provide greater detail in each section. We are confident it is now better organized to explain our thinking at each step of TDLM, and its application scope.

2. The authors should start each section with a mathematical explanation of the X's – the equation(s) that describes how they are derived from specific data. Much of the discussion of cross validation actually refers to this mapping.

We thank the reviewer for this helpful suggestion. We now include both the mathematical explanations and concrete examples of states (i.e., X) for each definition wherever appropriate. We also include an algorithm box for the process of cross validation in the section “Supplementary Note 2: Pseudocode of sensory code and abstract code cross-validations” in the revised manuscript.

The revised texts on this section reads:

**“**Getting the states

As described above, the input to TDLM is a set of time series of decoded neural representations, or states.[…] This more sophisticated use of TDLM merits its own consideration and is discussed in section “Sequences of sequences” in the supplemental material (Supplementary Note 1: Extension to TDLM).”

3. Equation 5 also needs a clearer explanation – it would be better to write it as a sum of matrices (because that is clearer) than with the strange "vec" notation. And TAUTO, TF and TR should be described properly – TAUTO is "the identity matrix", TF and TR are "shift matrices, with ones on the first upper and lower off diagonals".

Thanks for the suggestion. We have done so in the revised manuscript.

This part in the revised manuscript now reads:

**“**Testing the hypothesized transitions

The first level sequence analysis assesses evidence for all possible state-to-state transitions. […] Repeating the regression of Equation 5 at each time lag (∆𝑡 = 1, 2, 3, …) results in time courses of the sequenceness as a function of time lag (e.g., the solid black line in Figure 1f), in which 𝑍_F_, 𝑍_B_ are the forward and backward sequenceness respectively (e.g., red and blue lines in Figure 1g).”

4. The cross validation schemes need a clear description. Preferably using something like a LaTeX "algorithm" box so that they are precisely explained.

Thanks for the suggestion. We have included the algorithm box for the process of the cross validation in the section “Supplementary Note 2: Pseudocode for sensory code and abstract code cross-validations” in the revised manuscript.

“In the consideration of the formatting, we have attached the Latex-based algorithm box in a picture form.”

Recognizing the need to balance readability for a general reader and interest, perhaps the details could be given for the first few problems, and then for subsequent results, the detail could go into a Methods section. Alternatively, the methods section could be done away with (though some things, such as the MEG data acquisition methods are reasonably in the methods).

Thanks. We have done so in the revised manuscript.

Usually, we think about latent variable model problems from a generative perspective. The approach taken in this paper seems to be similar to a Kalman filter with a multinomial observation (which would be equivalent to the logistic regression?), but it's unclear. Making the connection to the extensive literature on dynamical latent variable models would be helpful.

Thanks for this suggestion. It is not a standard latent variable model, but it is related. The relevant latent variable model is not a Kalman filter (which tracks continuous variables), but instead a Hidden Markov Model (which tracks the transitions between discrete states).

However, TDLM is different from a traditional HMM as it:

1. Does not need to estimate the emission matrix from the test data, as it gets it from independent training data.

2. Does not assign every timepoint to a state, as the classifiers do not sum to 1.

3. Needs to estimate many transition matrices – 1 for each potential time lag.

As the reviewer points out, we could use the Bayesian generative model that underlies an HMM and account for these changes. Indeed, we are proponents of Bayesian modelling in much of our other work. However, in this situation we think the costs of the Bayesian approach outweigh its potential benefits.

The benefits of doing so would be a small reduction in state uncertainty that would come with using the estimated transition dynamics as prior distributions on the states.

The costs of doing so would be to lose the linear framework. This would mean

a. We would lose access to well-studied and well-behaved test statistics for inference.

b. We would lose flexibility in the available strategies for controlling for autocorrelations.

c. We would dramatically increase computation time (which is important for point 3 as we are estimating the model many times).

We have thought carefully about this, and we are unsure whether making the analogy to an HMM is really useful, except for methodological experts. We have chosen not to do so in the paper, but if the reviewer feels strongly, we are happy to introduce a section. We note that this answer will be available to experts on bioRxiv (and on the *eLife* website with the paper if it is published).

References:

Buzsáki, G. (2002). Theta oscillations in the hippocampus. Neuron, 33(3), 325-340.

Chadwick, A., van Rossum, M. C., and Nolan, M. F. (2015). Independent theta phase coding accounts for CA1 population sequences and enables flexible remapping. *eLife*, 4, e03542.

Davidson, T. J., Kloosterman, F., and Wilson, M. A. (2009). Hippocampal replay of extended experience. Neuron, 63(4), 497-507.

Denovellis, E. L., Gillespie, A. K., Coulter, M. E., Sosa, M., Chung, J. E., Eden, U. T., and Frank, L. M. (2020). Hippocampal replay of experience at real-world speeds. bioRxiv.

Dragoi, G., and Tonegawa, S. (2011). Preplay of future place cell sequences by hippocampal cellular assemblies. Nature, 469(7330), 397.

Dragoi, G., and Tonegawa, S. (2014). Selection of preconfigured cell assemblies for representation of novel spatial experiences. Philosophical Transactions of the Royal Society of London B: Biological Sciences, 369(1635), 20120522.

Eldar, E., Bae, G. J., Kurth-Nelson, Z., Dayan, P., and Dolan, R. J. (2018).

Magnetoencephalography decoding reveals structural differences within integrative decision processes. Nature human behaviour, 2(9), 670-681.

Fyhn, M., Hafting, T., Treves, A., Moser, M.-B., and Moser, E. I. (2007). Hippocampal remapping and grid realignment in entorhinal cortex. Nature, 446, 190.

Grosmark, A. D., and Buzsáki, G. (2016). Diversity in neural firing dynamics supports both rigid and learned hippocampal sequences. Science, 351(6280), 1440-1443.

Harris, K. D. (2020). Nonsense correlations in neuroscience. bioRxiv.

Higgins, C. (2019). Uncovering temporal structure in neural data with statistical machine learning models. University of Oxford.

Higgins, C., Liu, Y., Vidaurre, D., Kurth-Nelson, Z., Dolan, R., Behrens, T., and Woolrich, M. (2020). Replay bursts in humans coincide with activation of the default mode and parietal α networks. Neuron.

Karlsson, M. P., and Frank, L. M. (2009). Awake replay of remote experiences in the hippocampus. Nature neuroscience, 12(7), 913.

Kurth-Nelson, Z., Economides, M., Dolan, Raymond J., and Dayan, P. (2016). Fast Sequences of Non-spatial State Representations in Humans. Neuron, 91(1), 194-204.

Lee, A. K., and Wilson, M. A. (2002). Memory of sequential experience in the hippocampus during slow wave sleep. Neuron, 36(6), 1183-1194.

Liu, Y., Dolan, R. J., Kurth-Nelson, Z., and Behrens, T. E. J. (2019). Human replay spontaneously reorganizes experience. Cell, 178(3), 640-652.

Liu, Y., Mattar, M. G., Behrens, T. E. J., Daw, N. D., and Dolan, R. J. (2020). Experience replay supports non-local learning. bioRxiv. doi:10.1101/2020.10.20.343061

Maboudi, K., Ackermann, E., de Jong, L. W., Pfeiffer, B. E., Foster, D., Diba, K., and Kemere, C. (2018). Uncovering temporal structure in hippocampal output patterns. *eLife*, 7, e34467.

Nádasdy, Z., Hirase, H., Czurkó, A., Csicsvari, J., and Buzsáki, G. (1999). Replay and time compression of recurring spike sequences in the hippocampus. Journal of Neuroscience, 19(21), 9497-9507.

Ólafsdóttir, H. F., Carpenter, F., and Barry, C. (2016). Coordinated grid and place cell replay during rest. Nature neuroscience, 19(6), 792.

Schuck, N. W., and Niv, Y. (2019). Sequential replay of nonspatial task states in the human hippocampus. Science, 364(6447), eaaw5181.

Sun, C., Yang, W., Martin, J., and Tonegawa, S. (2020). Hippocampal neurons represent events as transferable units of experience. Nature neuroscience, 1-13.

Tingley, D., and Peyrache, A. (2020). On the methods for reactivation and replay analysis. Philosophical Transactions of the Royal Society B, 375(1799), 20190231.

Toft, P. A. (1996). The Radon transform-theory and implementation.

van der Meer, M. A., Kemere, C., and Diba, K. (2020). Progress and issues in second-order analysis of hippocampal replay. Philosophical Transactions of the Royal Society B, 375(1799), 20190238.

Wimmer, G. E., Liu, Y., Vehar, N., Behrens, T. E. J., and Dolan, R. J. (2020). Episodic memory retrieval success is associated with rapid replay of episode content. Nature neuroscience.

Wittkuhn, L., and Schuck, N. W. (2020). Faster than thought: Detecting sub-second activation sequences with sequential fMRI pattern analysis. bioRxiv.

Zhang, K., Ginzburg, I., McNaughton, B. L., and Sejnowski, T. J. (1998). Interpreting neuronal population activity by reconstruction: unified framework with application to hippocampal place cells. Journal of neurophysiology, 79(2), 1017-1044.

[Editors’ note: what follows is the authors’ response to the second round of review.]

While impressed by the novelty and potential utility of the method, the reviewers had a specific critical concern. Namely, do high sequenceness scores truly capture activation of patterns that widely span the specified sequence space (i.e., many complete "ABCDE" sequences) or only a collection of pairwise correlations (i.e., "AB", "DE", "BC")? Presenting more examples from experimental data that demonstrate the former was perceived as critical to demonstrate the utility of TDLM as an approach for "replay detection". Ultimately, the reviewers reached a consensus that in the current presentation, what TLDM actually detects remains opaque, and the impact of the work is diminished. We considered requesting an action plan, but upon reflection, I think that the main issue is one of semantics. However, if you wish to describe TDLM as a method for detecting "replay", it needs to be critically addressed.In addition to the reviews below, in our discussion, one reviewer noted: "even though they now explain in more detail what they did to analyze theta sequences, the result (~12 Hz) is still seemingly at odds with the ~8-9 Hz repetition one would expect from the literature. I'm actually not sure this adds a whole lot to the paper so I think it would be better to just take this part out."

Thanks for the suggestion.

1. We have now carefully gone through the paper and ensure it is clear that we are using TDLM to measure the average replay strength over the whole measured time, rather than single replay events.

For example, in the introduction:

“Here we show TDLM is suited to measure the average amount of replay across many events (i.e., replay strength) in linear modelling. […] Applying TDLM on non-invasive neuroimaging data in humans, we, and others, have shown it is possible to measure the average sequenceness (propensity for replay) in spontaneous neural representations ^1-4^.”

In the section on applying TDLM to rodent replay:

“Note, TDLM is applied directly to the concatenated rather than individual replay events. […] Because TDLM addresses statistical questions in linear modelling, it does not require secondary statistics to ask whether the “counts” of individual events are more likely than chance, or more likely in one situation than another.”

In the discussion:

“TDLM adds a new analysis toolkit to the replay field. It is especially suited for summarising replay strength across many events, for comparing replay strength between conditions, and for analysing replay strength in complex behavioural paradigms. […] Unlike alternative tools, we have not shown TDLM applied to individual replay events.”

2. Also following suggestion, we have deleted the section regarding single replay event analysis in both the main text and the supplementary note 5 (we assume this is what the reviewers suggested given the original supplementary figure 5 is replay analysis on human EEG data).

3. We have made it clear in the discussion that “TDLM is an *addition* to the analysis toolboxes of the field rather than a *replacement*”.

“TDLM adds a new analysis toolkit to the replay field. It is especially suited for summarising replay strength across many events, for comparing replay strength between conditions, and for analysing replay strength in complex behavioural paradigms. […] Unlike alternative tools, we have not shown TDLM applied to individual replay events.”

4. We have dropped the theta sequence analysis in the supplementary note 1 and have updated the figures accordingly.

Reviewer #1:Overall, I find great value in the effort to provide researchers working with very different animal models and datasets a similar toolkit to apply and analyze reactivation and replay. But I also have significant concerns about the potential for these methods, if poorly understood and applied, to further confound the field. Fully understanding this paper and the described methods and its caveats is not easy for most experimentalists, yours truly included. I am concerned that investigators will misapply these tools and claim significant replay in instances where there is none. These concerns may be addressable by better diagnostics and related guidance for interpretation.Nevertheless, an important caveat in the work is that it does not detect "replay" per se, but rather temporal biases in decoded activity. Thus I think the title should be amended. In some places, the authors describe this as "sequenceness", or "temporal delays" which are both preferable to "replay". Prior work (e.g. Skaggs and McNaughton) used a similar measure, but referred to it as "temporal bias". While this temporal bias is certainly related to replay and sequences, it is only an indirect measure of these, and more akin to "reactivation", as it's largely pairwise analyses. Clarity on such issues is particularly important in this area given the excessive ambiguity in terminology for the replay and reactivation phenomena.My other major concern is that the analysis is rather opaque in an area where there is much need for transparency, especially considering the existing debates and controversy surrounding appropriate methodology and conclusions that can be drawn from it. For example, in most of the figures that the authors provide, it's unclear whether the sequenceness score is driven by one particular pair of stimuli, or equally so among most possible pairs. Perhaps a transition graph could be composed from the data, but I did not find this except in cartoon diagrams. I think it would be important for the authors to provide guidance regarding these details. A related question is whether these biased pairs typically appear in isolation, or as part of larger multi-item events? It's not clear if there is a straightforward way to extract this important information. Some sample events could be shown, but examples can also be cherry-picked and non-representative. Probably a histogram of sequence bout lengths would be valuable.

Thank you for the great comments. The first two questions were the subject of a query we sent, which is copied below, and the response to this query is addressed above.

“Below is our take on the discussion. If you find this persuasive, then we are happy to do exactly as suggested by the reviewers on all other points. If you do not find this persuasive, then we are a bit stuck. So, we would like to understand the position of the reviewers before proceeding.

Here is what we think:

1. We are sorry if it seems opaque what the method is measuring. It is, in fact, mathematically defined, and we try to state it clearly here: We are measuring whether B is more likely to be activated at a certain time lag after A, compared to the average chance of B being activated in the rest of the data. We are testing against the null hypothesis that the frequency of B does not depend on the previous activation of A (Averaged across all pairs, AB, BC, CD ...).

2. If you want us to call the measure “sequenceness” instead of “replay” that is fine with us. However, we ask that you might consider the following points before coming to this conclusion (Note it is *not* the same as reactivation, as it does require a sequence of at least 2 elements).

The measure might be construed as sequenceness as opposed to replay because it only relies on AB, BC, CD as you say. However:

i. Importantly, every technique that we are aware of for detecting replay on the decoded state space has the same issue, or a more pronounced version of this issue. For example, the radon transform method will return a statistically significant result for “replay” if it detects AB (alone). It will also declare replay for AC alone, or AD alone or AE alone. We demonstrate this in supplementary figure 8. Any technique that asks, “Is there more replay than chance?” will return a positive for any of these situations. Nevertheless, they are described as “replay detection methods”.

ii.Like the radon transform, or linear correlation, or all other techniques, TDLM will return a stronger measure of replay if there is ABCD than if there is only AB. This is particularly true of multiscale TDLM.

iii. Uniquely, as far as we know TDLM does allow you to test for ABCD because, in the linear framework, it is possible to test for the interaction of AB, BC, CD. This asks “Does an AB pair precede a BC pair. Etc. We describe this in section “Multi-step sequences”. Again, we can test this formally in the linear modelling framework. This section really *does* provide a method for detecting ABCD.

We therefore think, on this issue, there is no difference between TDLM and all other methods that have previously been described as detecting “Replay”, except that there is a setting in which TDLM will require the whole sequence, but there is not for other techniques.

However, there is a difference between what we have demonstrated with TDLM and other methods. Whilst it is conceptually possible, we have not demonstrated that TDLM can detect individual replay events. Instead, we have shown that TDLM can measure the average amount of replay across many events (for example all events in a sleep session, or in early sleep). We note, the sleep dataset is a real dataset in rodent hippocampal electrophysiology, it is from “Ólafsdóttir, H. F., Carpenter, F., & Barry, C. (2016). Coordinated grid and place cell replay during rest. Nature neuroscience”.

We’re afraid we are not going to change this. We believe that this is a more rigorous approach, more amenable to formal statistics and less likely to suffer the statistical problems that have plagued the replay literature to date. It does not require secondary statistics to ask whether the “Counts” of individual events are more likely than chance, or more likely in one situation than another (as highlighted by reviewer 2). We would therefore like to encourage the field to take this approach.

There is one other point we would like to raise. Reviewer 1 is concerned that we might further confound the experimental community with this method. We do not think this is true. Having developed a number of methods, this concern has been raised on many occasions. We have been advised that it is important to keep methods to the experts who understand them. We have instead found that by far the best way for a field to understand a method is for the authors to release it transparently to the experimental community, to try to support their work and to listen to their requests and criticisms. Then the strengths and weaknesses of a method become clear. We hope to do the same with this method.

Overall, therefore, we don’t think there is a strong reason, by comparison to other methods, for limiting the use of the word “replay”, but we are happy to do so if you disagree. We are happy to call it sequenceness. However, we are not going to analyse individual events.”

Part of the claimed value in these methods is their possible application to spike trains, e.g. from rodents, particularly for detecting and describing individual events. The authors claim that this is possible. However, while they analyze two rodent datasets in different parts, they do not apply it to any real data, but only on rather contrived (low noise) simulated data. This raises the concern that TDLM is not sufficiently sensitive for detecting individual events. The theta sequence analysis results shown in Supplementary Figure 3d are also concerning to me. They show a very noisy signal that passes threshold in one particular bin. If such a figure were a major result in a new manuscript, and the current eLife manuscript was cited as validation for the methods, would reviewers be obliged to accept it for publication? If not, what additional criterion or diagnostic steps would be recommended?

We hope that the changes in the manuscript demonstrate that we think there is value in an approach that measures replay across entire conditions and provides a natural framework for comparing between conditions, or between types of replays e.g., forward vs backward. We are now much clearer about how linear modelling can tackle these problems. See responses to Reviewer 2 below. We are also much clearer that we think that this method provides an additional tool, not replacing existing tools (see text above).

Comments for the authors:P2, Line 32: "and" seems misplaced.P2, Line 34: "periods of activity".P3, Line 4: perhaps "single neuron" rather than "cellular".P4, Lines 34-35: it is not clear here what the authors mean by "over and above variance that can be explained by other states at the same time." It gets more clear later in page 11, section "Moving to multiple linear regression". The authors might consider either referring to that section at the end of the sentence or removing the unnecessary details that might confuse the reader at this juncture.P5, Line 31: This sentence is a bit confusing. It is not clear what "in which" refers to. It might be better to start a new sentence for describing Zf and Zb.P6, Lines 7-8: The authors might refer to the section "Correcting for multiple comparisons" on page 16, where more details are provided.

Thank you very much indeed, we have made the changes following your suggestions.

P8, Lines 42-46: the description of abstraction may benefit from additional background and exposition.P9, Line 18-24, I found this entire paragraph very confusing.

Thanks very much indeed for these comments. This whole section was confusing and overly detailed. It was perhaps appropriate for the original paper where we were aiming predominantly at human researchers, but we agree it now adds confusion. We have dramatically shortened it (as pasted here), and referred to the supplement and the Liu et al. 2019 paper for details that may be important to a small minority of human researchers. We hope this new paragraph is clearer.

This paragraph now reads:

“As well as sequences of sensory representations, it is possible to search for replay of more abstract neural representations. […] One way that excludes the possibility of sensory contamination is if the structural representations can be shown to sequence before the subjects have ever seen their sensory correlates ^4^.”

P10, Line 7: "that share the same abstract representation" is redundant.P10, Line 12: "tested" should be corrected to test it.P10, Lines 23-24: Confusing sentence with multiple "whiches".

Thanks again, we have now deleted this whole section and replaced with the paragraph above.

P12, Line 24: Is it possible for the auto-correlation structure of Xi(t) to generate false positives for Xj(t+dt), or would this necessarily be entirely accounted for by Xj(t)?

We are not sure we understand this question. If any previous measurement from i predicts j(t+dt) better than all previous j(t), then this can’t be just because of autocorrelation (because the measurement we are predicting is j, not i). It needs to be (lagged) crosscorrelation. This lagged cross-correlation is what we are trying to measure. Sorry if this does not quite get at your question, but we are struggling to understand.

P13, Lines 23-24: How the regularization (penalizing for the norm of W) is supposed to make the model "trained on stimulus-evoked data" generalize better to off-task/rest data? The authors might add a sentence or two at the end of the paragraph to make the aim more clear and motivate the next paragraphs in the section. Based on the descriptions in the first paragraph of Page 14, the regularization seems to add more sparsity to the estimated W that minimizes spatial correlations between the states, etc. Something similar to these descriptions could be used here.

Thanks, we have added a sentence at the end of the paragraph to make it clear. It now reads:

“A key parameter in training high dimensional decoding models is the degree of regularization. […] Here it has the added potential benefit of reducing spatial correlation between classifier weights.”

P13, Line 28: It is not exactly clear what authors mean by "the prior term"? Does it refer to the equation before adding any regularization or to some prior probability distribution over W? I think in this context, we should be cautious with using the words like prior, posterior, etc.

Thanks for the suggestion, we have now removed the “prior term”, and changed it to “regularization term”.

P16, Line 30, extra "as".

Thanks for pointing it out!

P17, section "Considerations on second-order inferences". It seems that this section should be placed later in the manuscript, in the section "TDLM for rodent replay".

Thanks, we have made the change accordingly.

P23 Line 39, missing "that".

Thanks, we have added it in the next text.

Supplementary Note 5: what shuffle was used?

It is randomly shuffling the rate map. In the revised manuscript, we have deleted the single replay event analysis as per request.

Reviewer #2:[…] There is one important area that the paper still falls short on, but that should be straightforward to address with some simple additional analysis and discussion. A distinctive strength of the GLM framework is that it easily accommodates testing important and ubiquitous "second-order" questions, such as whether there is more forward than reverse replay, is replay strength changing over time, and so on. However, the message in the current version that one could simply subtract sequenceness scores doesn't address how one would formally test for a difference, or test for some factor like time being important. For forward vs. reverse, because this is fit to the same data, this is a comparison between different betas in the second-level GLM (Figure 1f). I am not a statistician, but my textbooks say there are a few ways of doing this, for instance z = (β_fwd_ – β_rev_) / σ_(β_fwd_ – β_rev_) where the crucial variance term can be estimated as the sqrt(σ(β_fwd_)^2^ + σ(β_rev_)^2^), matching the intuition that a formal test requires an estimate of variance of the difference between the betas, not just the means.For early vs. late sleep, things are more complicated because you are not comparing models/betas fit to the same data. I suppose the above approach could also work as long as all the betas are standardized, but wouldn't a better test be to include time as a regressor in the second-level GLM and formally test for an interaction between time and T_fwd_ (and T_bwd_)?In either case, there are two important points to make to demonstrate the utility of the GLM framework for sequence analysis: (1) these ubiquitous second-order inference questions require a bit more consideration than just saying you can subtract the sequenceness scores; there is another level of statistical inference here, and (2) the TDLM framework, unlike many other approaches, is in fact very well suited to doing just that – it benefits from the existing machinery of linear statistical modeling and associated model comparison tools to make such comparisons.These points are not currently clear from the "Considerations on second-order inferences" section. If in addition the GLM framework allows for the regressing out of some potential confounds that can come up in rodent data that the authors now reference that's an additional benefit but not the main point I am asking the authors to speak to.

We are delighted to know the reviewer appreciates our revision effort, and thanks again for all the insightful comments.

We agree that simply subtracting sequenceness scores does not constitute a formal statistical testing procedure. We also agree that we can rely on classical parametric tests, e.g., paired t test for comparison of replay strength between forward and backward replay, as long as there is no multiple comparison problem. If we do not have a replay speed of interest a priori, we will have to control for multiple comparisons (Figure 8A, right panel, pink line), as have discussed in section “Statistical inference”. The reviewer is also correct that we can directly model the time effect as TDLM is a linear framework. In the revised manuscript, we also used the test of “time varying effect on replay strength” as an example to illustrate how TDLM is approaching the second-order questions.

New text is copied below:

**“**Second order inferences.

As pointed out by van der Meer, et al. ^19^, there are two types of statistical questions: a "first-order" sequence question, which concerns whether an observed sequenceness is different from random (i.e., do replays exist?); and a “second-order” question, which requires a comparison of sequenceness across conditions (i.e., do replays differ?). […] There is no selection bias in performing statistics on the difference of sequence effects, or effects relating to time (green rectangle).”

References:

Wimmer, G. E., Liu, Y., Vehar, N., Behrens, T. E. J. and Dolan, R. J. Episodic memory retrieval success is associated with rapid replay of episode content. Nature Neuroscience 23, 1025–1033 (2020).

1. Nour, M. M., Liu, Y., Arumuham, A., Kurth-Nelson, Z. and Dolan, R. Impaired neural replay of inferred relationships in schizophrenia. Cell in press (2021).

2. Liu, Y., Mattar, M. G., Behrens, T. E. J., Daw, N. D. and Dolan, R. J. Experience replay is associated with efficient non-local learning. Science in press (2021).

3. Liu, Y., Dolan, R. J., Kurth-Nelson, Z. and Behrens, T. E. J. Human replay spontaneously reorganizes experience. Cell 178, 640-652 (2019).

4. Fyhn, M., Hafting, T., Treves, A., Moser, M.-B. and Moser, E. I. Hippocampal remapping and grid realignment in entorhinal cortex. Nature 446, 190 (2007).

5. Dehaene, S., Meyniel, F., Wacongne, C., Wang, L. and Pallier, C. The neural representation of sequences: from transition probabilities to algebraic patterns and linguistic trees. Neuron 88, 2-19 (2015).

6. Kriegeskorte, N., Simmons, W. K., Bellgowan, P. S. and Baker, C. I. Circular analysis in systems neuroscience: the dangers of double dipping. Nature neuroscience 12, 535 (2009).

7. Ólafsdóttir, H. F., Carpenter, F. and Barry, C. Coordinated grid and place cell replay during rest. Nature Neuroscience 19, 792 (2016).

8. Wilson, M. A. and McNaughton, B. L. Reactivation of hippocampal ensemble memories during sleep. Science 265, 676-679 (1994).

9. Skaggs, W. E. and McNaughton, B. L. Replay of neuronal firing sequences in rat hippocampus during sleep following spatial experience. Science 271, 1870-1873 (1996).

10. Louie, K. and Wilson, M. A. Temporally structured replay of awake hippocampal ensemble activity during rapid eye movement sleep. Neuron 29, 145-156 (2001).

11. Lee, A. K. and Wilson, M. A. Memory of sequential experience in the hippocampus during slow wave sleep. Neuron 36, 1183-1194 (2002).

12. Foster, D. J. Replay comes of age. Annual Review of Neuroscience 40, 581-602 (2017).

13. Ólafsdóttir, H. F., Bush, D. and Barry, C. The Role of Hippocampal Replay in Memory and Planning. Current Biology 28, R37-R50 (2018).

14. Pfeiffer, B. E. The content of hippocampal “replay”. Hippocampus 30, 6-18 (2020).

15. Carr, M. F., Jadhav, S. P. and Frank, L. M. Hippocampal replay in the awake state: a potential substrate for memory consolidation and retrieval. Nature Neuroscience 14, 147 (2011).

16. Lisman, J. et al. Viewpoints: how the hippocampus contributes to memory, navigation and cognition. Nature Neuroscience 20, 1434-1447 (2017).

17. Davidson, T. J., Kloosterman, F. and Wilson, M. A. Hippocampal replay of extended experience. Neuron 63, 497-507 (2009).

18. Grosmark, A. D. and Buzsáki, G. Diversity in neural firing dynamics supports both rigid and learned hippocampal sequences. Science 351, 1440-1443 (2016).

19. Maboudi, K. et al. Uncovering temporal structure in hippocampal output patterns. *eLife* 7, e34467 (2018).

20. van der Meer, M. A., Kemere, C. and Diba, K. Progress and issues in second-order analysis of hippocampal replay. Philosophical Transactions of the Royal Society B: Biological Sciences 375, 20190238 (2020).

21. Tingley, D. and Peyrache, A. On the methods for reactivation and replay analysis. Philosophical Transactions of the Royal Society B: Biological Sciences 375, 20190231 (2020).

22. Rosenberg, M., Zhang, T., Perona, P. and Meister, M. Mice in a labyrinth: Rapid learning, sudden insight, and efficient exploration. bioRxiv (2021).

[Editors' note: further revisions were suggested prior to acceptance, as described below.]

We shared the revised manuscript with the reviewers. After some discussion, it was concluded that the manuscript has been improved but there are some remaining issues that need to be addressed, as outlined below:1. The reviewers were confused by the data in Figure 7e. We finally concluded that it was an attempt to explain how the regression was formed, but it took lots of back and forth. Given that this is a tools paper, there seems to be no reason why each analysis figure can't be backed with equations that identify the regressions being done, and the variables being regressed.

Thanks for the suggestion and the close reading. It is very valuable to know what is and what is not clear. We have now expanded the text and written the linear equation.

**“**Interactions

A second method for performing second order tests is to introduce them into the linear regression as interaction terms, and then perform inference on the regression weights for these interactions.[…] This has no effect on the regression coefficients of the interaction term but, by rendering the interaction approximately orthogonal to 𝑋_k_(𝑡), it makes it possible to estimate the main effect and the interaction in the same regression.”

2. Figure 5 appears to be about analyzing the MEG data when events are detected. (Isn't TDLM an approach for measuring sequenceness over a population of events rather than finding single ones?)

Sorry for the misunderstanding. As with the rest of the manuscript Figure 5 is an average over likely reactivations. The difference is that to construct this average it gives each timepoint a score based on the instantaneous sequenceness. Some timepoints have low scores. Others have high scores. We then average the temporal frequency plot over those with high scores. We believe this is very much of the spirit of the rest of the paper. It creates one average measure for the whole timeseries. Not a measure for each event. We are sorry if this was confusing. We have amended in the text below.

Even though this is previously published work, the methods need significant expansion (see Point 1). The text refers to a section that appears to be missing? Here's the text: "We want to identify the time when a given sequence is very likely to unfold. We achieve this, by transforming from the space of states to the space of sequence events. This is the same computation as in the section "States as sequence events". " (Search for "sequence events" yields no results.) Perhaps this refers to Appendix 3, but the text there doesn't really help much.

Here is the expanded text.

“We want to identify the *time* when a given sequence is very likely to unfold, so we can construct averages of independent data over these times.[…] Note that although this method assigns a score for individual replay events as an intermediary variable, it results in an *average* measure across many events.”

3. One reviewer had previously asked "in most of the figures that the authors provide, it's unclear whether the sequenceness score is driven by one particular pair of stimuli, or equally so among most possible pairs". To clarify, it seems that the question is: if the model proposed is A→B→C→D, and the data are largely A→B, can that be detected? Or alternatively, can you give a proper way of comparing two models (in this case, A→B→C→D vs A→B)?

Thanks for the question. We realise now that there are two ways to interpret this question. In our query document, we gave an answer to the first interpretation (which we repeat below). However, maybe we got the interpretation wrong, so we also provide a new answer below. We are grateful to the reviewer for this as it has allowed us to discover something new about the rodent data (maybe others knew it already?).

Interpretation 1: Can we discern whether TDLM is measuring an average of short transitions (AB, BC, CD) or long sequences (ABCD).

Yes. This issue is dealt with in Appendix 1: Multistep sequences. As far as we know, other techniques do not do this. In TDLM, for a particular time lag, we can compute the interaction between A and B (AB) and ask if this new variable AB predicts C at the next time lag better than A and B alone, or by some other combination (XB). The procedure works for any length sequence (so includes ABCD). We have re-read this section, and we believe it is clear, but if the reviewer has specific suggestions, we would be happy to incorporate them.

Interpretation 2: Can we discern whether the transitions that are contributing to the average are disproportionately some (e.g., AB), rather than others (CD)?

Yes. TDLM measures each transition independently, so by examining these transitions before averaging them, we can see what the underlying contributions are. To demonstrate this approach, we re-examine the rodent data. We show below (and in the revised paper), that forward replay disproportionately occurs at the beginning of the track (significantly), that this is not true of backward replay (which prefers the end of the track), and that the difference is significant.

We used the same ROI we have defined previously based on the significant replay speeds (forward + backward, cf. Figure 7d, right panel, green shading). For visualization purposes, we have plotted the estimated strength for each pairwise forward sequence (Figure 8A), separately within each scale (from 1 to 4, with increasing spatial scales). The pairwise sequences are ordered from the start of the maze to the end of the maze. Alongside the pairwise sequence plot, we have plotted the mean replay strength over all possible pairwise transitions (in red), in comparison to the mean of all control transitions (in grey. As expected, they are all around 0). Note that we cannot perform inference on the difference between the red and grey bars here because they have been selected from a biased ROI. It is simply for illustration purposes. We have therefore put them in red squares to match figure 7F.

It is evident from Figure 8 that the TDLM average is not dominated by a single transition. Many transitions contribute. However, it also appears that there is a tendency for replay strength to decrease going from the start of maze to the end of maze (especially in the larger scales, which measure bigger replay movements). To formally test this, we can adopt the linear contrast approach described in the last round of revisions.

Our previous text on linear contrast read as follows:

“If we want to test whether replay increases linearly over 5 conditions [A, B, C, D, E], we can compute the linear contrast -2*A – B + 0*C + D + 2*E, (which would be zero under the null hypothesis) and perform statistics on this new measure.”

Here, we can use this exact same approach to test a linear increase or decrease across different *transitions*. We take the linear contrast weight vector, 𝑐 ([-2,-1,0,1,2] for the largest scale, [-3:3] for the next scale, [-5:5] for the next scale, [-12:12] for the smallest scale) and multiply these by the β estimates of the transitions:

𝑐𝑜𝑛𝑡𝑟𝑎𝑠𝑡 = 𝑐^0^𝛽

If this new measure, 𝑐𝑜𝑛𝑡𝑟𝑎𝑠𝑡, is different from zero, then there is a linear increase/decrease from one end of the track to the other. Note that this new contrast is no longer biased by the ROI selection as each transition contributed equally to the ROI selection, but we are now comparing between transitions. Inference on this contrast is therefore valid. We have therefore put them in green boxes to match figure 7F (Figure 8B, C).

Within the larger two scales, these contrasts are significantly negative (tested against permutations in exactly the same way as the “mean” contrasts). Since we are still in the linear domain, we can now just average these contrasts across the 4 scales and get a single measure for spatial modulation of replay. This average measure is significantly negative (Figure 8B). Hence, on average, forward replay is stronger at the beginning of the track.

We can do the same thing for backward replay. We found a opposite pattern, i.e., strength of backward replay is stronger at the end of the track, and similarly, it is not significant in the smallest scale, and become significant in the largest scale, and also significant on average across all scales (Figure 8C). Again, since we are in the linear domain, we can further contrast these contrasts, asking if this effect is different for forwards and backward replay. We found the difference is indeed significant (Figure 8D).

Note we would like to stress again, that this analysis is not about a single replay event but is testing for average differences across all replay events.

We have added this point in the revised manuscript:

“In addition to the time varying effect, we can also test the spatial modulation effect, i.e., how replay strength (at the same replay speed) change as a function of its spatial content.[…] We found the difference is indeed significant (Figure 8d). Note we would like to stress again, that this analysis is not about a single replay event but is testing for average differences across all replay events.”